# Mantis: Lightweight Foundation Model for Time Series Classification

**Vasilii Feofanov** [1 2]  **Songkang Wen** [1]  **Shifeng Xie** [1 3]  **Simon Roschmann** [4 5]  **Marius Alonso** [6 1]  **Hongbo Guo** [1]
**Romain Ilbert** [7 1]  **Malik Tiomoko** [1]  **Quentin Bouniot** [4 5 8]  **Zeynep Akata** [4 5]  **Lujia Pan** [1]  **Jianfeng Zhang** [1]
**Ievgen Redko** [1]

## Abstract

While foundation models have revolutionized various domains, their application to time series classification remains rather under-explored, with existing literature predominantly focused on forecasting. To bridge this gap, we introduce **Mantis**, a transformer-based foundation model pre-trained exclusively on synthetic data via self-supervised contrastive learning. We demonstrate that effective tokenization is critical to unlocking the full potential of transformers, proposing a novel token generator unit. Furthermore, we introduce an enhanced test-time methodology that bridges the performance gap between Mantis and strong specialized approaches by leveraging intermediate-layer representations, self-ensembling, and cross-model embedding fusion. Extensive experiments demonstrate that Mantis establishes a new state-of-the-art, outperforming existing foundation models across four diverse dataset collections covering various application domains. The source code is available at https://github.com/vfeofanov/mantis.

## 1. Introduction

Classification of time series data is a fundamental problem in both academia and industry, arising in domains such as human activity recognition (Chen et al., 2025; Li et al., 2025b), power electronics (Liao et al., 2025; Li et al., 2025a), healthcare (Alchieri et al., 2025; Wong et al., 2025), finance (Lee et al., 2023), and neuroscience (Wang et al., 2024; Gnassounou et al., 2025). Following the success of foundation models in vision (Radford et al., 2021) and language (Achiam et al., 2023), the development of time series

foundation models (TSFMs) has become an active research direction. These models aim to serve as universal feature extractors for diverse downstream tasks, reducing the need for extensive labeled data required to train specialized models and simplifying the process of model selection and tuning.

Over the past three years, a wide variety of TSFMs have been introduced. Most of them focus on forecasting, either pre-training a model from scratch (Auer et al., 2025b; Ansari et al., 2025) or re-programming foundation models designed for other modalities (Zhou et al., 2023; Chen et al., 2024). Although in principle any TSFM can be used for any time series downstream task, using a forecasting model for classification may lead to suboptimal performance because certain pre-training paradigms are inherently more suitable for discriminative tasks (e.g., masked reconstruction loss may be better suited for imputation, while contrastive loss aligns more naturally with classification). Surprisingly, despite the omnipresence of time series classification tasks (Bagnall et al., 2018; Dau et al., 2019; Dempster et al., 2020), there are very few foundation models that focus specifically on them.

To address this gap, we introduce Mantis, a new lightweight time series classification foundation model that has a strong zero-shot feature extraction performance. Mantis establishes a state-of-the-art performance on common benchmarks by leveraging the following key ideas:

1. **Time-series multi-view tokenizer.** We design a dedicated tokenizer that maps a raw sequence to a fixed number of tokens to control attention cost, by combining convolutional patch features of the instance-normalized signal, convolutional patch features of its first-order difference, and patch-wise statistics' encoding. The three branches are concatenated and linearly projected into token embeddings.

2. **Synthetic contrastive pre-training for classification.** Instead of masked reconstruction or autoregressive modelling that are often forecasting-oriented, we pre-train Mantis with contrastive InfoNCE loss (Oord et al., 2018; He et al., 2020) to explicitly increase representation separability for classification. We use CauKer (Xie et al., 2026) to generate synthetic pre-

[1]Huawei Noah's Ark Lab [2]42.com [3]Paris Cité University [4]Helmholtz Munich [5]Technical University of Munich [6]Criteo [7]Meta [8]Telecom Paris. Correspondence to: Vasilii Feofanov <firstname.lastname@gmail.com>.

*Proceedings of the $43^{rd}$ International Conference on Machine Learning*, Seoul, South Korea. PMLR 306, 2026. Copyright 2026 by the author(s).

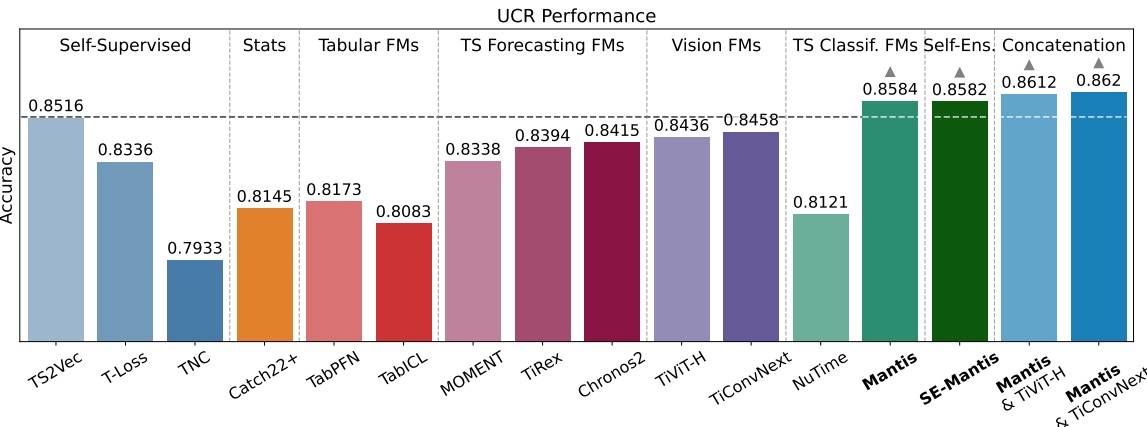

*Figure 1.* Final Performance on the UCR Benchmark.

training data, naturally matching the contrastive learning's need for broad diversity and helping to enforce an out-of-distribution generalization.

3. **Test-time optimization.** We introduce an inference-time pipeline that unlocks the potential of the pre-trained model, allowing us to reach strong generalization in a purely frozen-encoder (zero-shot) regime. We achieve this by leveraging intermediate-layer representations, improved output-token aggregation, input-perturbation self-ensembling, and cross-model embedding fusion. Jointly, these steps allow us to match the performance of strong supervised and self-supervised baselines trained on individual datasets.

Our extensive experiments on UCR (Dau et al., 2019), UEA (Bagnall et al., 2018), as well as domain-specific benchmarks including Human Activity Recognition (HAR) and EEG classification, demonstrate that Mantis consistently delivers strong performance in the frozen zero-shot regime, as shown in Figure 1.

Throughout this paper, we use the term "zero-shot feature extraction" to describe the linear probing setup and emphasize that the encoder's weights remain completely untouched (zero-shot transfer) during the downstream task adaptation. Both linear probing and zero-shot feature extraction refer to the same frozen-encoder paradigm.

**Conflict of Interest Disclosure.** V. Feofanov, S. Wen, S. Xie, M. Alonso, H. Guo, R. Ilbert, M. Tiomoko, L. Pan, J. Zhang, and I. Redko are/were employed by Huawei Noah's Ark Lab, where Mantis have been developed.

## 2. Related Work

**Forecasting.** Most of recently proposed TSFMs focus on forecasting, with existing approaches including pre-training decoder-only architectures (Das et al., 2023) or masked au-

toencoders (Goswami et al., 2024). Another approach consists in adapting large language (Zhou et al., 2023; Ashok et al., 2025) or vision models (Chen et al., 2024; Roschmann et al., 2025) to the time series modality. In terms of architecture, transformers (Cohen et al., 2025; Ansari et al., 2025; Liu et al., 2025) are a common choice, with a few exceptions such as state-space models (Bhethanabhotla et al., 2024; Graf et al., 2025) and xLSTM (Auer et al., 2025b).

**Classification.** While forecasting TSFMs can be applied to classification, these two tasks usually rely on fundamentally different cues (e.g., low-frequency global trends vs. high-frequency local patterns), which hinders models from achieving superior performance on both tasks simultaneously. This highlights the need to design foundation models tailored specifically for time series classification. In this vein, NuTime (Lin et al., 2024) stands out that pre-trains a transformer using a self-distillation approach. However, this paper has not explored the frozen-encoder paradigm, thus heavily relying on fine-tuning to achieve strong performance. Prior to the era of TSFMs, self-supervised approaches (Yue et al., 2022; Franceschi et al., 2019; Tonekaboni et al., 2021) were introduced to learn task-specific representations rather than a single universal one.

**Pre-training Data.** Most TSFMs are pre-trained on a mix of real-world datasets from various domains (Auer et al., 2025b; Lin et al., 2024), whereas Cohen et al. (2025) focus specifically on observability data. In contrast, we adopt an alternative approach that involves pre-training on synthetically generated data (Moroshan et al., 2025; Xie et al., 2026), which yields both sample efficiency and high diversity.

## 3. Proposed Method

In this section, we present the main technical details behind Mantis. including the problem setup, its architecture, the pre-training process, and test-time optimization.

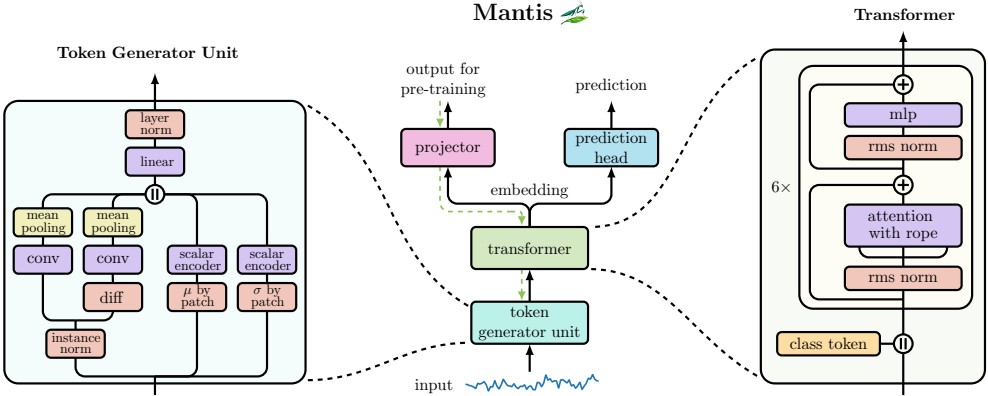

*Figure 2.* Architecture. By symbol $+$ we denote the sum operator, while $||$ designates the vector concatenation operator.

## 3.1. Problem Setup

Mathematically speaking, our time series classification foundation model is an encoder $F : \mathbb{R}^t \to \mathbb{R}^q$ that projects any time series $\mathbf{x} \in \mathbb{R}^t$ with a fixed sequence length $t$ to a discriminative hidden space $\mathbb{R}^q$. During the pre-training phase, we observe an unlabeled pre-training set $X_0$ that is sufficiently large to learn rich embeddings that generalize well across different tasks. After pre-training, using the model on a new supervised downstream task with observations X and labels Y can be done it two following ways. *Zero-shot feature extraction:* we use $F$ to extract deep embeddings $Z = \{F(\mathbf{x}),\ \mathbf{x} \in X\}$ and then learn any classifier $h : \mathbb{R}^q \to \{1, \ldots, K\}$ using Z as features and Y as corresponding labels. *Fine-tuning:* append a classification head $h : \mathbb{R}^q \to \mathbb{R}^K$ and update parameters of $h \circ F$ by minimizing a loss function evaluated on the downstream task. In this work, we primarily focus on zero-shot feature extraction. When time series with multiple channels $\mathbf{x} = [\mathbf{x}_1, \ldots, \mathbf{x}_d] \in \mathbb{R}^{d \times t},\ d > 1$, are considered, we send each channel $\mathbf{x}_i,\ i \in [1, d]$, to the TSFM independently, i.e., the embedding of $\mathbf{x}$ is defined as $\mathbf{z} = \text{concat}\,[(F(\mathbf{x}_i))_{1 \le i \le d}]$, where concat denotes the vector concatenation operator, and the input dimension of the classifier (head) is $\mathbb{R}^{d \times q}$.

## 3.2. Architecture

In this section, we describe the architecture of Mantis, which adapts the Transformer architecture (Vaswani et al., 2017) to time series data modality through a dedicated tokenization strategy. The full architecture is illustrated in Figure 2. We fix the number of tokens to 32, which implies that the input sequence length $t$ must be a multiple of 32. This design choice differs from approaches such as Lin et al. (2024) and Goswami et al. (2024), which instead fix the patch (token) length. However, fixing the number of tokens is preferable under computational constraints, as the self-attention mechanism scales quadratically with respect to the

number of tokens. As a requirement, an input time series should be resized/padded to a certain length. By default, we resize all inputs to length 512. Further analysis of the impact of the interpolation length is provided in Section B.5.

**Token Generator Unit.** The first stage of the model encodes a raw time series into a set of meaningful tokens, which are subsequently processed by the Transformer. The token generation procedure consists of the following steps:

a. *Instance normalization.* For each time series instance, we subtract the mean and divide by the standard deviation computed across time steps. This normalization is implemented as part of the model architecture and applied during the forward pass. Note that for step d, we use the raw signal bypassing the normalization.

b. *Patch extraction from the signal.* We extract patch-level representations by applying a single convolutional layer with a fixed kernel size, followed by mean pooling configured to produce 32 patches. The convolution outputs 256 channels, resulting in patch embeddings of dimension 256.

c. *Patch extraction from the first-order differential.* We apply the same patching procedure to the first-order temporal difference of the time series, computed as the difference between adjacent time steps. This differential representation encourages stationarity and reduces the influence of long-term trends.

d. *Patch-wise statistics encoding.* To preserve information about the original measurement scale, we split the raw (unnormalized) time series into 32 non-overlapping patches. For each patch, we compute the mean and standard deviation and encode these statistics using the Multi-Scaled Scalar Encoder (Lin et al., 2024).

e. *Token projection.* All patch-level features (signal, differential signal, and statistical descriptors) are concatenated and passed through a linear projection layer followed by layer normalization (Ba et al., 2016), pro-

ducing 32 tokens with dimensionality 256.

**Transformer.** The resulting tokens are processed by a Transformer encoder, as summarized below:

a. *Class token.* A learnable class token is prepended to 32 generated tokens. This token attends to all other tokens and aggregates global information, summarizing the entire input sequence.

b. *Positional encoding.* Positional information is incorporated using positional encodings. We use Rotary Positional Encoding (RoPE; Su et al., 2024) that rotates the query and key representations.

c. *Transformer layers.* We apply six Transformer layers, each consisting of multi-head self-attention with eight heads, followed by a feedforward network. All layers use a pre-normalization design.

d. *Projector and Prediction Head.* During pre-training, a projector head is appended to produce embeddings used for similarity-based objectives. For fine-tuning, we append a task-specific classification head mapping the embeddings to class logits.

### 3.3. Pre-training

**Contrastive Loss.** We pre-train Mantis by self-supervised contrastive learning. The goal is to learn an encoder that produces similar representations for two random augmentations of the same time series (a positive pair), while producing dissimilar representations for augmentations of different time series (negative pairs). Formally, let $\mathcal{T}$ be a space of transformations (augmentations) such that for all $\phi \in \mathcal{T}$ and $\mathbf{x} \in \mathcal{X}$ we have $\phi(\mathbf{x}) \in \mathcal{X}$. To measure the similarity between two embeddings, we first project the output of the foundation model $F(\mathbf{x})$ to a new dimension using a projector $g : \mathbb{R}^q \to \mathbb{R}^{q'}$ and then compute the cosine similarity between the two vectors as

$$s_{\cos}(\mathbf{a}, \mathbf{b}) := \frac{\mathbf{a}^\top \mathbf{b}}{\|\mathbf{a}\| \cdot \|\mathbf{b}\|}, \qquad \forall (\mathbf{a}, \mathbf{b}) \in \mathbb{R}^{2q'}.$$

Given a batch $B = \{\mathbf{x}_i\}_{i=1}^b$, we uniformly sample two augmentation functions $\phi$ and $\psi$ from $\mathcal{T}$, and then, for each example $\mathbf{x}_i$, compute the similarities scores as follows:

$$\mathbf{s}_i(\phi, \psi) = [s_{\cos}(g \circ F \circ \phi(\mathbf{x}_i), g \circ F \circ \psi(\mathbf{x}_j))]_{j=1}^b \in \mathbb{R}^b.$$

Following He et al. (2020) and denoting the cross-entropy loss by $l_{\text{ce}} : \mathbb{R}^b \times \{1, \ldots, b\} \to \mathbb{R}$, we update the parameters of $F$ and $g$ by minimizing the contrastive objective:

$$\sum_{i=1}^b l_{\text{ce}} \left( \frac{\mathbf{s}_i(\phi, \psi)}{T}, \; i \right),$$

where $T \in (0, +\infty)$ is a temperature that we fixed to 0.1.

**Augmentation.** We empirically evaluated several time-series augmentation strategies and observed that their effectiveness is highly dataset-dependent, as they may aggressively distort the signal and remove discriminative information. For pre-training, we have chosen the Random Crop Resize (RCR) augmentation (Figure 3). This transformation randomly crops a contiguous segment covering $(1 - c)\%$ of the original time series and then resizes it back to the original sequence length. We apply moderate distortions by sampling the crop rate $c$ uniformly between $0\%$ and $20\%$, thereby preserving the overall temporal structure of the signal.

A key advantage of contrastive learning with RCR is that the encoder is encouraged to be invariant to small temporal stretches and compressions, which in turn enables flexible resizing of the input without degrading performance. It is important to note

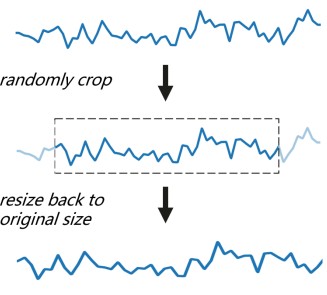

*Figure 3.* Random Crop Resize.

that RCR perturbs the original unit measurements, making it less suitable for forecasting tasks. However, since our focus is on time series classification, the primary goal is to capture discriminative temporal patterns rather than preserve absolute scales. Empirically, we find RCR to be the most effective augmentation for this purpose.

**Synthetic data.** To avoid any potential data leakage and improve sample efficiency, we pre-train Mantis exclusively on large-scale synthetic time series generated by CauKer (Xie et al., 2026). In contrast to earlier pre-training corpora (Lin et al., 2024) that may overlap with popular evaluation suites (e.g., UCR/UEA), synthetic data are out-of-distribution by construction, yielding a cleaner and more reliable zero-shot evaluation protocol. Empirically, we observe that synthetic pre-training is highly sample-efficient: even with only $100K$ synthetic sequences, Mantis reaches strong zero-shot feature extraction accuracy on UCR (see Section B.1 for more details). In addition, synthetic data are particularly useful for our contrastive objective that promotes *uniformity*, evenly distributing representations (Wang and Isola, 2020). Achieving such uniformity requires high data diversity, which CauKer can provide in a scalable and controllable manner.

### 3.4. Enhanced Test-Time Feature Extraction

Inspired by recent advances in test-time scaling of LLMs, we identified several strategies for enhanced feature extraction that we detail below.

**Layer-wise Representations.** Prior work has shown that intermediate layers of large foundation models often en-

code rich and transferable representations, sometimes outperforming final layers on downstream tasks (Skean et al., 2025). (Roschmann et al., 2025) have discovered that intermediate layers of vision models can be repurposed for time series classification. These findings suggest that embeddings produced by the final layer may be suboptimal, particularly in the frozen-encoder setting. Motivated by this line of work, we extend layer-wise representation analysis to Mantis. Rather than assuming that the final Transformer layer yields the most informative embedding, we explicitly consider intermediate-layer outputs as potential feature representations. In Section B.3, we show that intermediate representations are consistently more discriminative than the final ones, even for compact models such as Mantis.

Furthermore, we find that the benefits of increased pre-training data scale are primarily reflected in intermediate representations rather than in the final layer (see Appendix B.3 for more details). While final-layer performance tends to saturate (or even degrade) with larger pre-training datasets, the best-performing intermediate layer improves monotonically as the number of pre-training samples increases. This observation indicates that intermediate layers are crucial for unlocking the scaling benefits of pre-training in the zero-shot regime.

Finally, we observe that this behavior is not specific to Mantis but generalizes across a range of time series foundation models, including both classification- and forecasting-oriented architectures. Taken together, these insights motivate our test-time strategy of selecting and leveraging intermediate-layer representations for feature extraction, which we later show to yield consistent performance gains.

**Output-token aggregation.** Standard discriminative Transformers use the hidden state of the `[CLS]` token as the final representation and discard all other tokens (Devlin et al., 2019; Dosovitskiy et al., 2021). While this is often efficient when the `[CLS]` token has sufficiently aggregated global information through self-attention, it can be suboptimal when intermediate-layer representations are exploited. Therefore, we revisit token aggregation for Mantis and consider a strategy that explicitly incorporates non-classification tokens. More specifically, we use a concatenation of `[CLS]` token with the mean of non-classification tokens. As we show in Section B.4, non-classification tokens encode complementary information that is not fully captured by the classification token. Thus, the proposed combined aggregation strategy yields an improved zero-shot performance, increasing the embedding dimension from 256 to 512.

**Self-Ensembling (SE).** We introduce a simple test-time self-ensembling strategy based on input perturbations. The key idea is to exploit the fact that interpolating a time series to different sequence lengths changes the effective receptive field of each token, resulting in complementary representa-

tions even for the same input. As the number of tokens is fixed, varying the interpolation length modifies the degree of overlap between tokens: shorter sequences induce more overlap, while longer sequences approach non-overlapping patching. We therefore generate multiple interpolated versions of the same input, encode each independently, and concatenate the resulting embeddings. In addition, inspired by Auer et al. (2025b), we extend this strategy to first-order differences of the input signal and combine both representations. We refer to this strategy as *Self-Ensembling (SE)*, denoting the resulting model as *SE-Mantis*.

**Cross-model Embedding Fusion.** Since foundation models can be pre-trained with different objectives and architectures, their representations may capture complementary aspects of time series data. Motivated by this and by findings of Roschmann et al. (2025), we investigate whether zero-shot performance can be further improved by combining Mantis embeddings with those of competing models pretrained on other modalities and dedicated to the forecasting task. In Section 4.4, we experimentally show that Mantis benefits the most when combined with vision models.

## 4. Experimental Results

In this section, we present several sets of results. First, we perform a large-scale study that shows the best performance we obtained when using pre-trained Mantis with the enhanced feature extraction techniques described above. Then, we carefully break down the obtained performance by highlighting the performance of the studied models on collections of datasets commonly used in the literature. Finally, we ablate our design choices and test-time improvements.

### 4.1. Broad evaluation on UCR-91

Our first experimental evaluation is to broadly compare a wide variety of the different families of time series approaches on the common UCR-91 benchmark. We choose this particular benchmark as it allows us to compare Mantis and its variations with a maximum possible number of baselines by using the results reported in (Goswami et al., 2024). We include the following models in our study:

- **Self-supervised baselines**: TS2Vec (Yue et al., 2022), T-Loss (Franceschi et al., 2019), TNC (Tonekaboni et al., 2021), which learn representation for each dataset independently;

- Catch22 (Lubba et al., 2019) is a set of powerful statistical features designed for time series classification. In this paper, we have significantly improved this baseline by incorporating also patched statistics. We name this modification as **Catch22+** and refer to Appendix A.3 for more details and the corresponding ablation study.

- **TabPFN** (Hollmann et al., 2022) and **TabICL** (Qu

et al., 2025) are tabular foundation models. In this case, we treat each timestamp as a feature and ignore the sequential nature of the data.

- **MOMENT** (Goswami et al., 2024) is a T5-based auto-encoder (Raffel et al., 2020) pre-trained in a self-supervised way on 1.13 billion samples.

- **TiRex** (Auer et al., 2025b) a time series forecasting foundation model based on the xLSTM architecture (Beck et al., 2024).

- **Chronos2** (Ansari et al., 2025) is a transformer-based time series forecasting foundation model.

- **TiViT-H** is the approach proposed by (Roschmann et al., 2025) to leverage a pre-trained vision model for time series classification. **TiConvNext** leverages the pre-trained CLIP ConvNext in the same spirit.

- **NuTime** (Lin et al., 2024) is a classification TSFM proposed before Mantis. Their pre-training dataset is similar to the one used for pre-training of the first version of Mantis (1.89 million time series examples).

- **Mantis**, **SE-Mantis** and their cross-model fused variations.

We perform zero-shot feature extraction evaluation for all FMs, i.e., keeping the encoder frozen and using the produced embeddings together with the training labels to learn a classifier. While prior work (Goswami et al., 2024) used a linear classifier, we follow Roschmann et al. (2025) and couple the Standard Scaler and Logistic Regression from the scikit-learn package (Pedregosa et al., 2011). We set the maximum number of iterations to 500 and leave the other hyperparameters to the default ones, including the solver, which is L-BFGS-B (Zhu et al., 1997; Morales and Nocedal, 2011). For all foundation models, except TabPFN and TabICL, we perform an intermediate layer analysis to determine a layer with the highest zero-shot performance. This allows us to significantly improve the performance for most of these models, not to mention a drastic model size reduction. We give more experimental details in Section A.

For the self-supervised baselines, we extract the performance results from Goswami et al. (2024) and run the other baselines ourselves. The experimental results are presented in Figure 1, while in Appendix D we display the full results with 9 additional baselines (Figure 19 and Table 17). From the results, we note that several Mantis and its variations (SE-Mantis, Mantis & TiViT-H, Mantis & TiConvNext) outperform all other FMs and beat a very strong baseline TS2Vec, which performs self-supervised learning *for each dataset independently*. To the best of our knowledge, our paper is the first one to report this level of performance in a zero-shot feature extraction regime.

### 4.2. Performance breakdown

We now turn our attention to a more detailed evaluation that showcases the different domains and setups where Mantis excels. To this end, we consider the following four collections of datasets:

- Full **UCR** (Dau et al., 2019) benchmark that consists of all 128 univariate datasets (contrary to 91 datasets that were previously used above, following prior work).

- Multivariate **UEA** (Bagnall et al., 2018) benchmark that has 30 datasets. We exclude 3 datasets due to small test size or small sequence length, and subsample one dataset to ease computations (see more details in Appendix A). We further tag the collection by UEA-27.

- We additionally gathered a collection of datasets for Human Activity Recognition (**HAR**) that is one of the most widespread applications of time series classification models (Chen et al., 2025; Li et al., 2025b). We consider 7 datasets, where data come from either an inertial measurement unit (Ego4D, HHAR, UCI-HAR) or motion capture (EMOPain, MP8, MP50). For HHAR dataset, we follow (Gagnon-Audet et al., 2022) and consider two splits: in-distribution (ID) and out-of-distribution (OOD). Please refer to Appendix A.1.2 for more details.

- In a similar fashion, we test Mantis also on electroencephalogram (**EEG**) data that represent recordings of the brain's electrical activity. We consider various tasks, including sleep stage prediction (CAP, SEDFx), brain–computer interface (BCI) control tasks (Finger-Movements, PCL, SelfRegulationSCP), seizure detection (Epilepsy-EEG), blink-type classification (Blink). We describe the datasets in Appendix A.1.3.

To ease the computational burden, in what follows, we restrict our attention to the comparison of Mantis only with FMs, keeping one non-deep-learning baseline, Catch22+, for reference. Furthermore, due to the extended number of runs performed in this evaluation, we replace Logistic regression with a faster and computationally cheaper Random Forest with 200 trees and maximal tree depth (Breiman, 2001). We note beforehand that this choice doesn't alter the relative ranking of the studied methods.

**Results on UCR.** Table 1 displays the average performance of the considered models over 128 univariate datasets, while the complete table with results is deferred to Appendix D (Table 18 and 19). We can see that Mantis significantly outperforms the other models. Modern forecasting models (TiRex, Chronos2) and adapted vision models (TiViT-H, TiConvNext) are very competitive, establishing strong baselines. Interestingly, Catch22+, which is a set of statistical features, also establishes a good baseline, even outperforming foundation models such as NuTime and MOMENT. Tab-

*Table 1.* Classification accuracy on UCR and UEA-27 datasets of models grouped in different families.

| Dataset | Stats | Tabular FMs | | TS Forecasting FMs | | | Vision FMs | | TS Classification FMs | |
|---|---|---|---|---|---|---|---|---|---|---|
| | Catch22+ | TabPFN | TabICL | MOMENT | TiReX | Chronos2 | TiVIT-H | TiConvNext | NuTime | Mantis |
| **UCR** | 0.7969 | 0.7806 | 0.7707 | 0.7789 | 0.8013 | 0.8002 | 0.7943 | 0.8029 | 0.7732 | **0.8195** |
| **UEA-27** | 0.6963 | 0.6721 | 0.6850 | 0.6991 | 0.6952 | 0.6967 | 0.7091 | 0.7105 | 0.7199 | **0.7420** |

*Table 2.* Classification accuracy on HAR and EEG dataset collections. Boldface highlights the best method per row; NaN indicates the model did not pass the scale.

| Collection | Dataset | Catch22+ | TabPFN | TabICL | MOMENT | TiRex | Chronos2 | TiViT-H | TiConvNext | NuTime | Mantis |
|---|---|---|---|---|---|---|---|---|---|---|---|
| HAR | Ego4D | $0.4397_{\pm 0.002}$ | NaN | NaN | $0.4068_{\pm 0.001}$ | $0.5037_{\pm 0.001}$ | $0.4742_{\pm 0.0}$ | $0.1912$ | $0.1907$ | $0.5108_{\pm 0.0}$ | $\mathbf{0.5258}_{\pm 0.001}$ |
| | EMOPain | $0.8826_{\pm 0.006}$ | $0.7831$ | $0.7831$ | $0.8469_{\pm 0.002}$ | $0.7915_{\pm 0.003}$ | $0.8075_{\pm 0.004}$ | $0.83_{\pm 0.002}$ | $0.8225_{\pm 0.003}$ | $\mathbf{0.8901}_{\pm 0.005}$ | $0.8798_{\pm 0.009}$ |
| | HHAR-ID | $0.9738_{\pm 0.001}$ | $0.8938$ | $0.9073$ | $0.9299_{\pm 0.0}$ | $0.9481_{\pm 0.0}$ | $0.9475_{\pm 0.002}$ | $0.9338_{\pm 0.001}$ | $0.9461_{\pm 0.001}$ | $0.9808_{\pm 0.001}$ | $\mathbf{0.9845}_{\pm 0.001}$ |
| | HHAR-OOD | $0.4808_{\pm 0.008}$ | $0.5311$ | $0.5441$ | $0.3081_{\pm 0.001}$ | $0.5135_{\pm 0.014}$ | $0.4174_{\pm 0.008}$ | $0.3748_{\pm 0.007}$ | $0.4031_{\pm 0.01}$ | $0.56_{\pm 0.005}$ | $\mathbf{0.5822}_{\pm 0.005}$ |
| | MP8 | $0.572_{\pm 0.008}$ | $0.6235$ | $0.6185$ | $0.6392_{\pm 0.011}$ | $0.5994_{\pm 0.016}$ | $0.5966_{\pm 0.006}$ | $0.6022_{\pm 0.009}$ | $0.5966_{\pm 0.003}$ | $0.6504_{\pm 0.008}$ | $\mathbf{0.6857}_{\pm 0.012}$ |
| | MP50 | $0.3602_{\pm 0.016}$ | $0.5664$ | $0.5042$ | $\mathbf{0.7434}_{\pm 0.017}$ | $0.6644_{\pm 0.01}$ | $0.6639_{\pm 0.007}$ | $0.6908_{\pm 0.008}$ | $0.6801_{\pm 0.008}$ | $0.6913_{\pm 0.012}$ | $0.7345_{\pm 0.007}$ |
| | UCI-HAR | $0.8273_{\pm 0.003}$ | $0.809$ | $0.8157$ | $0.8782_{\pm 0.001}$ | $0.8744_{\pm 0.002}$ | $0.8873_{\pm 0.002}$ | $0.8936_{\pm 0.001}$ | $0.8918_{\pm 0.001}$ | $0.8842_{\pm 0.001}$ | $\mathbf{0.9013}_{\pm 0.002}$ |
| | Avg | $0.6481$ | NaN | NaN | $0.6789$ | $0.6993$ | $0.6849$ | $0.6452$ | $0.6473$ | $0.7382$ | $\mathbf{0.7562}$ |
| | Avg UCR HAR | $0.7564$ | $0.7479$ | $0.7285$ | $0.7572$ | $0.7687$ | $0.7702$ | $0.7726$ | $0.772$ | $0.756$ | $\mathbf{0.8007}$ |
| | Avg UEA HAR | $0.8138$ | $0.7591$ | $0.764$ | $0.8431$ | $0.846$ | $0.8492$ | $0.8535$ | $0.8488$ | $0.8539$ | $\mathbf{0.8739}$ |
| EEG | Blink | $0.9963_{\pm 0.001}$ | $0.9178$ | $0.8978$ | $0.9674_{\pm 0.003}$ | $\mathbf{0.997}_{\pm 0.001}$ | $0.9733_{\pm 0.01}$ | $0.9852_{\pm 0.006}$ | $0.98_{\pm 0.002}$ | $0.66_{\pm 0.006}$ | $0.9956_{\pm 0.002}$ |
| | CAP-ID | $0.751_{\pm 0.001}$ | NaN | NaN | $0.7374_{\pm 0.002}$ | $0.8104_{\pm 0.002}$ | $\mathbf{0.8179}_{\pm 0.001}$ | $0.7983_{\pm 0.001}$ | $0.8126_{\pm 0.001}$ | $0.8044_{\pm 0.0}$ | $0.8152_{\pm 0.001}$ |
| | CAP-OOD | $0.71_{\pm 0.003}$ | NaN | NaN | $0.7408_{\pm 0.001}$ | $0.7821_{\pm 0.001}$ | $0.7857_{\pm 0.001}$ | $0.7781_{\pm 0.001}$ | $0.784_{\pm 0.001}$ | $0.767_{\pm 0.002}$ | $\mathbf{0.7859}_{\pm 0.0}$ |
| | Epilepsy-EEG | $0.9507_{\pm 0.002}$ | $0.9496$ | $0.9447$ | $0.952_{\pm 0.001}$ | $\mathbf{0.9614}_{\pm 0.001}$ | $0.95_{\pm 0.001}$ | $0.9548_{\pm 0.0}$ | $0.9495_{\pm 0.001}$ | $0.9234_{\pm 0.002}$ | $0.9558_{\pm 0.001}$ |
| | FingerMovements | $0.4967_{\pm 0.064}$ | $0.5$ | $0.49$ | $0.53_{\pm 0.02}$ | $0.52_{\pm 0.04}$ | $0.5233_{\pm 0.021}$ | $0.5367_{\pm 0.045}$ | $0.5333_{\pm 0.059}$ | $0.5267_{\pm 0.015}$ | $\mathbf{0.55}_{\pm 0.01}$ |
| | PCL-ID | $0.5241_{\pm 0.003}$ | NaN | NaN | $0.5431_{\pm 0.012}$ | $0.5272_{\pm 0.011}$ | $0.5308_{\pm 0.002}$ | $0.5347_{\pm 0.007}$ | $0.5385_{\pm 0.005}$ | $0.5615_{\pm 0.003}$ | $\mathbf{0.5716}_{\pm 0.003}$ |
| | PCL-OOD | $0.5025_{\pm 0.004}$ | NaN | NaN | $0.5129_{\pm 0.001}$ | $0.4988_{\pm 0.001}$ | $0.5072_{\pm 0.004}$ | $0.5003_{\pm 0.005}$ | $0.5132_{\pm 0.001}$ | $\mathbf{0.5415}_{\pm 0.004}$ | $0.5311_{\pm 0.009}$ |
| | SEDFx-ID | $0.7574_{\pm 0.0}$ | NaN | NaN | $0.7516_{\pm 0.001}$ | $0.7904_{\pm 0.0}$ | $\mathbf{0.8}_{\pm 0.0}$ | $0.7884_{\pm 0.001}$ | $0.8008_{\pm 0.001}$ | $0.7822_{\pm 0.001}$ | $\mathbf{0.8}_{\pm 0.0}$ |
| | SEDFx-OOD | $0.7142_{\pm 0.001}$ | NaN | NaN | $0.7244_{\pm 0.001}$ | $0.7714_{\pm 0.0}$ | $\mathbf{0.7758}_{\pm 0.0}$ | $0.7709_{\pm 0.0}$ | $0.7755_{\pm 0.001}$ | $0.741_{\pm 0.0}$ | $0.7636_{\pm 0.0}$ |
| | SelfRegulationSCP1 | $0.7702_{\pm 0.007}$ | $\mathbf{0.8942}$ | $0.8874$ | $0.7747_{\pm 0.006}$ | $0.7884_{\pm 0.012}$ | $0.785_{\pm 0.003}$ | $0.7986_{\pm 0.007}$ | $0.7929_{\pm 0.01}$ | $0.7952_{\pm 0.003}$ | $0.8134_{\pm 0.009}$ |
| | SelfRegulationSCP2 | $0.4926_{\pm 0.031}$ | $0.4778$ | $0.5056$ | $0.4907_{\pm 0.023}$ | $0.4963_{\pm 0.049}$ | $\mathbf{0.5167}_{\pm 0.006}$ | $0.4907_{\pm 0.033}$ | $0.5056_{\pm 0.011}$ | $0.5074_{\pm 0.018}$ | $\mathbf{0.5167}_{\pm 0.006}$ |
| | Avg | $0.6969$ | NaN | NaN | $0.7023$ | $0.7221$ | $0.7242$ | $0.7215$ | $0.726$ | $0.6919$ | $\mathbf{0.7363}$ |

ular foundation models are competitive as well: although their average performance is not very high, their win rate is similar to Mantis (see Appendix D). This also indicates that a considerable portion of datasets in UCR are nearly tabular, so their sequential nature can simply be ignored. We compare Mantis and TabPFN more thoroughly in Section C.3.

**Results on UEA.** The average performance of the considered models in average over 27 multivariate datasets is illustrated in Table 1 while the complete results are deferred to Appendix D (Table 20). One can see that Mantis significantly outperforms the other models by more than 2%. Interestingly, the model ranking is quite different in this benchmark. For example, NuTime, being one of the worst on UCR, is the third-best model this time. Since its hidden dimension is the smallest among the foundation models (128, to be exact), we hypothesize that NuTime may be less sensitive to the curse of dimensionality as the total number of deep features grows linearly with the number of channels.

**Results on HAR.** Table 2 displays the experimental results for human activity recognition tasks. We also include the average performance over 19 UCR and 9 UEA datasets that are related to HAR. We can see that the gap between Mantis and the other models is quite noticeable, indicating high relevance of the proposed models to this application domain. Similar to UEA, NuTime performs quite well, while other models show less robust results.

**Results on EEG.** The experimental results for EEG data can be found in Table 2. We note that this benchmark includes several big datasets (CAP, PCL, SEDFx) on which tabular foundation models do not pass the memory and/or time constraints (we set 10 hours as the time deadline for the runtime of one method over one dataset). In contrast to the HAR results, the performance gap between Mantis and other models is smaller, and the model ranking is close to that seen on UCR. As EEG classification is notoriously known to be a difficult task, it will be interesting to see if time series foundation models can have an impact on the progress in this field.

### 4.3. Complexity Analysis

In this study, we compare all the models in terms of memory consumption and running time. As a rule of thumb, the memory consumption of a deep learning model correlates with the total number of its parameters. Figure 4 shows the trade-off between the UCR accuracy and the model size after choosing the layer with the best performance (see Section B.3 for more details). The exact number of parameters before and after model pruning can be found in the Appendix (Table 6). For TabPFN and TabICL, it is not possible to perform layer pruning due to the architectural design. From the results, we can see that the four biggest models are TiConvNext, TiViT-H, MOMENT, and Chronos2 with more than 100 million parameters. After layer pruning, their sizes

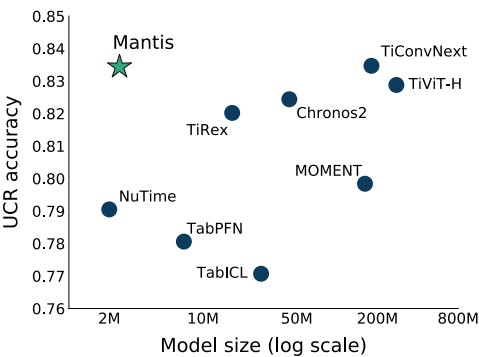

*Figure 4.* Mantis is the best performing and most lightweight model among the existing FMs.

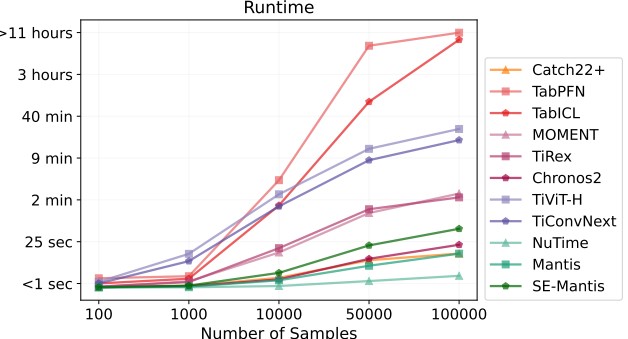

*Figure 5.* The inference time of the considered models as a function of the number of samples. The time is in log scale.

are significantly reduced, though still being above 100 million parameters for TiViT-H, TiConvNext, and MOMENT. Layer pruning also helps to reduce the size of Mantis to 2.2 million parameters, making it as small as NuTime.

To measure running time, we compare how much it takes for a forward pass over an entire dataset with a batch size equal to 256. We generate univariate synthetic data with length 100, varying the total number of samples within $n \in \{10^2, 10^3, 10^4, 5 \cdot 10^4, 10^5\}$. As TabPFN and TabICL require both train and test as an input, for them, we use $n/2$ samples as train data and $n/2$ for test, while using a batch strategy (256) for test data. The experimental results are displayed in Figure 5 and reveal that tabular foundation models are the slowest for large sample sizes, taking more than 10 hours for 100,000 examples. They are followed by vision-based models, TiViT-H and TiConvNext, that are slow but, nevertheless, feasible. Next cluster is represented by MOMENT and TiRex[1]. The other models, including Chronos2, Mantis, and NuTime are significantly faster. Interestingly, Mantis performs as fast as Catch22+, which simply calculates pre-determined statistics. Together with Figure 4, this result demonstrates that Mantis can be deployed in compute-limited environments.

### 4.4. Ablation Study

We perform a comprehensive ablation study to understand which design choices drive the performance of Mantis.

**Pre-training data.** We compare pre-training Mantis on $100K$ synthetic CauKer sequences against several real-world time series corpora. We note that pre-training on synthetic data exceeds the performance of pre-training on equally sized real-world datasets and approaches the accuracy of pre-training on the full $1.89M$ NuTime corpus (see Table 7 in Section B.1).

---

[1]Note that we did not manage to compile TiRex as they suggest, otherwise their model should be faster.

**Architecture.** We ablate the Token Generator Unit and the transformer backbone. We show that every branch of the tokenizer (encoding raw signal, its first-order difference, and patch-wise unnormalized statistics) contributes to the performance (Section B.2, Figure 8). In addition, we ablate the mean pooling and the convolution operators. Second, we vary the convolution kernel size and observe monotonically increasing performance up to a kernel of 41 (Figure 9), beyond which gains saturate. Third, we compare a classical Transformer architecture with sinusoidal positional encoding, LayerNorm, and GELU to a modern variant with RoPE, RMSNorm, and SwiGLU. The modern design yields a small yet consistent improvement (Figure 10 and Table 8).

**Layer-wise representations.** We analyze the pre-training phase by illustrating the performance layer-by-layer, epoch-by-epoch. We find that with a sufficient number of parameter updates, the final layer starts to overfit the contrastive objective, so intermediate layers perform better (Section B.3, Figure 11). As the number of pre-training samples increases, we see that the final layer stagnates, but the *best performance across intermediate layers* monotonically increases. This finding suggests that **intermediate representations unlock the scaling benefits of pre-training**. Overall, we find the third layer of Mantis to be the most powerful one. In addition, we compute the layer-by-layer performance of NuTime, MOMENT, TiRex, and Chronos2 to identify their most powerful intermediate layers, which we used in the final comparisons (Figure 12).

**Output token aggregation.** We compare different ways to aggregate the output tokens. Concatenating the classification token with the mean of the remaining tokens yields consistent accuracy gain compared with using only the classification token (Section B.4, Fig. 14). Note that this concatenation strategy may bring help only for intermediate layers, whereas at the last layer the `[CLS]` token aggregates the information from the remaining tokens sufficiently well (Section B.4, Fig. 15).

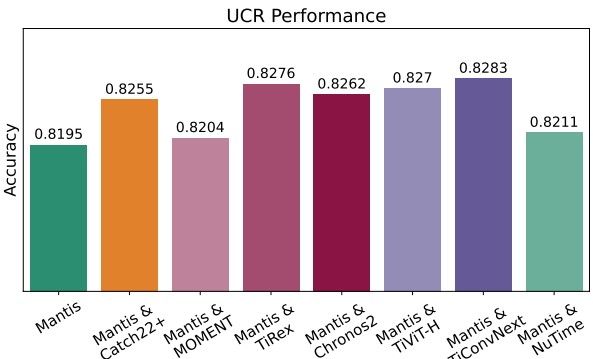

*Figure 6.* Combination of Mantis with other models.

**Self-ensembling.** We experimentally show that self-ensembling improves the performance at inference time (Section B.5, Fig. 17). We demonstrate that interpolating both the input and its first-order difference to multiple lengths allows us to get richer embeddings. Note that SE-Mantis outperforms Mantis on the full UCR collection, but not on the subset of 91 datasets as it can be seen in Figure 1. All in all, we view SE as an optional technique that can yield significant performance boost on some datasets (e.g., see Car, Phoneme and Worms datasets in Table 21 and Table 22).

**Cross-model embedding fusion.** The experimental results for combining the embeddings of different pretrained models are reported in Figure 6. Similar to Roschmann et al. (2025), we observe that model combination is consistently beneficial, even when Mantis is combined with hand-crafted statistical features (Catch22+). The smallest gains are obtained when combining Mantis with MOMENT and NuTime, which may be explained by limited representational complementarity in the case of NuTime, or by lower standalone performance. The highest overall performance is achieved when Mantis is combined with TiConvNeXt.

**Choice of a predictor.** While simple linear probing is conventional in the deep learning community, it implicitly assumes that feature variances (across the dimensions of the embeddings) share the same order of magnitude. For Mantis, this is not the case as self-supervised pre-training does not guarantee similarly distributed features, thus standard scaling or batch normalization is generally recommended for probing (Marks et al., 2025).

*Table 3.* Performance of Mantis on UCR with different predictors.

| Log. Regression (L-BFGS-B) | | | PyTorch Linear (AdamW) | | | RF |
|---|---|---|---|---|---|---|
| W/o Norm | MinMax | Standard | W/o Norm | LayerNorm | BatchNorm | |
| 0.763 | 0.829 | **0.837** | 0.669 | 0.71 | 0.827 | 0.82 |

In Table 3, we compare several probing approaches: (a) scikit-learn logistic regression (optimized by L-BFGS-B)

without normalization, with a Min-Max Scaler, and with a Standard Scaler, (b) PyTorch (Paszke et al., 2019) logistic regression (optimized by AdamW optimizer) without normalization, with a Layer Norm, and with a BatchNorm, (c) Random Forest (RF). We can see that linear probing's performance degrades drastically without normalization, and layer pre-normalization does not fully resolve the issue. Conversely, scaling features over the entire dataset or batch yields superior results.

# 5. Conclusion and Future Work

In this paper, we presented Mantis, a time series classification foundation model pre-trained exclusively on synthetic data via contrastive learning. Our ablation studies highlight the importance of careful time series tokenization and demonstrate the effectiveness of the proposed Token Generator Unit. In addition, we introduced an enhanced test-time methodology that yields consistent performance gains. In particular, we showed that the zero-shot capabilities of a pre-trained model can be significantly improved by leveraging intermediate-layer representations, refined output-token aggregation, and self-ensembling. These findings suggest that scaling laws in time series classification are attainable, and that larger models can be trained whose full potential is unlocked at test time.

Beyond scaling, several promising research directions remain open, including multi-modal architectures, zero-shot classification via in-context learning, and joint foundation models for classification and forecasting. Finally, we believe that improving the interpretability of model outputs and ensuring strong calibration properties are crucial directions for future work, as they may enable reliable unsupervised assessment of model performance.

# Impact Statement

This paper advances the field of machine learning by proposing a lightweight foundation model for time series classification. Our experiments are conducted on standard, publicly available benchmarks, and the pre-training data are synthetic. There are many potential societal consequences of our work, none of which we feel must be specifically highlighted here.

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

# A. Experimental Setup

In this section, we provide more details on the chosen datasets and models. We also experimentally justify Catch22+, a strong baseline that we have proposed.

## A.1. Datasets

We consider four benchmarks. For UCR (Dau et al., 2019), we follow the standard protocol and detail the three other benchmarks below.

### A.1.1. UEA BENCHMARK

From the original set (Bagnall et al., 2018), we have excluded AtrialFibrillation and StandWalkJump datasets due to their very small test size and PenDigits due to its very small sequence length. For InsectWingbeat dataset, we subsampled 1000 examples from the original training set (which contains 30,000 examples) and 1000 from the original test set (of 20,000 examples) to reduce computational overhead while maintaining sufficient variety in the data for robust model evaluation.

### A.1.2. HAR DATASETS

Below we give more details on the human activity recognition datasets used in our experiments, which characteristics can be found in Table 4.

*Table 4.* Characteristics of HAR datasets.

| Dataset | Num. of Channels | Seq. Length | Train Size | Test Size | Num. of classes |
|---|---|---|---|---|---|
| Ego4D | 6 | 1000 | 247095 | 66994 | 31 |
| EMOPain | 30 | 200 | 968 | 355 | 3 |
| HHAR-ID | 6 | 500 | 8716 | 3419 | 6 |
| HHAR-OOD | 6 | 500 | 11150 | 2154 | 6 |
| MP8 | 8 | 161 | 1426 | 595 | 4 |
| MP50 | 50 | 161 | 1426 | 595 | 4 |
| UCI-HAR | 3 | 206 | 5881 | 2947 | 6 |

- *Ego4D* (Grauman et al., 2022) is a multi-modal dataset for ego-centric human activity recognition. The use of the dataset requires signing the license agreement. We take time series data coming from Inertial Measurement Units (IMU) and pre-process them using the script provided by Chen et al. (2025).

- *EMOPain* (Egede et al., 2020) is a movement-based chronic pain detection dataset. We downloaded it using the script provided by Gao et al. (2024).

- *HHAR* (Stisen et al., 2015): we take from the WOODS benchmark (Gagnon-Audet et al., 2022). For *ID* setting, we merge all domains and make a train-validation-test split in proportion 63.75%-11.25%-25%. For *OOD*

setting, we use three domains for training, namely, "nexus4", "s3" and "s3mini", and "lgwatch" domain for test.

- Military Press (MP, Singh et al., 2023) is a dataset for human exercise performance classification. We use two versions of the dataset provided by Sungu Isiacik et al. (2025): *MP8* (8 key body point coordinates) and *MP50* (all 50 coordinates from 25 body parts).

- *UCI-HAR* (Anguita et al., 2013) is preprocessed following Zhang et al. (2022).

- In the UCR collection, the following datasets are related to HAR: AllGestureWiimoteX,AllGestureWiimoteY, AllGestureWiimoteZ, CricketX, CricketY, CricketZ, GestureMidAirD1, GestureMidAirD2, GestureMidAirD3, GesturePebbleZ1, GesturePebbleZ2, GunPoint, GunPointAgeSpan, GunPointMaleVersusFemale, GunPointOldVersusYoung, PickupGestureWiimoteZ, ShakeGestureWiimoteZ, UWaveGestureLibraryAll, UWaveGestureLibraryX, UWaveGestureLibraryY, UWaveGestureLibraryZ.

- In the UEA collection, these datasets are related to HAR: BasicMotions, Cricket, ERing, Epilepsy, Handwriting, Libras, NATOPS, RacketSports, UWaveGestureLibrary.

### A.1.3. EEG DATASETS

Below we provide more details on the EEG datasets used in our experiments, which main characteristics are given in Table 5.

*Table 5.* Characteristics of EEG datasets.

| Dataset | Num. of Channels | Seq. Length | Train Size | Test Size | Num. of classes |
|---|---|---|---|---|---|
| Blink | 4 | 510 | 500 | 450 | 2 |
| CAP-ID | 19 | 3000 | 25748 | 10098 | 6 |
| CAP-OOD | 19 | 3000 | 27393 | 8265 | 6 |
| Epilepsy-EEG | 1 | 178 | 60 | 11420 | 2 |
| FingerMovements | 28 | 50 | 316 | 100 | 2 |
| PCL-ID | 48 | 750 | 14405 | 5650 | 2 |
| PCL-OOD | 48 | 750 | 9880 | 7800 | 2 |
| SEDFx-ID | 4 | 3000 | 152178 | 59678 | 6 |
| SEDFx-OOD | 4 | 3000 | 133746 | 52838 | 6 |
| SelfRegulationSCP1 | 6 | 896 | 268 | 293 | 2 |
| SelfRegulationSCP2 | 7 | 1152 | 200 | 180 | 2 |

- *Blink* (Chicaiza and Benalcázar, 2021) is a dataset for classification of eye blink types. We downloaded it using the script provided by Gao et al. (2024).

- *Epilepsy-EEG* (Andrzejak et al., 2001) is preprocessed following Zhang et al. (2022).

- *FingerMovements*, *SelfRegulationSCP1* and *SelfRegulationSCP2* are taken from the UEA archive.

*Table 6.* Comparison of Model Sizes.

| # of Params. | TabPFN | TabICL | MOMENT | TiRex | Chronos2 | TiViT-H | TiConvNext | NuTime | Mantis |
|---|---|---|---|---|---|---|---|---|---|
| Original | 7.2M | 27.1M | 341.2M | 35.3M | 119.5M | 630.8M | 843.4M | 2.4M | 4.2M |
| After Pruning | - | - | 161.4M | 16.5M | 44M | 276.6M | 180.3M | 2M | 2.2M |

- 3 datasets are taken from the WOODS benchmark (Gagnon-Audet et al., 2022): motor imagery classification with *PCL* (Schalk et al., 2004; Cho et al., 2017; Lee et al., 2019), and sleep stage classification with *CAP* (Terzano et al., 2002) and *SEDFx* (Kemp et al., 2000). For *ID* setting, we merge all domains and make a train-validation-test split in proportion 63.75%-11.25%-25%. For *OOD* setting, we split by domains. For PCL, we use "Cho2017" for training, "PhysionetMI" for validation, "Lee2019MI" for test. For CAP, we use Machine 0,1,3 for training, Machine 2 for validation, Machine 4 for test. For SEDFx, we use Age 20-60 for training, Age 60-80 for validation, Age 80-100 for test.

### A.2. Models

In Table 6, one can found the number of parameters for each foundation model under consideration. If relevant, the table show the number of parameters before and after pruning using the layer-by-layer approach (more details on the optimal intermediate layer for each model can be found in Section B.3). Below, we give more implementation details on the baselines.

- *Catch22.* We use the official python implementation `pycatch22==0.4.5`.

- *NuTime.* We use the pre-trained weights provided by the authors in their GitHub repository, while fixing the hyperparameters of the architecture according to this configuration file. In contrast to the original implementation, we do not use their adapter (described in Section 3.4 of their paper) but process all channels independently as for Mantis and MOMENT. This allows us to use NuTime in the zero-shot feature extraction setting as their adapter has to be fine-tuned.

- *TabPFN.* We use the default version of `TabPFNClassifier` from `tabpfn==2.2.1`. As TabPFN does not support more than 10 classes, we use the `ManyClassClassifier` wrapper from TabPFN Extensions (Ye et al., 2025).

- *TabICL.* We use the official implementation with default parameters from `tabicl==0.1.3`.

- *MOMENT:* We use the MOMENT-large model (`d_model=1024`), which pre-trained weights can be found on the corresponding HuggingFace repository.

To handle the multi-channel setup, we process every channel independently and concatenate all the embeddings before passing them to the classification head. In the paper, they have considered datasets with a sequence length $\leq 512$ and use zero-padding to fix the input size to 512. At the same time, we have also tried to interpolate sequences to 512 instead, and it did not affect the performance of MOMENT. Thus, we have decided to stick to the latter option as it allows us to evaluate MOMENT for any sequence length.

- *TiRex.* We use the official implementation with default parameters from `tirex-ts==1.1.1`.

- *Chronos2.* We use the official implementation with default parameters from `chronos-forecasting==2.0.0`.

- *TiViT-H* and *TiConvNext*. We use the official implementation available at GitHub. For CLIP ViT-H, we use this checkpoint, while CLIP ConvNext's checkpoint can be found here (Wolf et al., 2020; Radford et al., 2021; Ilharco et al., 2021; Liu et al., 2022; Schuhmann et al., 2022).

### A.3. Catch22+

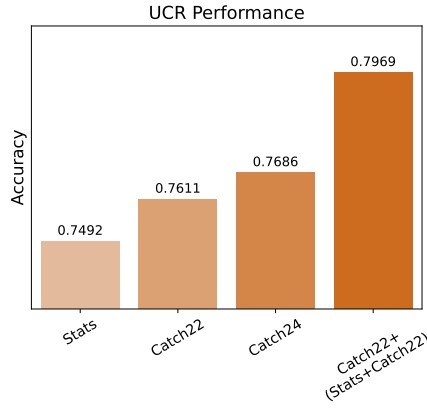

*Figure 7.* Catch22+ ablation study.

Catch22 (Lubba et al., 2019) is a manually selected set of time series statistics with a high discriminative power. Despite the authors later proposed an improvement by adding the global mean and standard deviation to the set (so-called Catch24), we have found that it can be further improved by splitting the input into non-overlapping patches (we follow Auer et al. (2025a) and set their number to 8) and computing

the mean and standard deviation for each of them. Using these means and standard deviations as features for classification we further refer by Stats. Figure 7 illustrates the average accuracy of different configurations across UCR datasets. One can notice that the combination of Catch22 with Stats yields a big improvement by 3.58%, while the improvement from adding global mean and standard deviation (Catch24) is more modest (0.75%). For this reason, we have considered Catch22+ (concatenation of Catch22 and Stats) as a baseline in the main part of our paper.

## B. Ablation Study

### B.1. Pre-training Data

To validate the benefit of synthetic data for pre-training, we compare 100,000 samples generated by CauKer with other pre-training datasets: (a) a set of 1.89 million real time series samples used to pre-train NuTime (Lin et al., 2024), which is not strictly out-of-distribution (OOD) as data included UCR and UEA training sets, (b) its 100,000 subset such that there are no UCR nor UEA samples in it, (c) Anomaly UCR (Wu and Keogh, 2021), a collection of datasets for anomaly detection. In Table 7 we display zero-shot feature extraction performance results. We can see that the obtained results confirm high efficiency of synthetic data for pre-training that outperform the real data of same size.

*Table 7.* Performance of Mantis on UCR collection with different pre-training datasets.

| Pre-training Data | Nature | Size | UCR In? | UCR acc. |
|---|---|---|---|---|
| NuTime (NT) | Real | 1.89M | Yes | $0.7921_{\pm 0.0012}$ |
| Subset of NT | Real | 100K | No | $0.7829_{\pm 0.0008}$ |
| Anomaly | Real | 38K | No | $0.7473_{\pm 0.0014}$ |
| CauKer | Synth | 100K | No | $0.7881_{\pm 0.001}$ |

This observation can be intuitively explained by the nature of the pre-training objective. Self-supervised contrastive learning explicitly promotes *uniformity* in the embedding space, encouraging representations to be evenly distributed (Wang and Isola, 2020). Achieving such uniformity requires high data diversity, which synthetic generation methods such as CauKer can provide in a scalable and controllable manner. As a result, synthetic data offer both strong sample efficiency and improved generalization performance.

### B.2. Architecture

First, we analyze the proposed Token Generator Unit, which combines three parallel branches: tokens extracted from the raw time series, tokens derived from its first-order differential, and the encoded patch-wise statistics (mean and standard deviation).

We compare the proposed tokenizer against the following

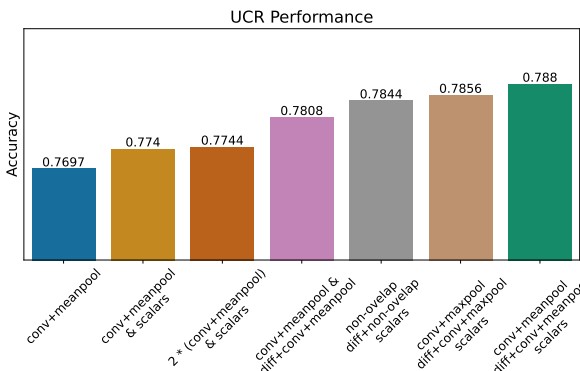

*Figure 8.* Ablation study for the proposed Token Generator Unit.

baselines:

- Only the first branch is used, consisting of a convolutional layer followed by mean pooling.

- The first and third branches are combined, i.e., patch-wise scalar encoding is added to the raw signal branch.

- The first branch is duplicated and combined with the scalar statistics branch. This baseline is introduced to determine whether the benefit of the second branch arises from incorporating the first-order differential or simply from increasing the number of parameters.

- The first and second branches are combined, removing the scalar encoder.

- All three branches are retained, but the convolution with mean pooling is replaced by embeddings of non-overlapping patches. This baseline is introduced to compare the proposed convolution-based patching against the non-overlapping patching used in NuTime (Lin et al., 2024).

- All three branches are retained, but mean pooling is replaced with max pooling to assess the impact of the pooling strategy.

Figure 8 presents the corresponding results. Each branch of the proposed Token Generator Unit contributes positively to performance (when comparing the 1st, 2nd, 4th, and 7th bars). Moreover, comparing the 3rd and 4th bars shows that incorporating features derived from the first-order differential yields a clear performance gain, confirming that the improvement is not merely due to increased model capacity. Regarding patching strategies, convolution with mean pooling consistently outperforms both max pooling and the non-overlapping patch embedder.

Next, we investigate the impact of the kernel size of the convolutional layers in Token Generator Unit. We vary kernel size within $\{9, 17, 25, 33, 41, 49\}$ and observe that performance improves monotonically up to a kernel size of 41, which provides the best trade-off between temporal

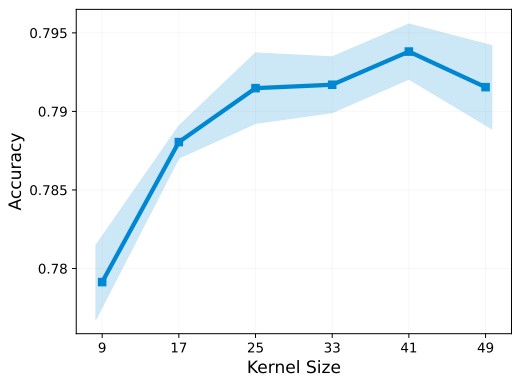

*Figure 9.* Ablation on convolution kernel size.

resolution and receptive field (Figure 9).

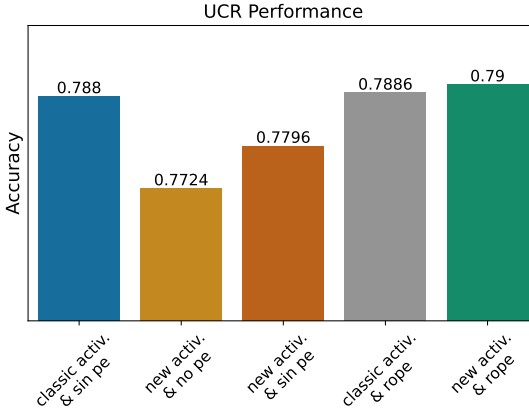

*Figure 10.* Comparison of different transformer configurations.

Finally, we explore two Transformer architectures: (a) the standard one with sinusoidal position encoding, layer norm, GELU in the feedforward layers (Vaswani et al., 2017; Dosovitskiy et al., 2021), and (b) a more modern version (Cohen et al., 2025) with Rotary Positional Encoding (RoPE; Su et al., 2024), RMS Layer Normalization (Zhang and Sennrich, 2019) in place of standard Layer Normalization, and the use of SwiGLU activations (Shazeer, 2020) in the feedforward layers. We empirically evaluate the following variants:

- Classical activations and normalization with sinusoidal positional encoding.

- SwiGLU and RMS normalization without positional encoding.

- SwiGLU and RMS normalization with sinusoidal positional encoding.

- Classical activations and normalization with RoPE.

- SwiGLU and RMS normalization with RoPE.

Figure 10 reports the corresponding results on the UCR benchmark, averaged over three seeds. The combination of SwiGLU, RMS Layer Normalization, and RoPE yields the best performance. Although the absolute improvement is modest, it is stable across runs. For additional validation, we also compared the two versions of Transformer on larger-scale pre-training with one million synthetic samples, performing a layer-by-layer analysis (as in this setup the last layer is not anymore the most powerful one). The results are illustrated in Table 8: again, a modern transformer yields slightly better results. Thus, we use the second version of Transformer for the final architecture of Mantis.

*Table 8.* Performance comparison between the first and the second version of transformer architecture keeping all the other parameters of Mantis fixed. Both architectures were pre-trained on 1 million synthetic examples and evaluated on UCR.

| Layer Number | V1 Transformer | V2 Transformer |
|:---:|:---:|:---:|
| 1 | 0.7872 | 0.7669 |
| 2 | 0.7979 | 0.7987 |
| 3 | 0.7989 | **0.7994** |
| 4 | 0.7927 | 0.7986 |
| 5 | 0.7927 | 0.7919 |
| 6 | 0.7892 | 0.78 |

### B.3. Layer-by-layer, Epoch-by-epoch

We extend the layer-by-layer analysis to Mantis (Skean et al., 2025) and other time series foundation models. In Figure 11, we track the UCR classification performance of all six Transformer layers of Mantis across pre-training epochs. We additionally vary the size of the synthetic pre-training dataset from 100,000 to 2,000,000 samples. Several notable patterns emerge. First, the relative improvement of intermediate layers appears to be closely tied to the total number of parameter updates. In our training setup, each epoch processes the full dataset, so increasing the dataset size directly increases the number of updates. With 100K samples, the final layer remains the strongest throughout training. In contrast, for larger datasets, intermediate layers steadily improve over time and eventually surpass the final layer. To test whether this behavior is indeed driven by the number of updates rather than dataset size per se, we pre-train Mantis on 100K samples for 1,000 epochs, matching the number of updates used for 1M samples over 100 epochs. The experimental results are displayed in Figure 13. One can see an interesting pattern where the intermediate layers start to dominate in performance when increasing the number of updates. This suggests that the final layer may overfit the contrastive objective, so intermediate layers start to generalize better.

However, increasing the number of epochs for the 100K dataset does not lead to further gains in overall performance.

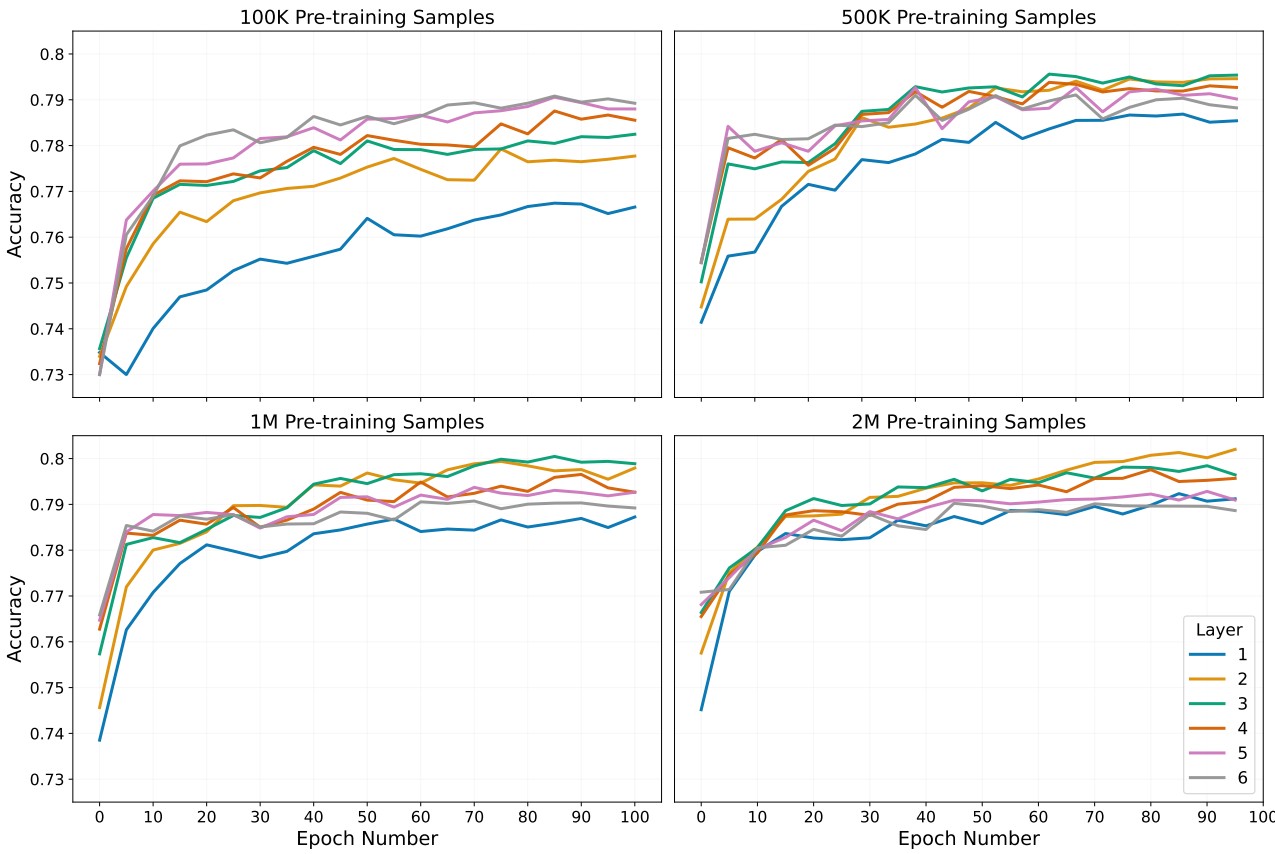

*Figure 11.* Intermediate layer results

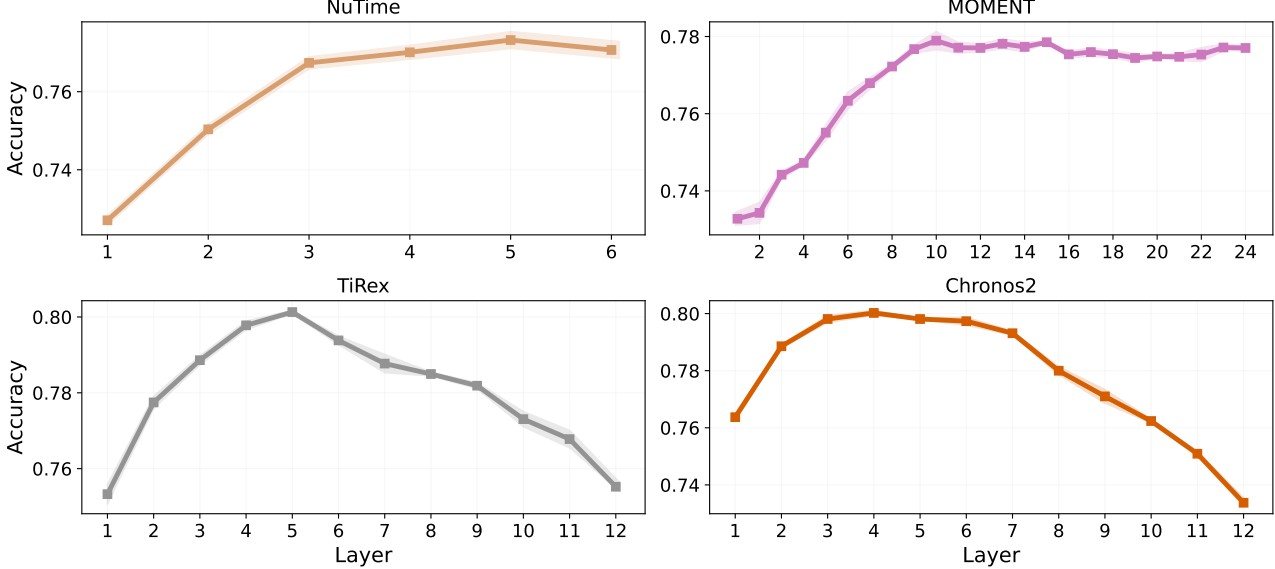

*Figure 12.* Layer-by-Layer for other models.

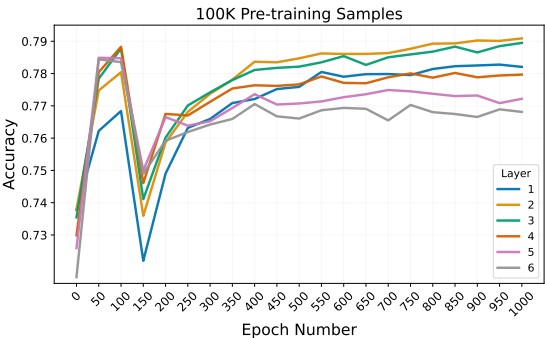

*Figure 13.* Layer by layer over 1000 epochs with 100K pre-training samples.

Instead, performance follows a double-descent–like behavior and converges to the same level achieved during the first 100 epochs. This leads to our second key observation from Figure 11. While the final-layer performance remains largely unchanged (or may even degrade) as the pre-training dataset grows, the *best-performing intermediate layer* improves consistently with increased data scale. In other words, **intermediate representations unlock the scaling benefits of pre-training**. By selecting the most informative layer, Mantis exhibits a clear and monotonic improvement in performance as the number of pre-training samples increases.

**Layer-by-layer for other TSFMs.** For a more comprehensive comparison, we conduct a layer-by-layer analysis of several other foundation models, including NuTime (Lin et al., 2024), MOMENT (Goswami et al., 2024), TiRex (Auer et al., 2025a), and Chronos2 (Ansari et al., 2025). Figure 12 summarizes the results. TiRex and Chronos2 exhibit strong discriminative power in early layers, while their final layers appear to be more specialized for forecasting. MOMENT's performance plateaus after approximately the 10th layer, suggesting that the model could be significantly compressed by truncating its final layers. NuTime benefits the least from intermediate representations, with its best-performing layer located immediately before the final one.

### B.4. Aggregation of Output Tokens

As we leverage intermediate-layer representations, we have proposed a new output-token aggregation strategy, which we validate in this section. Specifically, we evaluate three approaches to generate the final embedding:

- using only the classification token (as in the original Mantis),

- computing the mean of all tokens except the classification token,

- concatenating the classification token with the mean of

the remaining tokens.

Figure 14 reports the corresponding results on the UCR benchmark. We observe that non-classification tokens encode complementary and discriminative information and should not be discarded. Moreover, concatenating the classification token with the mean token consistently yields the best performance, improving accuracy by $0.8\%$.

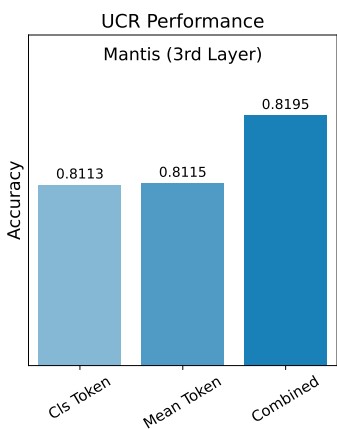

*Figure 14.* Ablation study on output-token aggregation.

We note that the effectiveness of this combined aggregation strategy may depend on the depth of the Transformer, as the number of layers determines how effectively the classification token aggregates information from the remaining tokens. We have performed the same experiment for the last layer of Mantis (Figure 15) and confirm this argument: the strategy is not beneficial anymore in this case, and the classification token is the most discriminative alone.

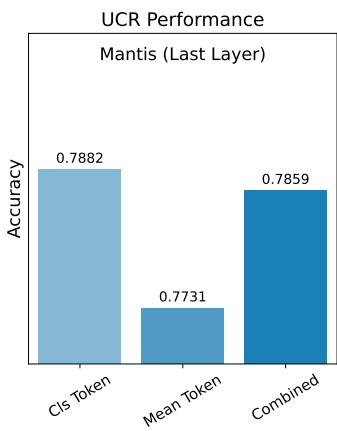

*Figure 15.* Ablation study on output-token aggregation.

### B.5. Self-Ensembling

First let us give more details on why different interpolation sizes lead to a different overlap between patches. In Figure 16, we illustrate how convolution + mean pooling generate one token for the same example but interpolated to two dif-

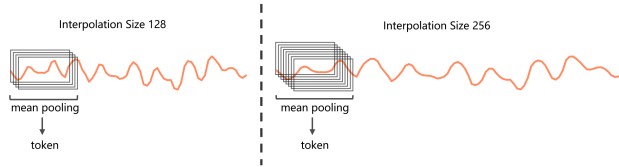

*Figure 16.* Motivation to look at different interpolation sizes that, in the context of our architecture, imply different degree of overlap between tokens.

ferent sizes. Since the number of tokens in our architecture is fixed, the pooling size increases with interpolation size, but not linearly. In this particular example, when the kernel size is set to 41, the receptive field of one token with interpolation sizes 128 and 256 will be $(41 + (128/32 - 1))/128 \approx 34\%$ and $(41 + (256/32 - 1))/256 \approx 19\%$, respectively. This implies that with different interpolation sizes the overlap between tokens will be different. As sequence length goes to infinity, the receptive field converges to $1/32$, i.e., to the case when the tokens are non-overlapping. Conversely, decreasing the sequence length makes tokens more and more overlapped.

We perform an ablation study over self-ensembling comparing the 4 following methods: (a) interpolate time series solely to 512 as we did before, (b) create 4 series by interpolating the input to 128, 256, 512 and 1024, pass all of them through the encoder and then concatenate all the embeddings, (c) perform the same multiple interpolations but for the first-order difference of the input, (d) concatenate the embeddings of (b) and (c). In Figure 17, the experimental results are displayed. One can see that multiple interpolations improve the performance Mantis by 0.55% on the UCR benchmark. Although the embeddings of the first-order difference are individually weak, their incorporation further improve the performance by 0.29%. It is an interesting observation since the feature extraction from the first-order differential is already incorporated to the architecture of Mantis, yet we can still benefit from this strategy.

## B.6. Pre-training Scheme

To validate the choice of contrastive pre-training, we investigate how Mantis performs when pre-trained using the following alternative schemes:

- Masked Autoencoder: we randomly mask a few non-overlapping patches of the original signal, pass the input through an encoder, and reconstruct the full signal with a decoder.

- Decoder-only: similar to next-token prediction, we mask the last patch and reconstruct it using a decoder-only architecture.

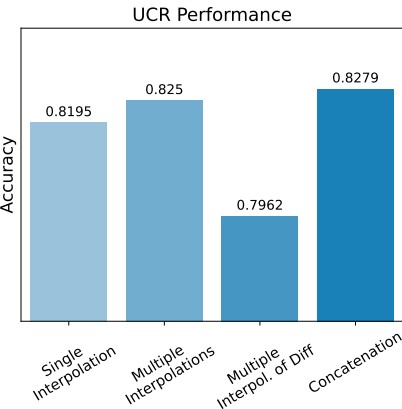

*Figure 17.* Self-Ensembling Ablation Study.

Table 9 shows the downstream performance averaged across 128 UCR datasets. Contrastive learning significantly outperforms the imputation-based schemes. Analyzing these results, we hypothesize that the specific architecture of Mantis may inherently be suboptimal for reconstruction-based pre-training. Identifying an optimal configuration for MAE and decoder-only setups remains a promising direction for future work, which could enable Mantis to perform well on forecasting tasks.

*Table 9.* Performance of Mantis with different pre-training schemes.

| Pre-training type | UCR Accuracy |
|---|---|
| Contrastive Learning | **0.788** |
| Masked Autoencoder | 0.716 |
| Decoder-only | 0.707 |

## B.7. Contrastive Loss

We conduct an ablation study on the contrastive learning scheme to evaluate our architectural and data choices. Specifically, we design two additional experiments. First, we pre-train Mantis with a triplet contrastive loss (TLoss, Franceschi et al., 2019) to evaluate it against our InfoNCE-based objective. The experimental results on the UCR benchmark, presented in Table 10, confirm the superior performance of our chosen pre-training loss.

*Table 10.* Performance of Mantis with different contrastive losses.

| Contrastive Loss | UCR Accuracy |
|---|---|
| Ours (InfoNCE) | **0.788** |
| TLoss | 0.767 |

Second, we validate the selection of the Random Crop Resize (RCR) augmentation used in our contrastive learning pipeline. We evaluate three alternative augmentations for comparison: (a) adding white noise (Noise), (b) a Fourier

transform surrogate (FTS, Rommel et al., 2022), and (c) Random Masking (RMask). Table 11 demonstrates the downstream performance on the UCR archive, confirming that RCR outperforms the other augmentation strategies.

*Table 11.* Performance of Mantis with different augmentation functions.

| Augmentation | UCR Accuracy |
|---|---|
| Noise | 0.751 |
| FTS | 0.768 |
| RMask | 0.698 |
| RCR | **0.788** |

## C. Additional Experiments

### C.1. Fine-tuning

In addition, we investigate the performance of Mantis when its encoder is fine-tuned to each dataset. We append a prediction head after an encoder and fine-tune all layers on the training data of a dataset. The prediction head is a layer normalization step followed by a linear layer. We fix a fine-tuning scheme: we minimize the cross-entropy loss for 500 epochs with a fixed batch size equal to 128, using an AdamW optimizer (Loshchilov et al., 2017) with a learning rate of $2 \cdot 10^{-4}$ and a weight decay of 0.05. We report the performance of a model at the last epoch in average over 3 experimental runs.

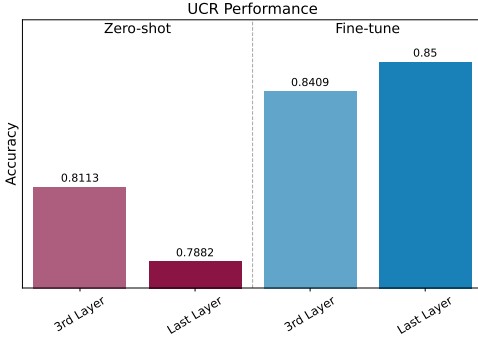

*Figure 18.* Performance of Mantis in zero-shot and fine-tuned regimes.

In Figure 18, we display the performance results for four configurations: (a) zero-shot feature extraction performance of the best intermediate layer, (b) zero-shot performance of the last layer, (c) the performance of fine-tuned Mantis when truncating all the layers after the best intermediate one, (d) fine-tuned Mantis without truncating. From the results we can see that the fine-tuning significantly boosts the performance, suggesting that Mantis can be easily adapted to specific downstream tasks. What is interesting is that it is reasonable to fine-tune the model without truncating the layers, despite its promise for the zero-shot feature extraction. We suggest that, despite of the low utility of the last

layers at the end of pre-training, they remain to be important for fine-tuning as they enrich the capacity of the model to fit the training data. It is important to mention that in Figure 18, we illustrated the zero-shot performance without other test-time optimization techniques such as output-token aggregation, self-ensembling and cross-model embedding fusion. Applying these techniques will substantially reduce the performance gap between the zero-shot and fine-tuning results.

### C.2. Adapters

The multivariate setting represents a primary challenge for time series foundation models since different tasks vary in the number of channels. We adopt the most common approach, which involves pre-training a TSFM univariately and subsequently treating all channels independently (Goswami et al., 2024; Auer et al., 2025b). We note that while the architecture of some foundation models naturally supports multivariate data (Cohen et al., 2025; Ansari et al., 2025), the gains from modeling cross-channel relationships do not appear significant (Ansari et al., 2025, Fig. 4a).

However, treating channels independently has its own limitations. First, cross-channel relationships are only considered at the prediction head stage. Second, during full fine-tuning with a large number of channels, this approach can lead to excessive resource consumption. To address these concerns, we follow Benechehab et al. (2025) and Ilbert et al. (2025) by prepending a channel-level adapter $a : \mathbb{R}^d \to \mathbb{R}^{d_{new}}$. This adapter precedes the foundation model and mixes the original $d$ channels into $d_{new}$ new ones, yielding a final embedding of $\mathbf{z} = \text{concat}\left[(F([a(\mathbf{x})]_i))_{1 \leq i \leq d_{new}}\right]$.

*Table 12.* Performance of Mantis with (PCA, Linear) and without (None) adapters on 10 multivariate datasets.

| Dataset | None | PCA | Linear |
|---|---|---|---|
| CharacterTrajectories | 0.976 | 0.977 | **0.988** |
| DuckDuckGeese | 0.54 | 0.58 | **0.6** |
| EigenWorms | 0.814 | **0.863** | 0.855 |
| FaceDetection | 0.549 | 0.553 | **0.592** |
| HandMovementDirection | 0.297 | **0.338** | **0.338** |
| Handwriting | 0.281 | 0.315 | **0.514** |
| MotorImagery | 0.467 | 0.48 | **0.5** |
| NATOPS | 0.889 | **0.939** | 0.906 |
| SelfRegulationSCP2 | 0.517 | **0.533** | **0.533** |
| SpokenArabicDigits | 0.937 | 0.955 | **0.987** |

We evaluate two types of adapters: (1) PCA: applies Principal Component Analysis across channels (i.e., flattening the sample and time axes) to project to the first $d_{new} = \min(10, d)$ principal components; and (2) Linear: utilizes a trainable linear layer to project $d$ channels to $d_{new} = 10$. Evaluating this approach against the default channel-independent pass, we observe improved performance across 10 UEA datasets (Table 12). Thus, adapters

offer a simple yet effective solution for enhancing performance on multivariate datasets.

## C.3. Mantis vs. TabPFN: Detailed Comparison

In our experiments, we observe that despite exhibiting a lower average accuracy overall, TabPFN achieves the highest win rate on the UCR archive (see Table 18 and Table 19). To better understand this dynamic, we conduct a deeper comparison between Mantis and TabPFN. Specifically, we compute the average number of training samples, sequence length, and number of classes across the UCR datasets where each model outperforms the other (Table 13). On average, Mantis excels on datasets characterized by fewer training samples, longer sequence lengths, and a higher number of unique classes.

*Table 13.* Average characteristics of UCR datasets where each model outperforms the other.

| Metric | TabPFN | Mantis |
|---|---|---|
| Num. of train samples | 521 | 436 |
| Sequence length | 452 | 597 |
| Num. of classes | 7.3 | 9.8 |

Furthermore, we stratify the UCR datasets based on these characteristics to analyze subgroup performance.

**Number of samples** (Table 14). Mantis outperforms TabPFN in all intervals except for large training sets ($n \geq 400$). As TabPFN operates as a supervised feature extractor, its representation quality scales with the number of available samples. It excels with larger sample sizes, albeit at a high computational cost as discussed in Section 4.3.

*Table 14.* UCR performance stratified by the number of training samples ($n$).

| Subgroup | TabPFN | Mantis |
|---|---|---|
| $16 \leq n < 55$ | 0.857 | **0.91** |
| $55 \leq n < 200$ | 0.743 | **0.831** |
| $200 \leq n < 400$ | 0.654 | **0.703** |
| $n \geq 400$ | **0.868** | 0.834 |

**Sequence length** (Table 15). TabPFN outperforms Mantis on shorter sequences ($t < 144$), which, we hypothesize, more closely resemble standard tabular configurations, naturally playing to TabPFN's architectural strengths.

*Table 15.* UCR performance stratified by sequence length ($t$).

| Subgroup | TabPFN | Mantis |
|---|---|---|
| $15 \leq t < 144$ | **0.865** | 0.854 |
| $144 \leq t < 345$ | 0.829 | **0.857** |
| $345 \leq t < 720$ | 0.767 | **0.82** |
| $t \geq 720$ | 0.661 | **0.748** |

**Number of classes** (Table 16). In this breakdown, Mantis consistently outperforms TabPFN across all subsets. Notably, TabPFN struggles severely with extreme few-shot classification problems such as PigCVP, PigAirwayPressure, and PigArtPressure, which have a high class count (52 classes) alongside only 2 training points per class. In these scenarios, TabPFN's performance degrades substantially, whereas Mantis remains highly robust.

*Table 16.* UCR performance stratified by the number of classes ($K$).

| Subgroup | TabPFN | Mantis |
|---|---|---|
| $K < 4$ | 0.843 | **0.871** |
| $4 \leq K < 10$ | 0.8 | **0.802** |
| $10 \leq K < 20$ | 0.735 | **0.771** |
| $K \geq 20$ | 0.529 | **0.713** |

# D. Complete Results

Below we provide full tables with experimental results.

## D.1. Broad Evaluation on UCR-91

In addition to 16 models considered in the main body of the paper, we add here 9 more baselines by extracting experimental results from (Ismail Fawaz et al., 2019) and (Goswami et al., 2024), including the official performance results for MOMENT, the self-supervised TS-TCC (Eldele et al., 2021), supervised deep learning methods such as CNN (Zhao et al., 2017), Encoder (Serra et al., 2018), TWIESN (Tanisaro and Heidemann, 2016), FCN, MLP and ResNet (Wang et al., 2017), and statistical methods such as DTW (Dau et al., 2019). In Figure 19, we illustrate the average performance over 91 UCR datasets for all considered models. Table 17 show the complete results per dataset.

## D.2. Performance Breakdown.

We attach the complete results for UCR in Table 18 and Table 19, and in Table 20 for UEA-27. In addtion, we provide the full results for self-ensembling and cross-model embedding fusion in Table 21 and Table 22.

*Table 17.* Comparison with more baselines on 91 datasets of UCR, which results are taken from (Goswami et al., 2024).

| Dataset | TS2Vec | T-Loss | TNC | TS-TCC | CNN | Encoder | FCN | MLP | ResNet | TWIESN | DTW | Official MOMENT | Improved MOMENT | Catch22+ | TabPFN | TabICL | TiRex | Chronos2 | TiViT-H | TiConvNext | NuTime | Mantis | SE-Mantis | Mantis & TiViT-H | Mantis & TiConvNext |
|---|---|---|---|---|---|---|---|---|---|---|---|---|---|---|---|---|---|---|---|---|---|---|---|---|---|
| Adiac | 0.762 | 0.675 | 0.767 | 0.767 | 0.393 | 0.318 | 0.841 | 0.391 | 0.833 | 0.428 | 0.604 | 0.688 | 0.7775 | 0.734 | 0.8031 | 0.8031 | 0.7826 | 0.8286 | 0.7187 | 0.7059 | 0.8005 | 0.8363 | **0.8465** | 0.7519 | 0.7749 |
| AllGestureWiimoteX | **0.777** | 0.763 | 0.697 | 0.697 | 0.411 | 0.475 | 0.713 | 0.477 | 0.741 | 0.522 | 0.716 | 0.607 | 0.6614 | 0.5957 | 0.6229 | 0.5043 | 0.7014 | 0.6843 | 0.67 | 0.6943 | 0.6071 | 0.73 | 0.6871 | 0.7229 | 0.7357 |
| AllGestureWiimoteY | 0.793 | 0.726 | 0.741 | 0.741 | 0.479 | 0.509 | 0.784 | 0.571 | **0.794** | 0.6 | 0.729 | 0.666 | 0.7029 | 0.6319 | 0.6329 | 0.5114 | 0.7314 | 0.7043 | 0.7371 | 0.7229 | 0.6529 | 0.7471 | 0.7271 | 0.7557 | 0.7486 |
| AllGestureWiimoteZ | **0.746** | 0.723 | 0.689 | 0.689 | 0.375 | 0.396 | 0.692 | 0.439 | 0.726 | 0.516 | 0.643 | 0.537 | 0.6057 | 0.5462 | 0.5329 | 0.4529 | 0.6343 | 0.66 | 0.6771 | 0.6586 | 0.5414 | 0.6814 | 0.6357 | 0.6886 | 0.7043 |
| ArrowHead | 0.857 | 0.766 | 0.737 | 0.737 | 0.717 | 0.63 | 0.843 | 0.784 | 0.838 | 0.689 | 0.703 | 0.743 | 0.7771 | 0.741 | 0.7543 | 0.7429 | 0.7829 | 0.8514 | 0.8229 | 0.8229 | 0.7257 | 0.8514 | 0.8514 | **0.88** | 0.8457 |
| BME | 0.993 | 0.993 | 0.933 | 0.933 | 0.947 | 0.827 | 0.836 | 0.905 | 0.999 | 0.819 | 0.9 | 0.96 | 0.9933 | **1.0** | **1.0** | 0.98 | 0.9933 | **1.0** | **1.0** | **1.0** | 0.9667 | **1.0** | **1.0** | **1.0** | **1.0** |
| Beef | 0.767 | 0.667 | 0.6 | 0.6 | 0.767 | 0.707 | 0.68 | 0.713 | 0.753 | 0.527 | 0.633 | 0.833 | 0.8 | 0.6889 | 0.8 | **0.9333** | 0.7 | 0.7667 | 0.8333 | 0.7333 | 0.7333 | 0.8 | 0.7667 | 0.8333 | |
| BeetleFly | 0.9 | 0.8 | 0.8 | 0.8 | 0.9 | 0.62 | 0.91 | 0.88 | 0.85 | 0.79 | 0.7 | 0.9 | **0.95** | 0.7833 | 0.9 | 0.8 | 0.9 | 0.8 | 0.85 | **0.95** | 0.7 | 0.85 | **0.95** | **0.95** | 0.95 |
| BirdChicken | 0.8 | 0.85 | 0.65 | 0.65 | 0.71 | 0.51 | 0.94 | 0.74 | 0.88 | 0.62 | 0.75 | 0.85 | **1.0** | 0.85 | 0.75 | 0.9 | 0.9 | 0.95 | **1.0** | 0.9 | 0.9 | 0.9 | 0.9 | 0.9 | |
| CBF | **1.0** | 0.983 | 0.998 | 0.998 | 0.959 | 0.977 | 0.994 | 0.869 | 0.996 | 0.896 | 0.997 | 0.96 | 0.9889 | 0.9763 | 0.9133 | 0.9244 | 0.9967 | **1.0** | 0.9989 | 0.9989 | 0.99 | 0.9967 | 0.9944 | 0.9989 | **1.0** |
| Chinatown | 0.965 | 0.951 | 0.983 | 0.983 | 0.977 | 0.966 | 0.98 | 0.872 | 0.978 | 0.825 | 0.957 | 0.965 | 0.9825 | 0.9796 | **0.9854** | 0.9796 | 0.9825 | **0.9854** | 0.9679 | 0.9388 | 0.9621 | 0.9621 | 0.9534 | 0.9592 | 0.9504 |
| ChlorineConcentration | 0.832 | 0.749 | 0.753 | 0.753 | 0.608 | 0.583 | 0.817 | 0.8 | 0.853 | 0.554 | 0.648 | 0.765 | 0.7789 | 0.6682 | 0.95 | **0.9773** | 0.8018 | 0.7672 | 0.7622 | 0.7771 | 0.6086 | 0.7115 | 0.7867 | 0.7914 | 0.8026 |
| Coffee | **1.0** | **1.0** | **1.0** | **1.0** | **1.0** | 0.886 | 1.0 | 0.993 | **1.0** | 0.979 | **1.0** | 0.893 | 0.9643 | 0.9881 | 0.9643 | **1.0** | **1.0** | **1.0** | **1.0** | **1.0** | **1.0** | **1.0** | **1.0** | **1.0** | **1.0** |
| CricketX | 0.782 | 0.713 | 0.731 | 0.731 | 0.535 | 0.644 | 0.794 | 0.591 | 0.799 | 0.627 | 0.754 | 0.749 | 0.7436 | 0.6974 | 0.6667 | 0.6641 | 0.7 | 0.7564 | 0.759 | 0.7487 | 0.7051 | 0.7821 | **0.8** | 0.7769 | 0.7718 |
| CricketY | 0.749 | 0.728 | 0.718 | 0.718 | 0.582 | 0.639 | 0.793 | 0.598 | 0.81 | 0.652 | 0.744 | 0.746 | 0.7077 | 0.6923 | 0.7 | 0.6333 | 0.7538 | 0.7359 | 0.7872 | 0.7462 | 0.6897 | 0.8154 | **0.8231** | 0.8026 | |
| CricketZ | 0.792 | 0.708 | 0.713 | 0.713 | 0.501 | 0.651 | 0.81 | 0.629 | 0.809 | 0.643 | 0.754 | 0.731 | 0.7436 | 0.7427 | 0.6718 | 0.6692 | 0.7333 | 0.7231 | 0.7974 | 0.759 | 0.659 | 0.8128 | **0.8179** | 0.8103 | 0.7846 |
| Crop | 0.756 | 0.722 | 0.742 | 0.742 | 0.67 | 0.76 | 0.738 | 0.618 | 0.743 | 0.489 | 0.665 | 0.734 | 0.7234 | 0.7523 | 0.7989 | **0.812** | 0.7029 | 0.7163 | 0.6895 | 0.6737 | 0.6895 | 0.7348 | 0.732 | 0.7231 | 0.7171 |
| DiatomSizeReduction | **0.984** | **0.984** | 0.977 | 0.977 | 0.954 | 0.88 | 0.346 | 0.909 | 0.301 | 0.914 | 0.967 | 0.879 | 0.9412 | 0.9401 | 0.9608 | 0.951 | 0.9281 | 0.9477 | 0.9673 | 0.9673 | 0.915 | 0.9281 | 0.9281 | 0.9575 | 0.9706 |
| DistalPhalanxOutlineAgeGroup | 0.727 | 0.727 | 0.755 | 0.755 | 0.758 | 0.761 | 0.718 | 0.647 | 0.718 | 0.705 | 0.77 | 0.669 | 0.6691 | 0.717 | 0.7626 | 0.7482 | 0.741 | 0.7122 | 0.7266 | 0.705 | **0.777** | 0.7122 | 0.705 | 0.7338 | |
| DistalPhalanxOutlineCorrect | 0.761 | 0.775 | 0.754 | 0.754 | 0.772 | 0.724 | 0.76 | 0.727 | 0.77 | 0.711 | 0.717 | 0.717 | 0.7536 | **0.7911** | 0.7826 | 0.7754 | 0.75 | 0.7645 | 0.7536 | 0.7681 | 0.7428 | 0.7681 | 0.7536 | 0.75 | 0.7681 |
| DistalPhalanxTW | 0.698 | 0.676 | 0.676 | 0.676 | 0.671 | 0.694 | 0.695 | 0.61 | 0.663 | 0.591 | 0.59 | 0.612 | 0.6259 | 0.6475 | 0.6978 | 0.6835 | 0.6547 | 0.6547 | 0.6691 | 0.6547 | 0.6835 | 0.6835 | **0.705** | 0.6691 | 0.6763 |
| DodgerLoopDay | 0.562 | nan | nan | nan | 0.312 | 0.487 | 0.143 | 0.16 | 0.15 | 0.593 | 0.5 | 0.438 | 0.4375 | 0.6417 | 0.6125 | **0.725** | 0.55 | 0.525 | 0.5 | 0.5625 | 0.55 | 0.5 | 0.525 | 0.5125 | 0.55 |
| DodgerLoopGame | 0.841 | nan | nan | nan | 0.816 | 0.81 | 0.768 | 0.865 | 0.71 | 0.716 | 0.877 | 0.623 | 0.8406 | 0.8333 | 0.7899 | 0.7971 | 0.7899 | **0.8913** | 0.8406 | 0.8406 | 0.8188 | 0.8623 | 0.8551 | 0.8188 | 0.8841 |
| DodgerLoopWeekend | 0.964 | nan | nan | nan | 0.974 | 0.983 | 0.904 | 0.978 | 0.952 | 0.954 | 0.949 | 0.826 | **0.9855** | **0.9855** | 0.9783 | 0.9565 | 0.9638 | 0.9493 | 0.913 | 0.9638 | 0.971 | **0.9855** | 0.9565 | | |
| ECG200 | 0.92 | **0.94** | 0.88 | 0.88 | 0.816 | 0.884 | 0.888 | 0.914 | 0.874 | 0.874 | 0.77 | 0.76 | 0.9 | 0.85 | 0.89 | 0.88 | 0.83 | 0.84 | 0.86 | 0.86 | 0.85 | 0.88 | 0.87 | 0.85 | 0.87 |
| ECG5000 | 0.935 | 0.933 | 0.941 | 0.941 | 0.928 | 0.941 | 0.94 | 0.935 | 0.935 | 0.922 | 0.924 | 0.942 | 0.9333 | 0.9398 | 0.9447 | 0.9391 | 0.9342 | 0.9333 | 0.9311 | 0.9193 | 0.9353 | 0.9329 | 0.9371 | 0.9289 | |
| ECGFiveDays | **1.0** | **1.0** | 0.878 | 0.878 | 0.874 | 0.842 | 0.985 | 0.973 | 0.966 | 0.723 | 0.768 | 0.804 | 0.9698 | 0.7975 | 0.9245 | 0.9826 | 0.9756 | 0.9907 | 0.9791 | 0.9895 | 0.899 | 0.993 | 0.9977 | 0.9895 | 0.9895 |
| Earthquakes | 0.748 | 0.748 | 0.748 | 0.748 | 0.709 | 0.74 | 0.725 | 0.727 | 0.712 | 0.748 | 0.719 | 0.748 | 0.705 | **0.7506** | 0.7482 | 0.7482 | 0.6906 | 0.7482 | 0.705 | 0.6978 | 0.7338 | 0.7194 | 0.7122 | 0.6978 | 0.7122 |
| ElectricDevices | 0.721 | 0.707 | 0.686 | 0.686 | 0.686 | 0.702 | 0.706 | 0.593 | 0.728 | 0.605 | 0.602 | 0.646 | 0.7404 | 0.7396 | 0.7025 | 0.6614 | 0.6709 | 0.7168 | 0.7635 | **0.7764** | 0.6786 | 0.7175 | 0.7257 | 0.7636 | 0.7679 |
| FaceAll | 0.771 | 0.786 | 0.813 | 0.813 | 0.774 | 0.794 | **0.938** | 0.794 | 0.867 | 0.673 | 0.808 | 0.791 | 0.7787 | 0.7682 | 0.8077 | 0.771 | 0.7876 | 0.7533 | 0.7521 | 0.7367 | 0.7207 | 0.7574 | 0.7568 | 0.7562 | 0.7444 |
| FaceFour | 0.932 | 0.92 | 0.773 | 0.773 | 0.905 | 0.852 | 0.93 | 0.836 | 0.955 | 0.857 | 0.83 | 0.852 | 0.875 | 0.8977 | 0.9091 | 0.8864 | 0.9545 | 0.9091 | 0.875 | 0.9205 | 0.9659 | **0.9886** | 0.9432 | 0.9659 | 0.9659 |
| FacesUCR | 0.924 | 0.884 | 0.863 | 0.863 | 0.873 | 0.867 | 0.943 | 0.831 | 0.954 | 0.641 | 0.905 | 0.811 | 0.8912 | 0.8766 | 0.8771 | 0.8776 | 0.8576 | 0.8751 | 0.8561 | 0.8195 | 0.9117 | 0.9083 | 0.9088 | 0.898 | |
| FiftyWords | 0.771 | 0.732 | 0.653 | 0.653 | 0.624 | 0.658 | 0.646 | 0.708 | 0.74 | 0.518 | 0.69 | 0.802 | 0.7758 | 0.726 | 0.7385 | 0.7165 | 0.7231 | 0.7956 | 0.7868 | 0.7714 | 0.7165 | 0.8198 | 0.8088 | **0.8264** | 0.8088 |
| Fish | 0.926 | 0.891 | 0.817 | 0.817 | 0.855 | 0.734 | 0.961 | 0.848 | 0.981 | 0.823 | 0.8 | 0.96 | 0.7619 | 0.88 | 0.8857 | 0.9029 | 0.9429 | 0.9714 | 0.9143 | 0.96 | 0.9714 | 0.96 | 0.9657 | 0.9829 | |
| FordA | 0.936 | 0.928 | 0.93 | 0.93 | 0.896 | 0.826 | 0.914 | 0.816 | 0.937 | 0.555 | 0.555 | 0.936 | 0.9061 | 0.9101 | 0.897 | 0.8758 | **0.947** | 0.928 | 0.9136 | 0.9227 | 0.9061 | 0.9432 | 0.9462 | 0.9242 | 0.9318 |
| FordB | 0.794 | 0.793 | 0.815 | 0.815 | 0.794 | 0.772 | 0.777 | 0.707 | 0.813 | 0.512 | 0.62 | 0.798 | 0.7556 | 0.7292 | 0.7556 | 0.7136 | 0.816 | 0.8074 | 0.7963 | 0.779 | 0.7654 | 0.8099 | 0.8259 | 0.8136 | 0.8111 |
| FreezerRegularTrain | 0.986 | 0.956 | 0.989 | 0.989 | 0.987 | 0.76 | 0.997 | 0.906 | 0.998 | 0.946 | 0.899 | 0.982 | 0.9881 | **0.9996** | 0.9986 | 0.9877 | 0.9912 | 0.9951 | 0.9965 | 0.9975 | 0.994 | 0.9961 | 0.9954 | 0.9972 | 0.9982 |
| FreezerSmallTrain | 0.87 | 0.933 | 0.979 | 0.979 | 0.739 | 0.676 | 0.683 | 0.686 | 0.832 | 0.917 | 0.753 | 0.902 | 0.8186 | 0.9251 | 0.8933 | 0.8098 | 0.8912 | 0.9688 | 0.9863 | 0.9849 | 0.9906 | 0.9891 | 0.9895 | | |
| Fungi | 0.957 | **1.0** | 0.753 | 0.753 | 0.961 | 0.934 | 0.018 | 0.863 | 0.177 | 0.439 | 0.839 | 0.898 | **1.0** | 0.9229 | 0.8656 | 0.7849 | 0.9516 | 0.9462 | 0.9785 | 0.9946 | 0.7473 | 0.9731 | 0.9946 | 0.9839 | **1.0** |
| GestureMidAirD1 | 0.608 | 0.608 | 0.369 | 0.369 | 0.534 | 0.528 | 0.695 | 0.575 | 0.698 | 0.569 | 0.569 | 0.646 | 0.6846 | 0.6795 | 0.6231 | 0.6769 | 0.6923 | 0.7308 | 0.7538 | 0.7615 | 0.6846 | **0.7769** | 0.7769 | **0.7769** | 0.7538 |
| GestureMidAirD2 | 0.469 | 0.546 | 0.254 | 0.254 | 0.518 | 0.48 | 0.631 | 0.545 | 0.668 | 0.575 | 0.608 | 0.608 | 0.5769 | 0.6205 | 0.5615 | 0.5846 | 0.6154 | 0.7 | 0.6615 | 0.6769 | 0.5615 | 0.6462 | 0.6231 | 0.7 | 0.6923 |
| GestureMidAirD3 | 0.292 | 0.285 | 0.177 | 0.177 | 0.317 | 0.368 | 0.326 | 0.382 | 0.34 | 0.275 | 0.323 | 0.323 | 0.3692 | 0.3923 | 0.3692 | 0.3615 | 0.4077 | 0.3538 | 0.4462 | 0.4923 | 0.4462 | 0.4462 | 0.4308 | 0.4385 | 0.5 |
| GesturePebbleZ1 | 0.93 | 0.919 | 0.395 | 0.395 | 0.844 | 0.821 | 0.88 | 0.792 | 0.901 | 0.84 | 0.791 | 0.849 | 0.8953 | 0.8779 | 0.8488 | 0.8779 | 0.8837 | 0.9012 | 0.907 | 0.8547 | 0.8953 | 0.9302 | 0.9302 | **0.936** | 0.9302 |
| GesturePebbleZ2 | 0.873 | 0.899 | 0.43 | 0.43 | 0.778 | 0.796 | 0.781 | 0.701 | 0.777 | 0.843 | 0.671 | 0.816 | 0.8924 | 0.7848 | 0.7532 | 0.8418 | 0.8671 | 0.8544 | 0.8165 | 0.7722 | 0.9051 | 0.8924 | **0.9114** | 0.8797 | |
| GunPoint | 0.98 | 0.98 | 0.993 | 0.993 | 0.948 | 0.784 | **1.0** | 0.928 | 0.991 | 0.989 | 0.907 | 0.927 | **1.0** | 0.9467 | 0.9667 | 0.9533 | 0.98 | 0.9933 | 0.9933 | 0.9933 | 0.9733 | 0.98 | 0.98 | **1.0** | 0.9933 |
| GunPointAgeSpan | 0.987 | 0.994 | 0.994 | 0.994 | 0.912 | 0.89 | 0.996 | 0.934 | 0.997 | 0.965 | 0.918 | 0.962 | 0.9778 | 0.9778 | 0.9667 | 0.9842 | 0.9905 | 0.9905 | 0.9937 | 0.9873 | 0.9684 | 0.9937 | 0.9937 | 0.9968 | 0.9905 |
| GunPointMaleVersusFemale | **1.0** | 0.997 | 0.997 | 0.997 | 0.977 | 0.978 | 0.997 | 0.98 | 0.992 | 0.988 | 0.997 | 0.991 | 0.9968 | 0.9937 | 0.9968 | **1.0** | 0.9937 | 0.9937 | **1.0** | 0.9968 | 0.9873 | **1.0** | **1.0** | **1.0** | **1.0** |
| GunPointOldVersusYoung | **1.0** | **1.0** | **1.0** | **1.0** | 0.922 | 0.923 | 0.989 | 0.941 | 0.989 | 0.975 | 0.838 | 0.981 | 0.927 | **1.0** | **1.0** | **1.0** | 0.9651 | 0.9778 | 0.9873 | 0.9937 | **1.0** | 0.9968 | 0.9968 | 0.9968 | 0.9968 |
| Ham | 0.714 | 0.724 | 0.743 | 0.743 | 0.72 | 0.682 | 0.707 | 0.699 | 0.758 | **0.768** | 0.467 | 0.581 | 0.6571 | 0.6667 | 0.7429 | 0.7238 | 0.7048 | 0.6952 | 0.7429 | 0.7238 | 0.7333 | 0.7048 | 0.6667 | 0.7429 | 0.7333 |
| Herring | 0.641 | 0.594 | 0.594 | 0.594 | 0.531 | 0.512 | 0.644 | 0.491 | 0.6 | 0.625 | 0.531 | 0.594 | 0.6406 | 0.5208 | 0.5938 | 0.6406 | 0.6094 | 0.5156 | 0.5312 | 0.6094 | 0.5312 | **0.6875** | 0.6562 | 0.6562 | 0.6406 |
| InsectWingbeatSound | 0.63 | 0.597 | 0.415 | 0.415 | 0.585 | 0.63 | 0.392 | 0.604 | 0.499 | 0.435 | 0.355 | 0.607 | 0.6212 | 0.629 | **0.6672** | 0.6556 | 0.6253 | 0.6313 | 0.551 | 0.5652 | 0.5101 | 0.596 | 0.5793 | 0.5899 | 0.6081 |
| ItalyPowerDemand | 0.925 | 0.954 | 0.955 | 0.955 | 0.954 | 0.964 | 0.963 | 0.953 | 0.962 | 0.871 | 0.95 | 0.911 | 0.9543 | 0.9537 | **0.9699** | 0.9631 | 0.9602 | 0.964 | 0.9407 | 0.9378 | 0.9116 | 0.9456 | 0.9475 | 0.9427 | 0.9446 |
| Lightning7 | 0.863 | 0.795 | 0.685 | 0.685 | 0.647 | 0.696 | 0.825 | 0.616 | 0.827 | 0.68 | 0.726 | 0.726 | 0.7397 | 0.7671 | 0.6986 | 0.7397 | 0.8082 | 0.7945 | 0.8493 | 0.7671 | 0.7808 | 0.863 | 0.8219 | **0.8767** | 0.7945 |
| Meat | 0.95 | 0.95 | 0.883 | 0.883 | 0.913 | 0.787 | 0.803 | 0.893 | 0.99 | 0.97 | 0.933 | 0.917 | **1.0** | 0.9222 | 0.9833 | 0.9333 | 0.9 | 0.9333 | 0.8167 | 0.85 | 0.9333 | 0.9667 | 0.8833 | 0.8833 | 0.8833 |
| MedicalImages | 0.789 | 0.75 | 0.747 | 0.747 | 0.671 | 0.664 | 0.778 | 0.719 | 0.77 | 0.649 | 0.737 | 0.642 | 0.7461 | 0.7237 | 0.7947 | **0.8079** | 0.725 | 0.7474 | 0.7316 | 0.7513 | 0.7092 | 0.7711 | 0.7592 | 0.7566 | 0.7711 |
| MelbournePedestrian | 0.959 | 0.944 | 0.949 | 0.949 | 0.813 | 0.884 | 0.912 | 0.863 | 0.909 | 0.73 | 0.791 | 0.876 | 0.8954 | 0.9597 | **0.9803** | **0.9803** | 0.8938 | 0.9049 | 0.8626 | 0.87 | 0.9332 | 0.9582 | 0.9537 | 0.9377 | 0.9426 |
| MiddlePhalanxOutlineAgeGroup | 0.636 | **0.656** | 0.63 | 0.63 | 0.534 | 0.577 | 0.545 | 0.522 | 0.545 | 0.578 | 0.5 | 0.461 | 0.5065 | 0.5065 | 0.6234 | 0.6299 | 0.487 | 0.5519 | 0.5584 | 0.526 | 0.5519 | 0.5455 | 0.6455 | 0.539 | 0.539 |
| MiddlePhalanxOutlineCorrect | 0.838 | 0.825 | 0.818 | 0.818 | 0.744 | 0.752 | 0.795 | 0.75 | 0.825 | 0.826 | 0.743 | 0.698 | 0.467 | 0.8076 | 0.811 | **0.8522** | 0.8351 | 0.8076 | 0.8351 | 0.7938 | 0.8179 | 0.7491 | 0.8419 | 0.811 | 0.8007 |
| MiddlePhalanxTW | 0.584 | 0.591 | 0.61 | 0.61 | 0.551 | 0.597 | 0.501 | 0.536 | 0.495 | 0.506 | 0.532 | 0.5455 | 0.5844 | 0.6169 | **0.6234** | 0.5 | 0.5 | 0.5195 | 0.4805 | 0.4416 | 0.513 | 0.513 | 0.5195 | 0.5 | |
| MoteStrain | 0.861 | 0.851 | 0.843 | 0.843 | 0.885 | 0.824 | 0.936 | 0.855 | 0.924 | 0.809 | 0.835 | 0.774 | 0.905 | 0.8818 | 0.889 | 0.8794 | 0.9241 | 0.9257 | 0.9161 | 0.9065 | **0.9553** | 0.9393 | 0.9545 | 0.9481 | 0.9497 |
| OSULeaf | 0.851 | 0.76 | 0.723 | 0.723 | 0.482 | 0.554 | 0.979 | 0.56 | 0.98 | 0.62 | 0.591 | 0.785 | 0.7934 | 0.7642 | 0.5661 | 0.595 | 0.8384 | 0.938 | 0.938 | 0.8595 | 0.9628 | 0.9393 | **0.9917** | 0.9876 | 0.9835 |
| PhalangesOutlinesCorrect | 0.809 | 0.784 | 0.804 | 0.804 | 0.799 | 0.745 | 0.818 | 0.756 | 0.845 | 0.656 | 0.728 | 0.652 | 0.8135 | 0.8252 | 0.8403 | **0.8613** | 0.7925 | 0.8228 | 0.7995 | 0.7832 | 0.7471 | 0.7949 | 0.8054 | 0.7995 | 0.7972 |
| PickupGestureWiimoteZ | 0.82 | 0.74 | 0.6 | 0.6 | 0.608 | 0.496 | 0.744 | 0.604 | 0.704 | 0.66 | 0.66 | 0.62 | 0.7 | 0.6933 | 0.76 | 0.74 | 0.84 | 0.66 | **0.92** | 0.88 | 0.64 | 0.76 | 0.84 | **0.92** | 0.86 |
| Plane | **1.0** | 0.99 | **1.0** | **1.0** | 0.962 | 0.964 | **1.0** | 0.977 | **1.0** | **1.0** | **1.0** | 0.99 | **1.0** | **1.0** | 0.9905 | 0.9905 | **1.0** | **1.0** | **1.0** | **1.0** | **1.0** | **1.0** | **1.0** | **1.0** | **1.0** |
| PowerCons | 0.961 | 0.9 | 0.961 | 0.961 | 0.863 | **0.971** | 0.863 | 0.977 | 0.879 | 0.852 | 0.878 | 0.894 | 0.9556 | 0.9556 | **1.0** | **1.0** | 0.9389 | 0.9333 | 0.9056 | 0.9222 | 0.9722 | 0.9722 | 0.9556 | 0.9667 | 0.9556 |
| ProximalPhalanxOutlineAgeGroup | 0.834 | 0.844 | 0.839 | 0.839 | 0.812 | **0.872** | 0.825 | 0.849 | 0.847 | 0.839 | 0.805 | 0.863 | 0.8293 | 0.8472 | 0.8585 | 0.839 | 0.8488 | 0.8341 | 0.8293 | 0.8293 | 0.8537 | 0.839 | 0.8341 | 0.8293 | 0.8293 |
| ProximalPhalanxOutlineCorrect | 0.887 | 0.859 | 0.873 | 0.873 | 0.807 | 0.768 | 0.907 | 0.73 | 0.92 | 0.817 | 0.784 | 0.866 | 0.8866 | 0.866 | **0.9244** | 0.8866 | 0.8832 | 0.8729 | 0.8625 | 0.8385 | 0.8763 | 0.9107 | 0.8729 | 0.8557 | |
| ProximalPhalanxTW | 0.824 | 0.771 | 0.8 | 0.8 | 0.777 | 0.791 | 0.761 | 0.767 | 0.773 | 0.784 | 0.761 | 0.712 | 0.7951 | 0.8033 | 0.8098 | **0.8293** | 0.761 | 0.7561 | 0.761 | 0.7463 | 0.7854 | 0.761 | 0.7561 | 0.7561 | 0.761 |
| ShakeGestureWiimoteZ | 0.94 | 0.92 | 0.86 | 0.86 | 0.58 | 0.756 | 0.884 | 0.548 | 0.868 | 0.64 | 0.65 | 0.96 | 0.84 | 0.8467 | 0.82 | 0.74 | 0.88 | 0.92 | 0.84 | 0.86 | 0.9 | 0.92 | 0.9 | 0.88 | 0.88 |
| ShapeletSim | **1.0** | 0.672 | 0.683 | 0.683 | 0.497 | 0.51 | 0.706 | 0.513 | 0.752 | 0.546 | 0.65 | 0.961 | 0.9667 | 0.9704 | 0.4778 | 0.5056 | 0.9667 | **1.0** | 0.9278 | 0.9778 | 0.9944 | **1.0** | 1.0 | | |
| ShapesAll | 0.902 | 0.848 | 0.773 | 0.773 | 0.617 | 0.679 | 0.894 | 0.776 | **0.926** | 0.643 | 0.768 | 0.815 | 0.8483 | 0.8206 | 0.8017 | 0.7917 | 0.8633 | 0.8967 | 0.9117 | 0.8967 | 0.8717 | 0.9017 | 0.9167 | 0.925 | 0.9167 |
| SmoothSubspace | 0.98 | 0.96 | 0.953 | 0.953 | 0.976 | 0.964 | 0.975 | 0.98 | 0.98 | 0.849 | 0.827 | 0.82 | 0.98 | 0.9667 | **1.0** | 0.9667 | 0.9667 | 0.9667 | 0.98 | 0.9467 | 0.9733 | 0.96 | 0.9667 | 0.9667 | |
| SonyAIBORobotSurface1 | 0.903 | 0.902 | 0.899 | 0.899 | 0.69 | 0.729 | 0.958 | 0.692 | **0.961** | 0.725 | 0.725 | 0.729 | 0.8253 | 0.8397 | 0.772 | 0.6722 | 0.9368 | 0.8236 | 0.8636 | 0.8502 | 0.8153 | 0.8336 | 0.8952 | 0.8652 | 0.8552 |
| SonyAIBORobotSurface2 | 0.871 | 0.889 | 0.907 | 0.907 | 0.831 | 0.844 | **0.98** | 0.831 | 0.977 | 0.635 | 0.831 | 0.829 | 0.8909 | 0.809 | 0.8279 | 0.8909 | 0.8846 | 0.8835 | 0.937 | 0.9507 | 0.9486 | 0.9496 | | | |
| Strawberry | 0.962 | 0.954 | 0.965 | 0.965 | 0.952 | 0.959 | 0.975 | 0.959 | 0.98 | 0.911 | 0.941 | 0.951 | 0.9622 | 0.9333 | 0.9811 | **0.9838** | 0.9649 | 0.973 | 0.9568 | 0.9541 | 0.9595 | 0.9622 | 0.9622 | 0.9676 | 0.9541 |
| SwedishLeaf | 0.941 | 0.914 | 0.923 | 0.923 | 0.884 | 0.902 | 0.967 | 0.845 | 0.963 | 0.837 | 0.792 | 0.923 | 0.9312 | 0.9115 | 0.9504 | 0.9456 | 0.9472 | 0.9392 | **0.9728** | 0.936 | 0.968 | 0.9584 | | | |
| Symbols | 0.976 | 0.963 | 0.916 | 0.916 | 0.808 | 0.754 | 0.955 | 0.836 | 0.893 | 0.798 | 0.95 | 0.936 | 0.9558 | 0.9618 | 0.8824 | 0.8945 | 0.9668 | 0.9839 | 0.9859 | 0.9859 | 0.9688 | 0.9769 | 0.9839 | 0.9879 | **0.9889** |
| SyntheticControl | 0.997 | 0.987 | 0.99 | 0.99 | 0.987 | 0.973 | 0.989 | 0.973 | 0.997 | 0.879 | 0.993 | 0.99 | 0.9767 | 0.9922 | 0.99 | 0.9833 | 0.9933 | 0.9667 | **1.0** | 0.9733 | 0.99 | 0.9867 | **1.0** | | |
| ToeSegmentation1 | 0.917 | 0.939 | 0.93 | 0.93 | 0.598 | 0.706 | 0.961 | 0.589 | 0.957 | 0.882 | 0.772 | 0.925 | 0.9474 | 0.8553 | 0.5746 | 0.6667 | 0.9693 | 0.9167 | 0.9386 | 0.9035 | 0.8421 | 0.9649 | **0.9737** | 0.9561 | 0.9693 |
| ToeSegmentation2 | 0.892 | 0.9 | 0.877 | 0.877 | 0.752 | 0.702 | 0.889 | 0.745 | 0.894 | 0.794 | 0.838 | 0.915 | 0.7846 | 0.6538 | 0.8154 | 0.9154 | 0.8615 | 0.9 | 0.8769 | 0.8692 | **0.9308** | 0.9231 | 0.9154 | 0.9231 | |
| Trace | **1.0** | 0.99 | **1.0** | **1.0** | 0.952 | 0.74 | **1.0** | 0.806 | **1.0** | 0.934 | **1.0** | **1.0** | 0.99 | **1.0** | 0.91 | 0.98 | **1.0** | **1.0** | **1.0** | **1.0** | **1.0** | **1.0** | **1.0** | **1.0** | **1.0** |
| TwoLeadECG | 0.986 | 0.999 | 0.976 | 0.976 | 0.877 | 0.784 | 0.999 | 0.753 | **1.0** | 0.905 | 0.847 | 0.9921 | 0.8584 | 0.9508 | 0.9254 | 0.9851 | 0.993 | 0.9622 | 0.9991 | 0.9987 | **1.0** | 0.9956 | | | |
| TwoPatterns | **1.0** | 0.999 | 0.999 | 0.999 | 0.991 | **1.0** | 0.87 | 0.948 | **1.0** | 0.875 | **1.0** | 0.994 | 0.9918 | 0.9935 | 0.995 | 0.9032 | 0.996 | 0.9948 | 0.998 | 0.999 | 0.9415 | 0.997 | 0.9992 | 0.9995 | 0.9992 |
| UMD | **1.0** | 0.993 | 0.986 | 0.986 | 0.96 | 0.771 | 0.948 | 0.949 | 0.99 | 0.835 | 0.993 | 0.993 | 0.9931 | 0.9931 | 0.9375 | 0.9583 | 0.9931 | 0.9931 | 0.9722 | 0.9931 | 0.9931 | 0.9931 | 0.9931 | | |
| UWaveGestureLibraryX | 0.795 | 0.785 | 0.733 | 0.733 | 0.721 | 0.771 | 0.754 | 0.768 | 0.781 | 0.608 | 0.728 | 0.821 | 0.7931 | 0.8062 | 0.8079 | 0.7984 | 0.7998 | 0.821 | 0.8314 | 0.8417 | 0.8004 | 0.8445 | 0.8495 | 0.8431 | **0.8498** |
| UWaveGestureLibraryY | 0.719 | 0.71 | 0.641 | 0.641 | 0.626 | 0.676 | 0.642 | 0.699 | 0.666 | 0.497 | 0.634 | 0.738 | 0.7144 | 0.715 | 0.7164 | 0.7074 | 0.7387 | 0.7741 | 0.7744 | 0.6957 | 0.7658 | 0.7834 | 0.7783 | **0.787** | |
| UWaveGestureLibraryZ | 0.77 | 0.757 | 0.69 | 0.69 | 0.63 | 0.684 | 0.727 | 0.697 | 0.749 | 0.573 | 0.658 | 0.765 | 0.7401 | 0.7519 | 0.7513 | 0.7379 | 0.7328 | 0.7552 | 0.7739 | 0.7744 | 0.7554 | 0.7747 | **0.7956** | 0.7887 | 0.7859 |
| Wafer | 0.998 | 0.992 | 0.994 | 0.994 | 0.961 | 0.998 | 0.997 | 0.996 | 0.998 | 0.98 | 0.994 | 0.994 | 0.9959 | 0.9959 | 0.9977 | **0.9998** | 0.9961 | 0.9969 | 0.9995 | 0.9995 | | | | | |
| Wine | 0.87 | 0.815 | 0.778 | 0.778 | 0.519 | 0.556 | 0.611 | 0.541 | 0.722 | 0.744 | 0.574 | 0.537 | 0.8333 | 0.6358 | 0.7778 | 0.7222 | 0.8704 | 0.8519 | 0.6667 | 0.8333 | 0.7778 | 0.8333 | 0.8148 | 0.8333 | **0.8889** |
| WordSynonyms | 0.676 | 0.691 | 0.531 | 0.531 | 0.568 | 0.557 | 0.565 | 0.599 | 0.647 | 0.518 | 0.649 | 0.685 | 0.685 | 0.6442 | 0.5972 | 0.5987 | 0.6677 | 0.6775 | 0.6771 | 0.6003 | 0.7116 | 0.7194 | **0.732** | 0.721 | |
| Yoga | 0.887 | 0.837 | 0.791 | 0.791 | 0.786 | 0.753 | 0.837 | 0.856 | 0.867 | 0.626 | 0.837 | 0.834 | 0.8507 | 0.8289 | 0.8603 | 0.8653 | 0.7947 | 0.819 | 0.8403 | 0.8793 | 0.7493 | 0.8557 | 0.887 | 0.8713 | **0.901** |
| *Avg* | 0.8516 | 0.8336 | 0.7933 | 0.7933 | 0.752 | 0.7433 | 0.8093 | 0.7506 | 0.8255 | 0.7268 | 0.7641 | 0.7941 | 0.8338 | 0.8145 | 0.8173 | 0.8083 | 0.8394 | 0.8415 | 0.8436 | 0.8458 | 0.8121 | 0.8584 | 0.8582 | 0.8612 | **0.862** |
| *Best Counts* | 13 | 7 | 4 | 4 | 1 | 2 | 6 | 0 | 10 | 4 | 2 | 6 | 7 | 10.0 | 15.0 | 5 | 8 | 9 | 10 | 6 | 11 | 14 | **20** | 17 | |

*Table 18.* Comparison with baselines on UCR benchmark. First Part.

| | Catch22+ | TabPFN | TabICL | MOMENT | TiRex | Chronos2 | TiViT-H | TiConvNext | NuTime | Mantis |
|---|---|---|---|---|---|---|---|---|---|---|
| ACSF1 | $0.8233_{\pm0.021}$ | 0.8 | 0.81 | $0.82_{\pm0.026}$ | $0.85_{\pm0.01}$ | $0.8467_{\pm0.012}$ | $0.85_{\pm0.0}$ | $\mathbf{0.8667}_{\pm0.006}$ | $0.75_{\pm0.02}$ | $0.8233_{\pm0.012}$ |
| Adiac | $0.734_{\pm0.007}$ | 0.8031 | 0.8031 | $0.7894_{\pm0.003}$ | $0.7852_{\pm0.011}$ | $0.8107_{\pm0.009}$ | $0.7059_{\pm0.007}$ | $0.6547_{\pm0.018}$ | $0.7357_{\pm0.006}$ | $\mathbf{0.8483}_{\pm0.003}$ |
| AllGestureWiimoteX | $0.5957_{\pm0.01}$ | 0.6229 | 0.5043 | $0.6138_{\pm0.002}$ | $0.6495_{\pm0.012}$ | $0.6038_{\pm0.005}$ | $0.6029_{\pm0.011}$ | $0.6676_{\pm0.002}$ | $0.6505_{\pm0.005}$ | $\mathbf{0.6843}_{\pm0.006}$ |
| AllGestureWiimoteY | $0.6319_{\pm0.016}$ | 0.6329 | 0.5114 | $0.6624_{\pm0.004}$ | $\mathbf{0.7052}_{\pm0.019}$ | $0.6419_{\pm0.002}$ | $0.6514_{\pm0.01}$ | $0.6805_{\pm0.011}$ | $0.63_{\pm0.01}$ | $0.691_{\pm0.004}$ |
| AllGestureWiimoteZ | $0.5462_{\pm0.005}$ | 0.5329 | 0.4529 | $0.5767_{\pm0.007}$ | $0.6057_{\pm0.001}$ | $0.6205_{\pm0.004}$ | $0.6129_{\pm0.014}$ | $0.6067_{\pm0.011}$ | $0.6224_{\pm0.009}$ | $\mathbf{0.6881}_{\pm0.001}$ |
| ArrowHead | $0.741_{\pm0.012}$ | 0.7543 | 0.7429 | $0.7943_{\pm0.015}$ | $0.7886_{\pm0.01}$ | $0.779_{\pm0.037}$ | $0.7638_{\pm0.012}$ | $0.7714_{\pm0.006}$ | $0.7733_{\pm0.023}$ | $\mathbf{0.8057}_{\pm0.015}$ |
| BME | $\mathbf{1.0}_{\pm0.0}$ | 1.0 | 0.98 | $0.9867_{\pm0.007}$ | $0.96_{\pm0.007}$ | $0.9667_{\pm0.007}$ | $0.9778_{\pm0.017}$ | $0.9933_{\pm0.007}$ | $0.8467_{\pm0.035}$ | $0.9956_{\pm0.004}$ |
| Beef | $0.6889_{\pm0.051}$ | 0.8 | 0.7667 | $0.7889_{\pm0.019}$ | $\mathbf{0.8444}_{\pm0.019}$ | $0.7333_{\pm0.033}$ | $0.7556_{\pm0.019}$ | $0.8222_{\pm0.019}$ | $0.6667_{\pm0.067}$ | $0.6889_{\pm0.019}$ |
| BeetleFly | $0.7833_{\pm0.029}$ | 0.9 | 0.8 | $0.95_{\pm0.0}$ | $0.9167_{\pm0.029}$ | $0.8667_{\pm0.029}$ | $0.9333_{\pm0.029}$ | $0.95_{\pm0.0}$ | $0.8_{\pm0.05}$ | $\mathbf{0.9833}_{\pm0.029}$ |
| BirdChicken | $0.85_{\pm0.0}$ | 0.85 | 0.75 | $0.9833_{\pm0.029}$ | $0.8833_{\pm0.029}$ | $0.9_{\pm0.0}$ | $0.85_{\pm0.087}$ | $\mathbf{1.0}_{\pm0.0}$ | $0.9667_{\pm0.029}$ | $0.9_{\pm0.0}$ |
| CBF | $0.9763_{\pm0.002}$ | 0.9133 | 0.9244 | $0.9678_{\pm0.005}$ | $0.9915_{\pm0.003}$ | $0.993_{\pm0.002}$ | $\mathbf{0.9978}_{\pm0.001}$ | $0.9937_{\pm0.002}$ | $0.9726_{\pm0.002}$ | $0.9948_{\pm0.002}$ |
| Car | $0.7556_{\pm0.025}$ | 0.7833 | 0.8167 | $0.7556_{\pm0.025}$ | $0.6667_{\pm0.025}$ | $0.7389_{\pm0.025}$ | $0.7389_{\pm0.019}$ | $0.7222_{\pm0.035}$ | $0.7667_{\pm0.029}$ | $0.7556_{\pm0.025}$ |
| Chinatown | $0.9796_{\pm0.003}$ | $\mathbf{0.9854}$ | 0.9796 | $0.9776_{\pm0.002}$ | $0.9689_{\pm0.006}$ | $0.9718_{\pm0.004}$ | $0.9631_{\pm0.002}$ | $0.8717_{\pm0.02}$ | $0.9281_{\pm0.008}$ | $0.9534_{\pm0.003}$ |
| ChlorineConcentration | $0.6682_{\pm0.001}$ | 0.95 | $\mathbf{0.9773}$ | $0.6973_{\pm0.007}$ | $0.7018_{\pm0.004}$ | $0.7095_{\pm0.003}$ | $0.7076_{\pm0.001}$ | $0.7073_{\pm0.002}$ | $0.671_{\pm0.002}$ | $0.7041_{\pm0.001}$ |
| CinCECGTorso | $0.8872_{\pm0.013}$ | 0.8341 | 0.8225 | $0.6831_{\pm0.005}$ | $\mathbf{0.8976}_{\pm0.002}$ | $0.8135_{\pm0.018}$ | $0.8164_{\pm0.004}$ | $0.8101_{\pm0.012}$ | $0.7314_{\pm0.016}$ | $0.8089_{\pm0.01}$ |
| Coffee | $0.9881_{\pm0.021}$ | 0.9643 | 1.0 | $0.9286_{\pm0.0}$ | $0.9881_{\pm0.021}$ | $0.9286_{\pm0.0}$ | $\mathbf{1.0}_{\pm0.0}$ | $\mathbf{1.0}_{\pm0.0}$ | $0.9286_{\pm0.0}$ | $\mathbf{1.0}_{\pm0.0}$ |
| Computers | $0.7347_{\pm0.006}$ | 0.62 | 0.656 | $0.6267_{\pm0.022}$ | $0.76_{\pm0.011}$ | $0.7413_{\pm0.008}$ | $0.7333_{\pm0.008}$ | $0.7413_{\pm0.014}$ | $\mathbf{0.7627}_{\pm0.01}$ | $0.7213_{\pm0.012}$ |
| CricketX | $0.6974_{\pm0.01}$ | 0.6667 | 0.6641 | $0.6855_{\pm0.01}$ | $0.6923_{\pm0.017}$ | $0.6991_{\pm0.017}$ | $0.7137_{\pm0.005}$ | $0.6641_{\pm0.003}$ | $0.6667_{\pm0.01}$ | $\mathbf{0.7735}_{\pm0.011}$ |
| CricketY | $0.6923_{\pm0.008}$ | 0.7 | 0.6333 | $0.6735_{\pm0.005}$ | $0.7222_{\pm0.01}$ | $0.7137_{\pm0.012}$ | $0.7128_{\pm0.012}$ | $0.665_{\pm0.004}$ | $0.6889_{\pm0.01}$ | $\mathbf{0.788}_{\pm0.02}$ |
| CricketZ | $0.7427_{\pm0.008}$ | 0.6718 | 0.6692 | $0.7487_{\pm0.017}$ | $0.7248_{\pm0.004}$ | $0.7299_{\pm0.004}$ | $0.7641_{\pm0.007}$ | $0.712_{\pm0.008}$ | $0.6778_{\pm0.005}$ | $\mathbf{0.8171}_{\pm0.005}$ |
| Crop | $0.7523_{\pm0.001}$ | 0.7989 | $\mathbf{0.812}$ | $0.7141_{\pm0.001}$ | $0.7201_{\pm0.001}$ | $0.7208_{\pm0.003}$ | $0.6552_{\pm0.002}$ | $0.6409_{\pm0.002}$ | $0.6754_{\pm0.001}$ | $0.7341_{\pm0.0}$ |
| DiatomSizeReduction | $0.9401_{\pm0.008}$ | $\mathbf{0.9608}$ | 0.951 | $0.8824_{\pm0.006}$ | $0.8617_{\pm0.005}$ | $0.8998_{\pm0.014}$ | $0.8649_{\pm0.016}$ | $0.9009_{\pm0.008}$ | $0.8617_{\pm0.011}$ | $0.8301_{\pm0.02}$ |
| DistalPhalanxOutlineAgeGroup | $0.717_{\pm0.004}$ | 0.7626 | 0.7626 | $0.753_{\pm0.004}$ | $0.741_{\pm0.014}$ | $\mathbf{0.7818}_{\pm0.015}$ | $\mathbf{0.7818}_{\pm0.008}$ | $0.777_{\pm0.007}$ | $0.7386_{\pm0.004}$ | $0.7602_{\pm0.008}$ |
| DistalPhalanxOutlineCorrect | $0.7911_{\pm0.002}$ | 0.7826 | 0.7754 | $0.7959_{\pm0.006}$ | $0.7874_{\pm0.008}$ | $\mathbf{0.8031}_{\pm0.002}$ | $0.779_{\pm0.01}$ | $0.7548_{\pm0.004}$ | $0.7778_{\pm0.006}$ | $0.7886_{\pm0.002}$ |
| DistalPhalanxTW | $0.6475_{\pm0.019}$ | 0.6978 | 0.6835 | $0.6571_{\pm0.004}$ | $0.6667_{\pm0.011}$ | $0.693_{\pm0.015}$ | $0.6715_{\pm0.004}$ | $0.6954_{\pm0.015}$ | $0.6882_{\pm0.018}$ | $\mathbf{0.7074}_{\pm0.011}$ |
| DodgerLoopDay | $0.6417_{\pm0.052}$ | 0.6125 | 0.725 | $0.4583_{\pm0.052}$ | $0.5042_{\pm0.019}$ | $0.4917_{\pm0.007}$ | $0.4417_{\pm0.014}$ | $0.5333_{\pm0.014}$ | $0.5417_{\pm0.029}$ | $0.5333_{\pm0.019}$ |
| DodgerLoopGame | $\mathbf{0.8333}_{\pm0.007}$ | 0.7899 | 0.7971 | $0.8116_{\pm0.007}$ | $0.7633_{\pm0.004}$ | $\mathbf{0.8333}_{\pm0.014}$ | $0.8164_{\pm0.004}$ | $0.7826_{\pm0.0}$ | $0.7826_{\pm0.007}$ | $0.7995_{\pm0.022}$ |
| DodgerLoopWeekend | $0.9855_{\pm0.0}$ | 0.9855 | 0.9783 | $0.9758_{\pm0.004}$ | $0.9493_{\pm0.007}$ | $0.9638_{\pm0.013}$ | $0.9275_{\pm0.0}$ | $0.901_{\pm0.015}$ | $0.9589_{\pm0.004}$ | $0.9517_{\pm0.004}$ |
| ECG200 | $0.85_{\pm0.017}$ | 0.89 | 0.88 | $0.87_{\pm0.01}$ | $0.86_{\pm0.017}$ | $0.84_{\pm0.01}$ | $0.83_{\pm0.0}$ | $0.7833_{\pm0.025}$ | $0.8233_{\pm0.006}$ | $0.8367_{\pm0.012}$ |
| ECG5000 | $0.9398_{\pm0.001}$ | 0.942 | $\mathbf{0.9447}$ | $0.938_{\pm0.001}$ | $0.9331_{\pm0.002}$ | $0.9331_{\pm0.001}$ | $0.9403_{\pm0.001}$ | $0.9386_{\pm0.001}$ | $0.9332_{\pm0.0}$ | $0.9381_{\pm0.0}$ |
| ECGFiveDays | $0.7975_{\pm0.013}$ | 0.9245 | $\mathbf{0.9826}$ | $0.813_{\pm0.002}$ | $0.8955_{\pm0.022}$ | $0.9102_{\pm0.02}$ | $0.9346_{\pm0.015}$ | $0.971_{\pm0.004}$ | $0.7871_{\pm0.008}$ | $0.8707_{\pm0.001}$ |
| EOGHorizontalSignal | $0.5948_{\pm0.004}$ | 0.5276 | 0.5 | $0.5654_{\pm0.01}$ | $0.5313_{\pm0.02}$ | $0.5387_{\pm0.013}$ | $0.5387_{\pm0.007}$ | $0.5626_{\pm0.007}$ | $0.4521_{\pm0.008}$ | $0.5884_{\pm0.006}$ |
| EOGVerticalSignal | $0.5092_{\pm0.002}$ | 0.489 | 0.4337 | $0.453_{\pm0.008}$ | $0.4144_{\pm0.003}$ | $0.384_{\pm0.003}$ | $0.442_{\pm0.011}$ | $0.3757_{\pm0.007}$ | $0.267_{\pm0.002}$ | $0.4816_{\pm0.011}$ |
| Earthquakes | $\mathbf{0.7506}_{\pm0.008}$ | 0.7482 | 0.7482 | $0.7458_{\pm0.011}$ | $0.7482_{\pm0.0}$ | $0.7458_{\pm0.004}$ | $0.741_{\pm0.0}$ | $0.7458_{\pm0.004}$ | $0.7434_{\pm0.004}$ | $0.7434_{\pm0.004}$ |
| ElectricDevices | $0.7396_{\pm0.003}$ | 0.7025 | 0.6614 | $0.7258_{\pm0.002}$ | $0.704_{\pm0.003}$ | $0.7489_{\pm0.002}$ | $\mathbf{0.762}_{\pm0.003}$ | $0.7617_{\pm0.002}$ | $0.7066_{\pm0.002}$ | $0.7424_{\pm0.004}$ |
| EthanolLevel | $0.38_{\pm0.013}$ | $\mathbf{0.848}$ | 0.694 | $0.44_{\pm0.007}$ | $0.306_{\pm0.004}$ | $0.4367_{\pm0.005}$ | $0.4047_{\pm0.005}$ | $0.348_{\pm0.007}$ | $0.3487_{\pm0.005}$ | $0.3793_{\pm0.014}$ |
| FaceAll | $0.7682_{\pm0.028}$ | $\mathbf{0.8077}$ | 0.771 | $0.7178_{\pm0.003}$ | $0.7651_{\pm0.015}$ | $0.7024_{\pm0.005}$ | $0.6911_{\pm0.006}$ | $0.6866_{\pm0.004}$ | $0.6487_{\pm0.002}$ | $0.7205_{\pm0.003}$ |
| FaceFour | $0.8977_{\pm0.02}$ | 0.9091 | 0.8864 | $0.7689_{\pm0.046}$ | $0.7917_{\pm0.046}$ | $0.6818_{\pm0.057}$ | $0.6477_{\pm0.05}$ | $0.8182_{\pm0.011}$ | $0.803_{\pm0.013}$ | $\mathbf{0.9167}_{\pm0.007}$ |
| FacesUCR | $0.8558_{\pm0.004}$ | 0.8766 | $\mathbf{0.8771}$ | $0.7959_{\pm0.001}$ | $0.7995_{\pm0.005}$ | $0.7603_{\pm0.004}$ | $0.7815_{\pm0.005}$ | $0.7665_{\pm0.006}$ | $0.714_{\pm0.009}$ | $0.8439_{\pm0.006}$ |
| FiftyWords | $0.726_{\pm0.006}$ | $\mathbf{0.7385}$ | 0.7165 | $0.7099_{\pm0.004}$ | $0.6381_{\pm0.01}$ | $0.6689_{\pm0.009}$ | $0.6425_{\pm0.003}$ | $0.674_{\pm0.006}$ | $0.6139_{\pm0.004}$ | $0.7106_{\pm0.001}$ |
| Fish | $0.7619_{\pm0.013}$ | 0.88 | 0.8857 | $0.8476_{\pm0.014}$ | $0.8324_{\pm0.012}$ | $0.8629_{\pm0.006}$ | $0.9029_{\pm0.011}$ | $0.9067_{\pm0.012}$ | $0.9162_{\pm0.007}$ | $\mathbf{0.9314}_{\pm0.0}$ |
| FordA | $0.9101_{\pm0.004}$ | 0.897 | 0.8758 | $0.8611_{\pm0.006}$ | $\mathbf{0.9409}_{\pm0.002}$ | $0.9174_{\pm0.002}$ | $0.8975_{\pm0.004}$ | $0.9081_{\pm0.001}$ | $0.8934_{\pm0.003}$ | $0.9303_{\pm0.0}$ |
| FordB | $0.7292_{\pm0.007}$ | 0.7556 | 0.7136 | $0.7374_{\pm0.002}$ | $\mathbf{0.814}_{\pm0.006}$ | $0.7918_{\pm0.004}$ | $0.7794_{\pm0.003}$ | $0.7626_{\pm0.01}$ | $0.7527_{\pm0.008}$ | $0.7979_{\pm0.004}$ |
| FreezerRegularTrain | $\mathbf{0.9996}_{\pm0.0}$ | 0.9986 | 0.9877 | $0.9108_{\pm0.005}$ | $0.9261_{\pm0.002}$ | $0.9718_{\pm0.005}$ | $0.9827_{\pm0.002}$ | $0.9908_{\pm0.001}$ | $0.9766_{\pm0.002}$ | $0.9857_{\pm0.001}$ |
| FreezerSmallTrain | $0.9251_{\pm0.002}$ | 0.8933 | 0.8098 | $0.7878_{\pm0.011}$ | $0.8208_{\pm0.016}$ | $0.9185_{\pm0.009}$ | $0.9353_{\pm0.005}$ | $\mathbf{0.9559}_{\pm0.005}$ | $0.9395_{\pm0.004}$ | $0.935_{\pm0.004}$ |
| Fungi | $0.9229_{\pm0.041}$ | 0.8656 | 0.7849 | $\mathbf{1.0}_{\pm0.0}$ | $0.8405_{\pm0.05}$ | $0.9104_{\pm0.008}$ | $0.914_{\pm0.014}$ | $0.8835_{\pm0.03}$ | $0.7007_{\pm0.031}$ | $0.9427_{\pm0.028}$ |
| GestureMidAirD1 | $0.6795_{\pm0.036}$ | 0.6231 | 0.6769 | $0.6744_{\pm0.009}$ | $0.6615_{\pm0.013}$ | $0.6692_{\pm0.023}$ | $0.7_{\pm0.008}$ | $\mathbf{0.7359}_{\pm0.016}$ | $0.6564_{\pm0.009}$ | $0.7128_{\pm0.018}$ |
| GestureMidAirD2 | $0.6205_{\pm0.016}$ | 0.5615 | 0.5846 | $0.5769_{\pm0.013}$ | $0.6513_{\pm0.031}$ | $\mathbf{0.7436}_{\pm0.024}$ | $0.6667_{\pm0.019}$ | $0.6795_{\pm0.012}$ | $0.5667_{\pm0.012}$ | $0.6154_{\pm0.013}$ |
| GestureMidAirD3 | $0.3923_{\pm0.031}$ | 0.3692 | 0.3615 | $0.3846_{\pm0.013}$ | $0.3897_{\pm0.031}$ | $0.3538_{\pm0.023}$ | $0.441_{\pm0.019}$ | $\mathbf{0.4692}_{\pm0.012}$ | $0.3923_{\pm0.008}$ | $0.4385_{\pm0.013}$ |
| GesturePebbleZ1 | $0.8779_{\pm0.006}$ | 0.8488 | 0.8779 | $0.8779_{\pm0.006}$ | $0.8702_{\pm0.007}$ | $0.8857_{\pm0.009}$ | $0.8663_{\pm0.0}$ | $0.8295_{\pm0.003}$ | $0.8915_{\pm0.003}$ | $\mathbf{0.9205}_{\pm0.007}$ |
| GesturePebbleZ2 | $0.7384_{\pm0.004}$ | 0.7848 | 0.7532 | $0.865_{\pm0.01}$ | $0.8755_{\pm0.013}$ | $0.8186_{\pm0.007}$ | $0.8439_{\pm0.02}$ | $0.8439_{\pm0.01}$ | $0.8376_{\pm0.01}$ | $\mathbf{0.9114}_{\pm0.011}$ |
| GunPoint | $0.9467_{\pm0.018}$ | 0.9667 | 0.9533 | $0.9867_{\pm0.007}$ | $0.9644_{\pm0.01}$ | $\mathbf{0.9956}_{\pm0.004}$ | $0.9933_{\pm0.0}$ | $0.9667_{\pm0.018}$ | $0.9422_{\pm0.004}$ | $0.9778_{\pm0.004}$ |
| GunPointAgeSpan | $0.9884_{\pm0.002}$ | $\mathbf{0.9905}$ | 0.9842 | $0.9525_{\pm0.002}$ | $0.9821_{\pm0.005}$ | $0.9852_{\pm0.002}$ | $0.9873_{\pm0.0}$ | $0.9673_{\pm0.01}$ | $0.9715_{\pm0.003}$ | $0.9842_{\pm0.003}$ |
| GunPointMaleVersusFemale | $0.9937_{\pm0.0}$ | 0.9968 | 1.0 | $0.9916_{\pm0.004}$ | $0.9947_{\pm0.004}$ | $0.9884_{\pm0.002}$ | $0.9968_{\pm0.003}$ | $0.9979_{\pm0.004}$ | $0.9789_{\pm0.005}$ | $\mathbf{1.0}_{\pm0.0}$ |
| GunPointOldVersusYoung | $\mathbf{1.0}_{\pm0.0}$ | 1.0 | 1.0 | $0.9598_{\pm0.005}$ | $0.9672_{\pm0.007}$ | $0.9778_{\pm0.003}$ | $0.9852_{\pm0.005}$ | $0.9937_{\pm0.0}$ | $\mathbf{1.0}_{\pm0.0}$ | $0.9968_{\pm0.0}$ |
| Ham | $0.6063_{\pm0.005}$ | $\mathbf{0.7429}$ | 0.7238 | $0.7333_{\pm0.019}$ | $0.7143_{\pm0.01}$ | $0.6413_{\pm0.015}$ | $0.6508_{\pm0.04}$ | $0.6349_{\pm0.02}$ | $0.7111_{\pm0.048}$ | $0.6698_{\pm0.031}$ |
| HandOutlines | $0.9036_{\pm0.004}$ | 0.9162 | $\mathbf{0.927}$ | $0.909_{\pm0.006}$ | $0.8757_{\pm0.016}$ | $0.9108_{\pm0.003}$ | $0.8874_{\pm0.009}$ | $0.8874_{\pm0.01}$ | $0.8964_{\pm0.007}$ | $0.9036_{\pm0.006}$ |
| Haptics | $0.4838_{\pm0.015}$ | 0.4708 | 0.461 | $0.5238_{\pm0.008}$ | $0.5271_{\pm0.014}$ | $\mathbf{0.5444}_{\pm0.009}$ | $0.5076_{\pm0.012}$ | $0.5141_{\pm0.008}$ | $0.474_{\pm0.006}$ | $0.5011_{\pm0.011}$ |
| Herring | $0.5208_{\pm0.024}$ | 0.5938 | 0.6406 | $0.5885_{\pm0.024}$ | $0.6354_{\pm0.065}$ | $\mathbf{0.6667}_{\pm0.018}$ | $0.5938_{\pm0.031}$ | $0.5573_{\pm0.033}$ | $0.6458_{\pm0.013}$ | $0.6302_{\pm0.018}$ |
| HouseTwenty | $0.9692_{\pm0.005}$ | 0.8487 | 0.7563 | $0.9384_{\pm0.005}$ | $0.9692_{\pm0.005}$ | $0.958_{\pm0.008}$ | $\mathbf{0.9776}_{\pm0.005}$ | $0.9496_{\pm0.008}$ | $0.8852_{\pm0.013}$ | $0.9524_{\pm0.013}$ |
| InlineSkate | $0.3891_{\pm0.005}$ | 0.3345 | 0.3436 | $0.3224_{\pm0.006}$ | $0.4491_{\pm0.014}$ | $0.4194_{\pm0.007}$ | $0.3848_{\pm0.012}$ | $\mathbf{0.4576}_{\pm0.006}$ | $0.32_{\pm0.007}$ | $0.3855_{\pm0.005}$ |
| InsectEPGRegularTrain | $\mathbf{1.0}_{\pm0.0}$ | 1.0 | 1.0 | $0.9304_{\pm0.002}$ | $0.9692_{\pm0.01}$ | $0.9518_{\pm0.007}$ | $\mathbf{1.0}_{\pm0.0}$ | $0.9893_{\pm0.006}$ | $\mathbf{1.0}_{\pm0.0}$ | $\mathbf{1.0}_{\pm0.0}$ |
| InsectEPGSmallTrain | $\mathbf{1.0}_{\pm0.0}$ | 1.0 | 1.0 | $0.8246_{\pm0.006}$ | $0.8889_{\pm0.002}$ | $0.8715_{\pm0.018}$ | $0.9451_{\pm0.026}$ | $0.9505_{\pm0.012}$ | $\mathbf{1.0}_{\pm0.0}$ | $\mathbf{1.0}_{\pm0.0}$ |
| InsectWingbeatSound | $0.629_{\pm0.003}$ | $\mathbf{0.6672}$ | 0.6556 | $0.6258_{\pm0.003}$ | $0.651_{\pm0.001}$ | $0.6271_{\pm0.003}$ | $0.5258_{\pm0.013}$ | $0.5365_{\pm0.004}$ | $0.5167_{\pm0.006}$ | $0.6056_{\pm0.009}$ |

*Table 19.* Comparison with baselines on UCR benchmark. Second Part.

| | Catch22+ | TabPFN | TabICL | MOMENT | TiRex | Chronos2 | TiViT-H | TiConvNext | NuTime | Mantis |
|---|---|---|---|---|---|---|---|---|---|---|
| ItalyPowerDemand | $0.9537_{\pm0.002}$ | $\mathbf{0.9699}$ | $0.9631$ | $0.9482_{\pm0.004}$ | $0.9624_{\pm0.004}$ | $0.9559_{\pm0.003}$ | $0.8776_{\pm0.003}$ | $0.9248_{\pm0.002}$ | $0.8737_{\pm0.004}$ | $0.9248_{\pm0.004}$ |
| LargeKitchenAppliances | $0.8169_{\pm0.009}$ | $0.6373$ | $0.696$ | $0.7342_{\pm0.007}$ | $0.8053_{\pm0.005}$ | $\mathbf{0.8729}_{\pm0.003}$ | $0.816_{\pm0.003}$ | $0.8702_{\pm0.002}$ | $0.7556_{\pm0.009}$ | $0.7662_{\pm0.006}$ |
| Lightning2 | $0.7322_{\pm0.009}$ | $0.6721$ | $0.7049$ | $\mathbf{0.7869}_{\pm0.033}$ | $0.7432_{\pm0.009}$ | $0.7268_{\pm0.009}$ | $0.7596_{\pm0.009}$ | $0.7432_{\pm0.025}$ | $0.6776_{\pm0.009}$ | $0.7268_{\pm0.009}$ |
| Lightning7 | $\mathbf{0.7671}_{\pm0.014}$ | $0.6986$ | $0.7397$ | $0.7352_{\pm0.029}$ | $0.7352_{\pm0.016}$ | $0.653_{\pm0.021}$ | $0.7489_{\pm0.029}$ | $0.7306_{\pm0.016}$ | $0.6575_{\pm0.027}$ | $0.7215_{\pm0.044}$ |
| Mallat | $0.9555_{\pm0.009}$ | $\mathbf{0.9689}$ | $0.9484$ | $0.9164_{\pm0.016}$ | $0.9166_{\pm0.027}$ | $0.9016_{\pm0.009}$ | $0.9032_{\pm0.007}$ | $0.8387_{\pm0.005}$ | $0.8311_{\pm0.005}$ | $0.8887_{\pm0.002}$ |
| Meat | $0.9222_{\pm0.01}$ | $\mathbf{0.9833}$ | $0.9333$ | $0.9222_{\pm0.017}$ | $0.8611_{\pm0.01}$ | $0.9333_{\pm0.017}$ | $0.8389_{\pm0.038}$ | $0.8667_{\pm0.0}$ | $0.9278_{\pm0.019}$ | $0.9111_{\pm0.025}$ |
| MedicalImages | $0.7737_{\pm0.005}$ | $0.7947$ | $\mathbf{0.8079}$ | $0.714_{\pm0.008}$ | $0.7092_{\pm0.007}$ | $0.7246_{\pm0.009}$ | $0.7504_{\pm0.005}$ | $0.7368_{\pm0.0}$ | $0.7132_{\pm0.002}$ | $0.7522_{\pm0.006}$ |
| MelbournePedestrian | $0.9597_{\pm0.0}$ | $\mathbf{0.9803}$ | $\mathbf{0.9803}$ | $0.8643_{\pm0.0}$ | $0.8812_{\pm0.002}$ | $0.8927_{\pm0.006}$ | $0.8074_{\pm0.004}$ | $0.8195_{\pm0.003}$ | $0.9172_{\pm0.002}$ | $0.9513_{\pm0.001}$ |
| MiddlePhalanxOutlineAgeGroup | $0.5952_{\pm0.004}$ | $0.6234$ | $\mathbf{0.6299}$ | $0.5649_{\pm0.017}$ | $0.5996_{\pm0.016}$ | $0.5584_{\pm0.017}$ | $0.6039_{\pm0.006}$ | $\mathbf{0.6299}_{\pm0.017}$ | $0.6061_{\pm0.004}$ | $0.5801_{\pm0.004}$ |
| MiddlePhalanxOutlineCorrect | $0.811_{\pm0.012}$ | $0.8522$ | $0.8351$ | $0.858_{\pm0.008}$ | $0.8179_{\pm0.007}$ | $\mathbf{0.8717}_{\pm0.008}$ | $0.8247_{\pm0.009}$ | $0.8156_{\pm0.012}$ | $0.7892_{\pm0.009}$ | $0.8339_{\pm0.004}$ |
| MiddlePhalanxTW | $0.5844_{\pm0.006}$ | $0.6169$ | $\mathbf{0.6234}$ | $0.5779_{\pm0.017}$ | $0.5736_{\pm0.01}$ | $0.5714_{\pm0.013}$ | $0.5519_{\pm0.023}$ | $0.5693_{\pm0.019}$ | $0.5281_{\pm0.014}$ | $0.5325_{\pm0.006}$ |
| MixedShapesRegularTrain | $0.9306_{\pm0.003}$ | $0.9344$ | $0.9299$ | $0.9166_{\pm0.001}$ | $0.9472_{\pm0.001}$ | $0.9427_{\pm0.001}$ | $0.9513_{\pm0.001}$ | $\mathbf{0.9564}_{\pm0.003}$ | $0.9381_{\pm0.002}$ | $0.9467_{\pm0.0}$ |
| MixedShapesSmallTrain | $0.8827_{\pm0.002}$ | $0.8293$ | $0.8767$ | $0.8506_{\pm0.008}$ | $0.909_{\pm0.002}$ | $0.9019_{\pm0.003}$ | $\mathbf{0.919}_{\pm0.005}$ | $0.9182_{\pm0.002}$ | $0.908_{\pm0.003}$ | $0.9157_{\pm0.001}$ |
| MoteStrain | $0.8818_{\pm0.015}$ | $0.889$ | $0.8794$ | $0.8895_{\pm0.008}$ | $0.9193_{\pm0.003}$ | $0.9332_{\pm0.005}$ | $0.8586_{\pm0.005}$ | $0.9055_{\pm0.002}$ | $\mathbf{0.9481}_{\pm0.002}$ | $0.931_{\pm0.004}$ |
| NonInvasiveFetalECGThorax1 | $0.8877_{\pm0.004}$ | $\mathbf{0.941}$ | $0.9272$ | $0.8863_{\pm0.002}$ | $0.8656_{\pm0.0}$ | $0.8295_{\pm0.001}$ | $0.8137_{\pm0.005}$ | $0.8244_{\pm0.006}$ | $0.78_{\pm0.005}$ | $0.864_{\pm0.002}$ |
| NonInvasiveFetalECGThorax2 | $0.9091_{\pm0.0}$ | $0.9476$ | $0.9405$ | $0.9104_{\pm0.001}$ | $0.888_{\pm0.004}$ | $0.8675_{\pm0.001}$ | $0.8755_{\pm0.002}$ | $0.8692_{\pm0.004}$ | $0.8175_{\pm0.004}$ | $0.8867_{\pm0.0}$ |
| OSULeaf | $0.6832_{\pm0.009}$ | $0.5661$ | $0.595$ | $0.7521_{\pm0.007}$ | $0.9174_{\pm0.007}$ | $0.8953_{\pm0.01}$ | $0.9463_{\pm0.004}$ | $\mathbf{0.9793}_{\pm0.004}$ | $0.8003_{\pm0.005}$ | $0.9353_{\pm0.002}$ |
| OliveOil | $0.8444_{\pm0.019}$ | $\mathbf{0.9333}$ | $0.9$ | $0.8889_{\pm0.019}$ | $0.8778_{\pm0.019}$ | $0.8556_{\pm0.019}$ | $0.5778_{\pm0.038}$ | $0.8556_{\pm0.019}$ | $0.7_{\pm0.0}$ | $0.8667_{\pm0.033}$ |
| PLAID | $0.8752_{\pm0.009}$ | $0.7896$ | $0.5661$ | $0.7393_{\pm0.005}$ | $0.8684_{\pm0.003}$ | $0.8591_{\pm0.008}$ | $0.8709_{\pm0.007}$ | $\mathbf{0.892}_{\pm0.007}$ | $0.7765_{\pm0.004}$ | $0.8187_{\pm0.004}$ |
| PhalangesOutlinesCorrect | $0.8252_{\pm0.003}$ | $0.8403$ | $\mathbf{0.8613}$ | $0.8225_{\pm0.002}$ | $0.8197_{\pm0.005}$ | $0.8349_{\pm0.004}$ | $0.796_{\pm0.002}$ | $0.7949_{\pm0.003}$ | $0.7766_{\pm0.003}$ | $0.824_{\pm0.002}$ |
| Phoneme | $0.3216_{\pm0.005}$ | $0.1097$ | $0.1361$ | $0.2938_{\pm0.008}$ | $0.3771_{\pm0.004}$ | $\mathbf{0.3936}_{\pm0.002}$ | $0.355_{\pm0.001}$ | $0.3745_{\pm0.004}$ | $0.2913_{\pm0.008}$ | $0.3641_{\pm0.005}$ |
| PickupGestureWiimoteZ | $0.6933_{\pm0.012}$ | $0.76$ | $0.74$ | $0.5867_{\pm0.023}$ | $0.7333_{\pm0.023}$ | $0.68_{\pm0.0}$ | $\mathbf{0.82}_{\pm0.02}$ | $0.82_{\pm0.035}$ | $0.6667_{\pm0.031}$ | $0.7867_{\pm0.031}$ |
| PigAirwayPressure | $0.2372_{\pm0.003}$ | $0.0192$ | $0.1538$ | $0.1074_{\pm0.014}$ | $0.3462_{\pm0.029}$ | $0.3333_{\pm0.012}$ | $0.4744_{\pm0.029}$ | $\mathbf{0.5769}_{\pm0.005}$ | $0.3478_{\pm0.012}$ | $0.4904_{\pm0.017}$ |
| PigArtPressure | $0.891_{\pm0.007}$ | $0.0337$ | $0.2548$ | $0.5369_{\pm0.015}$ | $0.8734_{\pm0.02}$ | $0.8061_{\pm0.018}$ | $0.8173_{\pm0.027}$ | $0.9087_{\pm0.01}$ | $\mathbf{0.9359}_{\pm0.012}$ | $0.9343_{\pm0.003}$ |
| PigCVP | $0.5128_{\pm0.011}$ | $0.0192$ | $0.1731$ | $0.4407_{\pm0.007}$ | $0.8349_{\pm0.007}$ | $0.6731_{\pm0.024}$ | $0.6795_{\pm0.01}$ | $0.7131_{\pm0.025}$ | $0.8285_{\pm0.01}$ | $\mathbf{0.8974}_{\pm0.02}$ |
| Plane | $\mathbf{1.0}_{\pm0.0}$ | $0.9905$ | $0.9905$ | $0.9968_{\pm0.005}$ | $\mathbf{1.0}_{\pm0.0}$ | $\mathbf{1.0}_{\pm0.0}$ | $\mathbf{1.0}_{\pm0.0}$ | $\mathbf{1.0}_{\pm0.0}$ | $\mathbf{1.0}_{\pm0.0}$ | $\mathbf{1.0}_{\pm0.0}$ |
| PowerCons | $0.9944_{\pm0.0}$ | $\mathbf{1.0}$ | $\mathbf{1.0}$ | $0.9407_{\pm0.003}$ | $0.8981_{\pm0.008}$ | $0.9407_{\pm0.006}$ | $0.8907_{\pm0.008}$ | $0.9074_{\pm0.017}$ | $0.9333_{\pm0.008}$ | $0.9648_{\pm0.003}$ |
| ProximalPhalanxOutlineAgeGroup | $0.8472_{\pm0.007}$ | $\mathbf{0.8585}$ | $0.839$ | $0.8341_{\pm0.005}$ | $0.852_{\pm0.003}$ | $0.8569_{\pm0.007}$ | $0.8537_{\pm0.013}$ | $0.8537_{\pm0.01}$ | $0.8504_{\pm0.003}$ | $0.8293_{\pm0.005}$ |
| ProximalPhalanxOutlineCorrect | $0.8648_{\pm0.012}$ | $0.9038$ | $\mathbf{0.9244}$ | $0.8603_{\pm0.01}$ | $0.8774_{\pm0.009}$ | $0.8706_{\pm0.004}$ | $0.8511_{\pm0.005}$ | $0.8328_{\pm0.008}$ | $0.8373_{\pm0.004}$ | $0.8671_{\pm0.005}$ |
| ProximalPhalanxTW | $0.8033_{\pm0.007}$ | $0.8098$ | $\mathbf{0.8293}$ | $0.8114_{\pm0.01}$ | $0.8065_{\pm0.023}$ | $0.7984_{\pm0.016}$ | $0.7772_{\pm0.011}$ | $0.7919_{\pm0.01}$ | $0.8146_{\pm0.005}$ | $0.8211_{\pm0.016}$ |
| RefrigerationDevices | $0.5493_{\pm0.023}$ | $0.504$ | $0.4933$ | $0.5493_{\pm0.01}$ | $0.568_{\pm0.0}$ | $0.5502_{\pm0.009}$ | $0.5467_{\pm0.003}$ | $\mathbf{0.5911}_{\pm0.016}$ | $0.5369_{\pm0.009}$ | $0.5636_{\pm0.006}$ |
| Rock | $0.62_{\pm0.02}$ | $0.76$ | $0.64$ | $0.8333_{\pm0.012}$ | $\mathbf{0.9}_{\pm0.04}$ | $0.8733_{\pm0.031}$ | $0.8933_{\pm0.031}$ | $\mathbf{0.9}_{\pm0.02}$ | $0.6533_{\pm0.061}$ | $0.82_{\pm0.02}$ |
| ScreenType | $0.5271_{\pm0.007}$ | $0.4187$ | $0.4107$ | $0.4284_{\pm0.017}$ | $0.5156_{\pm0.015}$ | $0.4871_{\pm0.008}$ | $\mathbf{0.5449}_{\pm0.01}$ | $0.5396_{\pm0.026}$ | $0.5058_{\pm0.012}$ | $0.4596_{\pm0.014}$ |
| SemgHandGenderCh2 | $0.9272_{\pm0.003}$ | $\mathbf{0.9467}$ | $0.8867$ | $0.7778_{\pm0.003}$ | $0.8817_{\pm0.0}$ | $0.8906_{\pm0.008}$ | $0.8394_{\pm0.003}$ | $0.8717_{\pm0.002}$ | $0.8561_{\pm0.003}$ | $0.9139_{\pm0.006}$ |
| SemgHandMovementCh2 | $\mathbf{0.8615}_{\pm0.007}$ | $0.7711$ | $0.5689$ | $0.4252_{\pm0.005}$ | $0.6489_{\pm0.016}$ | $0.6259_{\pm0.016}$ | $0.537_{\pm0.008}$ | $0.597_{\pm0.013}$ | $0.6756_{\pm0.006}$ | $0.7311_{\pm0.008}$ |
| SemgHandSubjectCh2 | $0.8837_{\pm0.008}$ | $\mathbf{0.9356}$ | $0.8333$ | $0.6504_{\pm0.006}$ | $0.8244_{\pm0.008}$ | $0.8259_{\pm0.007}$ | $0.7822_{\pm0.008}$ | $0.8237_{\pm0.008}$ | $0.7622_{\pm0.004}$ | $0.8341_{\pm0.011}$ |
| ShakeGestureWiimoteZ | $0.8467_{\pm0.031}$ | $0.82$ | $0.74$ | $0.84_{\pm0.0}$ | $0.88_{\pm0.035}$ | $0.86_{\pm0.02}$ | $0.8467_{\pm0.042}$ | $0.8267_{\pm0.012}$ | $0.9133_{\pm0.012}$ | $\mathbf{0.9333}_{\pm0.012}$ |
| ShapeletSim | $0.9704_{\pm0.008}$ | $0.4778$ | $0.5056$ | $0.9593_{\pm0.008}$ | $0.9426_{\pm0.008}$ | $\mathbf{1.0}_{\pm0.0}$ | $\mathbf{1.0}_{\pm0.0}$ | $\mathbf{1.0}_{\pm0.0}$ | $0.9204_{\pm0.012}$ | $0.9593_{\pm0.003}$ |
| ShapesAll | $0.8206_{\pm0.002}$ | $0.8017$ | $0.7917$ | $0.8322_{\pm0.006}$ | $0.8356_{\pm0.012}$ | $\mathbf{0.8839}_{\pm0.003}$ | $0.8678_{\pm0.003}$ | $0.8444_{\pm0.005}$ | $0.8394_{\pm0.004}$ | $0.87_{\pm0.009}$ |
| SmallKitchenAppliances | $0.8302_{\pm0.008}$ | $0.7867$ | $0.7627$ | $0.7111_{\pm0.007}$ | $\mathbf{0.8382}_{\pm0.002}$ | $0.8373_{\pm0.009}$ | $0.8284_{\pm0.006}$ | $0.8302_{\pm0.011}$ | $0.8196_{\pm0.004}$ | $0.8373_{\pm0.004}$ |
| SmoothSubspace | $0.9867_{\pm0.007}$ | $\mathbf{1.0}$ | $\mathbf{1.0}$ | $0.9578_{\pm0.004}$ | $0.9222_{\pm0.01}$ | $0.9289_{\pm0.023}$ | $0.9333_{\pm0.007}$ | $0.9267_{\pm0.007}$ | $0.8733_{\pm0.007}$ | $0.9511_{\pm0.004}$ |
| SonyAIBORobotSurface1 | $0.8397_{\pm0.002}$ | $0.772$ | $0.6722$ | $0.8541_{\pm0.008}$ | $\mathbf{0.8625}_{\pm0.03}$ | $0.6628_{\pm0.02}$ | $0.7876_{\pm0.01}$ | $0.7643_{\pm0.012}$ | $0.8092_{\pm0.012}$ | $0.8231_{\pm0.011}$ |
| SonyAIBORobotSurface2 | $0.8835_{\pm0.021}$ | $0.809$ | $0.8279$ | $0.8279_{\pm0.004}$ | $0.8255_{\pm0.003}$ | $0.8475_{\pm0.002}$ | $0.9038_{\pm0.002}$ | $0.9185_{\pm0.004}$ | $0.8391_{\pm0.01}$ | $\mathbf{0.922}_{\pm0.009}$ |
| StarLightCurves | $0.9702_{\pm0.001}$ | $0.9732$ | $0.9718$ | $0.9768_{\pm0.0}$ | $0.9789_{\pm0.0}$ | $0.98_{\pm0.0}$ | $0.9788_{\pm0.0}$ | $0.9803_{\pm0.0}$ | $0.979_{\pm0.0}$ | $\mathbf{0.9806}_{\pm0.0}$ |
| Strawberry | $0.9333_{\pm0.003}$ | $0.9811$ | $\mathbf{0.9838}$ | $0.9568_{\pm0.005}$ | $0.9532_{\pm0.004}$ | $0.9432_{\pm0.003}$ | $0.927_{\pm0.007}$ | $0.9414_{\pm0.002}$ | $0.936_{\pm0.006}$ | $0.9595_{\pm0.007}$ |
| SwedishLeaf | $0.9115_{\pm0.002}$ | $0.9504$ | $0.9456$ | $0.9211_{\pm0.002}$ | $0.9381_{\pm0.004}$ | $0.9381_{\pm0.004}$ | $0.9408_{\pm0.002}$ | $0.9376_{\pm0.006}$ | $0.9221_{\pm0.004}$ | $\mathbf{0.9547}_{\pm0.004}$ |
| Symbols | $0.9618_{\pm0.005}$ | $0.8824$ | $0.8945$ | $0.9377_{\pm0.002}$ | $0.937_{\pm0.004}$ | $0.9745_{\pm0.006}$ | $\mathbf{0.9779}_{\pm0.005}$ | $0.9759_{\pm0.003}$ | $0.9387_{\pm0.005}$ | $0.9698_{\pm0.005}$ |
| SyntheticControl | $0.9922_{\pm0.002}$ | $0.99$ | $0.9833$ | $0.9578_{\pm0.002}$ | $0.9867_{\pm0.0}$ | $0.9933_{\pm0.003}$ | $0.9956_{\pm0.002}$ | $\mathbf{0.9978}_{\pm0.002}$ | $0.9722_{\pm0.002}$ | $0.99_{\pm0.0}$ |
| ToeSegmentation1 | $0.8553_{\pm0.016}$ | $0.5746$ | $0.6667$ | $0.9313_{\pm0.005}$ | $0.9474_{\pm0.008}$ | $0.924_{\pm0.024}$ | $0.9415_{\pm0.003}$ | $0.8611_{\pm0.028}$ | $0.8436_{\pm0.022}$ | $\mathbf{0.9547}_{\pm0.007}$ |
| ToeSegmentation2 | $0.7846_{\pm0.013}$ | $0.6538$ | $0.8154$ | $0.8462_{\pm0.008}$ | $\mathbf{0.9026}_{\pm0.004}$ | $0.8769_{\pm0.008}$ | $0.8564_{\pm0.012}$ | $0.8513_{\pm0.016}$ | $0.7385_{\pm0.024}$ | $0.8821_{\pm0.004}$ |
| Trace | $\mathbf{1.0}_{\pm0.0}$ | $0.91$ | $0.98$ | $0.99_{\pm0.0}$ | $\mathbf{1.0}_{\pm0.0}$ | $\mathbf{1.0}_{\pm0.0}$ | $\mathbf{1.0}_{\pm0.0}$ | $\mathbf{1.0}_{\pm0.0}$ | $0.99_{\pm0.0}$ | $\mathbf{1.0}_{\pm0.0}$ |
| TwoLeadECG | $0.8584_{\pm0.015}$ | $0.9508$ | $0.9254$ | $0.9485_{\pm0.007}$ | $0.9511_{\pm0.007}$ | $0.9309_{\pm0.011}$ | $0.9936_{\pm0.001}$ | $0.983_{\pm0.002}$ | $0.9166_{\pm0.025}$ | $\mathbf{0.9956}_{\pm0.001}$ |
| TwoPatterns | $0.9935_{\pm0.001}$ | $\mathbf{0.995}$ | $0.9032$ | $0.9158_{\pm0.004}$ | $0.9813_{\pm0.002}$ | $0.908_{\pm0.006}$ | $0.9474_{\pm0.001}$ | $0.9693_{\pm0.003}$ | $0.8502_{\pm0.002}$ | $0.9819_{\pm0.001}$ |
| UMD | $0.9398_{\pm0.033}$ | $\mathbf{1.0}$ | $0.9375$ | $0.9838_{\pm0.004}$ | $0.919_{\pm0.004}$ | $0.9931_{\pm0.0}$ | $0.9861_{\pm0.0}$ | $0.9907_{\pm0.004}$ | $0.9352_{\pm0.014}$ | $0.9815_{\pm0.004}$ |
| UWaveGestureLibraryAll | $0.9454_{\pm0.001}$ | $\mathbf{0.9665}$ | $0.9629$ | $0.9162_{\pm0.002}$ | $0.9073_{\pm0.002}$ | $0.9229_{\pm0.001}$ | $0.8637_{\pm0.0}$ | $0.8702_{\pm0.003}$ | $0.8882_{\pm0.002}$ | $0.8848_{\pm0.001}$ |
| UWaveGestureLibraryX | $0.8062_{\pm0.003}$ | $0.8079$ | $0.7984$ | $0.7938_{\pm0.002}$ | $0.7824_{\pm0.003}$ | $0.8036_{\pm0.001}$ | $0.7948_{\pm0.003}$ | $0.7992_{\pm0.001}$ | $0.8125_{\pm0.002}$ | $\mathbf{0.813}_{\pm0.001}$ |
| UWaveGestureLibraryY | $0.7248_{\pm0.002}$ | $0.715$ | $0.7164$ | $0.7139_{\pm0.002}$ | $0.7236_{\pm0.001}$ | $0.7344_{\pm0.004}$ | $0.739_{\pm0.002}$ | $\mathbf{0.7633}_{\pm0.001}$ | $0.7391_{\pm0.003}$ | $0.7572_{\pm0.002}$ |
| UWaveGestureLibraryZ | $0.7519_{\pm0.002}$ | $0.7513$ | $0.7379$ | $0.7339_{\pm0.004}$ | $0.7361_{\pm0.001}$ | $0.7464_{\pm0.004}$ | $0.743_{\pm0.006}$ | $0.7562_{\pm0.005}$ | $0.7518_{\pm0.005}$ | $\mathbf{0.7587}_{\pm0.0}$ |
| Wafer | $0.9988_{\pm0.0}$ | $0.9953$ | $0.9959$ | $0.9878_{\pm0.001}$ | $0.9986_{\pm0.0}$ | $0.985_{\pm0.001}$ | $0.9953_{\pm0.0}$ | $\mathbf{0.9993}_{\pm0.0}$ | $0.9935_{\pm0.001}$ | $0.9944_{\pm0.001}$ |
| Wine | $0.6358_{\pm0.039}$ | $0.7778$ | $0.7222$ | $0.7531_{\pm0.021}$ | $0.6296_{\pm0.049}$ | $\mathbf{0.8704}_{\pm0.019}$ | $0.5123_{\pm0.021}$ | $0.6605_{\pm0.028}$ | $0.7099_{\pm0.039}$ | $0.8086_{\pm0.06}$ |
| WordSynonyms | $0.639_{\pm0.008}$ | $\mathbf{0.6442}$ | $0.5972$ | $0.6186_{\pm0.01}$ | $0.5204_{\pm0.005}$ | $0.5711_{\pm0.001}$ | $0.5674_{\pm0.007}$ | $0.5569_{\pm0.008}$ | $0.5303_{\pm0.014}$ | $0.6181_{\pm0.006}$ |
| Worms | $0.7229_{\pm0.015}$ | $0.5714$ | $0.5584$ | $0.7359_{\pm0.015}$ | $0.8009_{\pm0.007}$ | $0.7662_{\pm0.0}$ | $\mathbf{0.8312}_{\pm0.013}$ | $0.8009_{\pm0.02}$ | $0.7229_{\pm0.02}$ | $0.7576_{\pm0.02}$ |
| WormsTwoClass | $0.8182_{\pm0.005}$ | $0.6104$ | $0.5714$ | $0.8009_{\pm0.007}$ | $\mathbf{0.8485}_{\pm0.015}$ | $0.8139_{\pm0.027}$ | $0.8355_{\pm0.02}$ | $0.8225_{\pm0.015}$ | $0.7835_{\pm0.007}$ | $0.8182_{\pm0.013}$ |
| Yoga | $0.8289_{\pm0.005}$ | $0.8603$ | $\mathbf{0.8653}$ | $0.8369_{\pm0.006}$ | $0.7677_{\pm0.002}$ | $0.8162_{\pm0.003}$ | $0.8091_{\pm0.003}$ | $0.8144_{\pm0.002}$ | $0.8207_{\pm0.002}$ | $0.848_{\pm0.003}$ |
| *Average* | $0.7969$ | $0.7806$ | $0.7707$ | $0.7789$ | $0.8013$ | $0.8002$ | $0.7943$ | $0.8029$ | $0.7732$ | $\mathbf{0.8195}$ |
| *Best Counts* | $14$ | $\mathbf{30}$ | $24$ | $2$ | $12$ | $15$ | $14$ | $21$ | $7$ | $28$ |

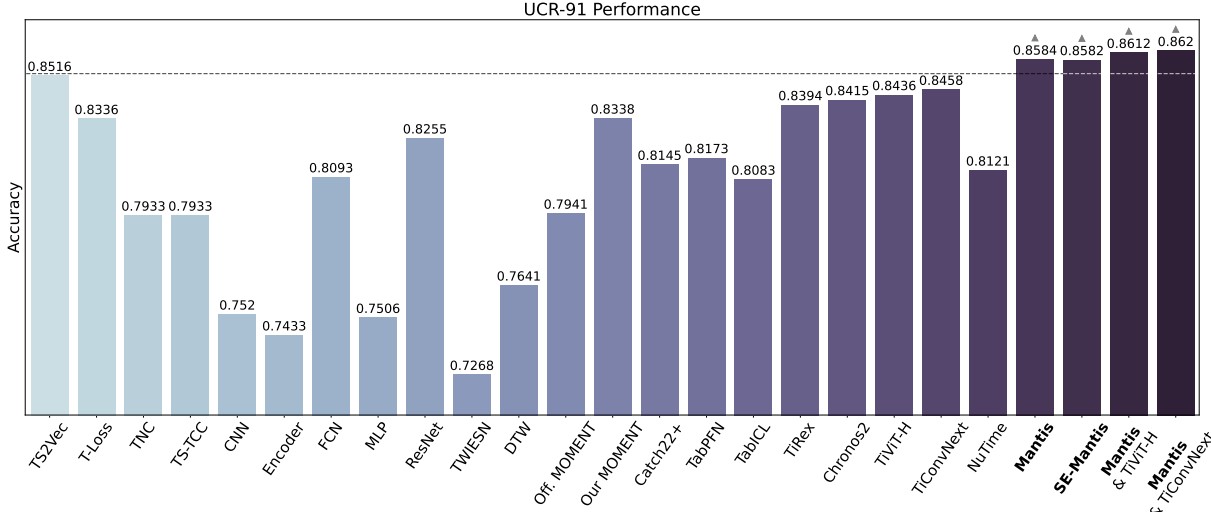

*Figure 19.* Average performance over 91 datasets from UCR.

*Table 20.* The performance comparison between Mantis in the zero-shot feature extraction setting in average over UEA-27 datasets.

| | Catch22+ | TabPFN | TabICL | MOMENT | TiRex | Chronos2 | TiViT-H | TiConvNext | NuTime | Mantis |
|---|---|---|---|---|---|---|---|---|---|---|
| ArticularyWordRecognition | $0.96_{\pm0.007}$ | 0.93 | 0.91 | $0.9844_{\pm0.002}$ | $0.99_{\pm0.0}$ | $\mathbf{0.9933}_{\pm0.003}$ | $0.9833_{\pm0.003}$ | $0.98_{\pm0.003}$ | $0.9911_{\pm0.004}$ | $0.9922_{\pm0.002}$ |
| BasicMotions | $\mathbf{1.0}_{\pm0.0}$ | 1.0 | 0.95 | $0.975_{\pm0.0}$ | $0.9667_{\pm0.014}$ | $\mathbf{1.0}_{\pm0.0}$ | $0.9833_{\pm0.014}$ | $\mathbf{1.0}_{\pm0.0}$ | $\mathbf{1.0}_{\pm0.0}$ | $\mathbf{1.0}_{\pm0.0}$ |
| CharacterTrajectories | $\mathbf{0.9814}_{\pm0.001}$ | 0.9673 | 0.9694 | $0.9698_{\pm0.002}$ | $0.9603_{\pm0.0}$ | $0.9631_{\pm0.001}$ | $0.9373_{\pm0.002}$ | $0.9517_{\pm0.002}$ | $0.967_{\pm0.002}$ | $0.9761_{\pm0.0}$ |
| Cricket | $0.9444_{\pm0.014}$ | 0.8194 | 0.8472 | $0.9722_{\pm0.0}$ | $0.9954_{\pm0.008}$ | $0.9861_{\pm0.0}$ | $\mathbf{1.0}_{\pm0.0}$ | $0.9954_{\pm0.008}$ | $0.9907_{\pm0.008}$ | $0.9861_{\pm0.0}$ |
| DuckDuckGeese | $0.5133_{\pm0.023}$ | 0.32 | 0.28 | $0.4467_{\pm0.042}$ | $0.42_{\pm0.02}$ | $0.4133_{\pm0.023}$ | $0.4333_{\pm0.023}$ | $0.44_{\pm0.04}$ | $0.4267_{\pm0.023}$ | $\mathbf{0.54}_{\pm0.02}$ |
| ERing | $0.8827_{\pm0.022}$ | 0.8889 | 0.8519 | $0.9679_{\pm0.008}$ | $0.9765_{\pm0.008}$ | $\mathbf{0.9877}_{\pm0.004}$ | $0.9815_{\pm0.004}$ | $0.963_{\pm0.007}$ | $0.9765_{\pm0.004}$ | $\mathbf{0.9877}_{\pm0.002}$ |
| EigenWorms | $0.8753_{\pm0.019}$ | 0.4198 | 0.5649 | $0.7608_{\pm0.004}$ | $0.7863_{\pm0.038}$ | $0.8015_{\pm0.015}$ | $0.888_{\pm0.027}$ | $\mathbf{0.9313}_{\pm0.008}$ | $0.7354_{\pm0.022}$ | $0.8142_{\pm0.016}$ |
| Epilepsy | $0.9855_{\pm0.0}$ | 0.913 | 0.9565 | $0.9928_{\pm0.0}$ | $\mathbf{1.0}_{\pm0.0}$ | $\mathbf{1.0}_{\pm0.0}$ | $\mathbf{1.0}_{\pm0.0}$ | $\mathbf{1.0}_{\pm0.0}$ | $\mathbf{1.0}_{\pm0.0}$ | $0.9976_{\pm0.004}$ |
| EthanolConcentration | $0.4271_{\pm0.002}$ | $\mathbf{0.7833}$ | 0.6046 | $0.2953_{\pm0.019}$ | $0.275_{\pm0.006}$ | $0.3485_{\pm0.01}$ | $0.3866_{\pm0.008}$ | $0.3663_{\pm0.012}$ | $0.4132_{\pm0.01}$ | $0.4183_{\pm0.007}$ |
| FaceDetection | $0.5273_{\pm0.0}$ | 0.6325 | $\mathbf{0.6402}$ | $0.554_{\pm0.005}$ | $0.6133_{\pm0.006}$ | $0.5716_{\pm0.011}$ | $0.55_{\pm0.005}$ | $0.5321_{\pm0.003}$ | $0.5691_{\pm0.004}$ | $0.5492_{\pm0.004}$ |
| FingerMovements | $0.4967_{\pm0.064}$ | 0.5 | 0.49 | $0.53_{\pm0.02}$ | $0.52_{\pm0.04}$ | $0.5233_{\pm0.0}$ | $0.5367_{\pm0.045}$ | $0.5333_{\pm0.005}$ | $0.5267_{\pm0.015}$ | $\mathbf{0.55}_{\pm0.01}$ |
| HandMovementDirection | $0.3378_{\pm0.049}$ | $\mathbf{0.4324}$ | 0.3919 | $0.2928_{\pm0.021}$ | $0.3153_{\pm0.028}$ | $0.2748_{\pm0.008}$ | $0.2883_{\pm0.031}$ | $0.3288_{\pm0.055}$ | $0.2883_{\pm0.021}$ | $0.2793_{\pm0.067}$ |
| Handwriting | $\mathbf{0.2812}_{\pm0.009}$ | 0.1388 | 0.2259 | $0.26_{\pm0.002}$ | $0.2525_{\pm0.002}$ | $0.2576_{\pm0.007}$ | $0.2373_{\pm0.009}$ | $0.2494_{\pm0.011}$ | $0.2055_{\pm0.007}$ | $0.2808_{\pm0.008}$ |
| Heartbeat | $0.7496_{\pm0.006}$ | 0.722 | 0.7268 | $0.7317_{\pm0.005}$ | $0.7252_{\pm0.006}$ | $0.7317_{\pm0.008}$ | $0.7252_{\pm0.003}$ | $0.735_{\pm0.007}$ | $0.7756_{\pm0.005}$ | $\mathbf{0.7919}_{\pm0.003}$ |
| InsectWingbeatSubset | $0.3617_{\pm0.01}$ | 0.238 | nan | $0.267_{\pm0.017}$ | $0.287_{\pm0.004}$ | $0.2803_{\pm0.018}$ | $0.3203_{\pm0.014}$ | $0.3243_{\pm0.0}$ | $\mathbf{0.6143}_{\pm0.012}$ | $0.6073_{\pm0.007}$ |
| JapaneseVowels | $\mathbf{0.955}_{\pm0.003}$ | 0.8135 | 0.8162 | $0.8847_{\pm0.006}$ | $0.8928_{\pm0.015}$ | $0.8162_{\pm0.026}$ | $0.882_{\pm0.006}$ | $0.8721_{\pm0.004}$ | $0.9342_{\pm0.006}$ | $0.9396_{\pm0.006}$ |
| LSST | $0.6158_{\pm0.002}$ | 0.5479 | 0.5016 | $0.6313_{\pm0.003}$ | $0.5733_{\pm0.003}$ | $0.5892_{\pm0.004}$ | $0.5953_{\pm0.006}$ | $0.6004_{\pm0.004}$ | $0.5673_{\pm0.005}$ | $\mathbf{0.6599}_{\pm0.004}$ |
| Libras | $0.8389_{\pm0.01}$ | 0.6722 | 0.6389 | $0.8463_{\pm0.008}$ | $0.8963_{\pm0.008}$ | $0.8963_{\pm0.008}$ | $0.9037_{\pm0.006}$ | $0.9074_{\pm0.008}$ | $0.8778_{\pm0.01}$ | $\mathbf{0.9241}_{\pm0.003}$ |
| MotorImagery | $0.4333_{\pm0.086}$ | $\mathbf{0.58}$ | 0.57 | $0.5233_{\pm0.015}$ | $0.4933_{\pm0.038}$ | $0.53_{\pm0.026}$ | $0.51_{\pm0.01}$ | $0.49_{\pm0.036}$ | $0.48_{\pm0.026}$ | $0.4667_{\pm0.025}$ |
| NATOPS | $0.7148_{\pm0.029}$ | 0.7778 | 0.7944 | $0.8352_{\pm0.012}$ | $0.8296_{\pm0.012}$ | $0.8389_{\pm0.011}$ | $0.8685_{\pm0.017}$ | $0.8574_{\pm0.014}$ | $0.8333_{\pm0.031}$ | $\mathbf{0.8889}_{\pm0.01}$ |
| PEMS-SF | $0.8401_{\pm0.009}$ | 0.948 | 0.9422 | $\mathbf{0.9961}_{\pm0.007}$ | $\mathbf{0.9961}_{\pm0.007}$ | $\mathbf{0.9961}_{\pm0.007}$ | $0.973_{\pm0.007}$ | $0.9769_{\pm0.0}$ | $0.9923_{\pm0.007}$ | $\mathbf{0.9961}_{\pm0.007}$ |
| PhonemeSpectra | $0.2499_{\pm0.006}$ | 0.1485 | 0.1712 | $0.2112_{\pm0.006}$ | $0.2686_{\pm0.003}$ | $0.2709_{\pm0.003}$ | $0.2713_{\pm0.005}$ | $0.2709_{\pm0.003}$ | $0.2664_{\pm0.005}$ | $\mathbf{0.3213}_{\pm0.004}$ |
| RacketSports | $0.8004_{\pm0.004}$ | 0.8158 | 0.8487 | $0.8465_{\pm0.025}$ | $0.8355_{\pm0.007}$ | $0.8246_{\pm0.03}$ | $0.8531_{\pm0.01}$ | $0.8509_{\pm0.004}$ | $\mathbf{0.9123}_{\pm0.019}$ | $0.9101_{\pm0.008}$ |
| SelfRegulationSCP1 | $0.7702_{\pm0.007}$ | $\mathbf{0.8942}$ | 0.8874 | $0.7747_{\pm0.006}$ | $0.7884_{\pm0.012}$ | $0.785_{\pm0.003}$ | $0.7986_{\pm0.007}$ | $0.7929_{\pm0.01}$ | $0.7952_{\pm0.003}$ | $0.8134_{\pm0.009}$ |
| SelfRegulationSCP2 | $0.4926_{\pm0.031}$ | 0.4778 | 0.5056 | $0.4907_{\pm0.023}$ | $0.4963_{\pm0.049}$ | $\mathbf{0.5167}_{\pm0.006}$ | $0.4907_{\pm0.033}$ | $0.5056_{\pm0.011}$ | $0.5074_{\pm0.018}$ | $\mathbf{0.5167}_{\pm0.006}$ |
| SpokenArabicDigits | $0.8898_{\pm0.005}$ | 0.9591 | $\mathbf{0.9613}$ | $0.9447_{\pm0.003}$ | $0.7547_{\pm0.01}$ | $0.7619_{\pm0.006}$ | $0.8931_{\pm0.005}$ | $0.9139_{\pm0.004}$ | $0.9016_{\pm0.002}$ | $0.9374_{\pm0.003}$ |
| UWaveGestureLibrary | $0.876_{\pm0.015}$ | 0.8062 | 0.7625 | $\mathbf{0.8917}_{\pm0.007}$ | $0.8615_{\pm0.002}$ | $0.8833_{\pm0.004}$ | $0.8542_{\pm0.004}$ | $0.8156_{\pm0.005}$ | $0.8885_{\pm0.007}$ | $0.8896_{\pm0.011}$ |
| *Avg* | 0.6963 | 0.6721 | 0.685 | 0.6991 | 0.6952 | 0.6967 | 0.7091 | 0.7105 | 0.7199 | **0.742** |
| *Best Counts* | 4 | 5 | 2 | 2 | 2 | 6 | 2 | 3 | 3 | **11** |

*Table 21.* Self-ensembling and cross-model embedding fusion on UCR. The first part of the table.

| | Mantis | SE-Mantis | Mantis & Catch22+ | Mantis & MOMENT | Mantis & TiRex | Mantis & Chronos2 | Mantis & TiViT-H | Mantis & TiConvNext | Mantis & NuTime |
|---|---|---|---|---|---|---|---|---|---|
| ACSF1 | $0.8233_{\pm0.012}$ | $0.8367_{\pm0.025}$ | $0.8433_{\pm0.015}$ | $0.8367_{\pm0.015}$ | $0.82_{\pm0.01}$ | $0.8333_{\pm0.021}$ | $0.8567_{\pm0.015}$ | $\mathbf{0.88}_{\pm0.01}$ | $0.83_{\pm0.01}$ |
| Adiac | $\mathbf{0.8483}_{\pm0.003}$ | $0.8457_{\pm0.004}$ | $0.8406_{\pm0.006}$ | $0.8329_{\pm0.003}$ | $0.8295_{\pm0.01}$ | $0.838_{\pm0.01}$ | $0.8278_{\pm0.01}$ | $0.8082_{\pm0.003}$ | $0.8389_{\pm0.005}$ |
| AllGestureWiimoteX | $0.6843_{\pm0.006}$ | $0.639_{\pm0.003}$ | $0.6762_{\pm0.008}$ | $0.6771_{\pm0.012}$ | $0.6824_{\pm0.008}$ | $0.6562_{\pm0.016}$ | $0.6624_{\pm0.002}$ | $0.6838_{\pm0.005}$ | $\mathbf{0.6986}_{\pm0.008}$ |
| AllGestureWiimoteY | $0.691_{\pm0.004}$ | $0.6795_{\pm0.002}$ | $0.6948_{\pm0.009}$ | $0.7019_{\pm0.009}$ | $0.7048_{\pm0.007}$ | $0.6871_{\pm0.011}$ | $0.6862_{\pm0.002}$ | $\mathbf{0.7062}_{\pm0.007}$ | $0.6871_{\pm0.002}$ |
| AllGestureWiimoteZ | $0.6881_{\pm0.001}$ | $0.6481_{\pm0.015}$ | $0.6814_{\pm0.011}$ | $0.659_{\pm0.005}$ | $0.659_{\pm0.014}$ | $0.6848_{\pm0.002}$ | $0.6562_{\pm0.004}$ | $0.6567_{\pm0.008}$ | $\mathbf{0.6948}_{\pm0.011}$ |
| ArrowHead | $0.8057_{\pm0.015}$ | $0.7752_{\pm0.017}$ | $0.8305_{\pm0.009}$ | $\mathbf{0.8438}_{\pm0.003}$ | $0.8248_{\pm0.003}$ | $0.8248_{\pm0.009}$ | $0.7981_{\pm0.013}$ | $0.7943_{\pm0.0}$ | $0.8171_{\pm0.015}$ |
| BME | $0.9956_{\pm0.004}$ | $\mathbf{1.0}_{\pm0.0}$ | $\mathbf{1.0}_{\pm0.0}$ | $0.9956_{\pm0.008}$ | $0.9978_{\pm0.004}$ | $0.9889_{\pm0.004}$ | $0.9911_{\pm0.004}$ | $0.9978_{\pm0.004}$ | $0.9978_{\pm0.004}$ |
| Beef | $0.6889_{\pm0.019}$ | $0.6667_{\pm0.0}$ | $0.6889_{\pm0.019}$ | $\mathbf{0.8111}_{\pm0.051}$ | $0.7778_{\pm0.019}$ | $0.7333_{\pm0.033}$ | $0.7333_{\pm0.033}$ | $0.7778_{\pm0.038}$ | $0.6667_{\pm0.033}$ |
| BeetleFly | $\mathbf{0.9833}_{\pm0.029}$ | $0.95_{\pm0.0}$ | $0.95_{\pm0.05}$ | $0.95_{\pm0.0}$ | $0.9333_{\pm0.029}$ | $0.95_{\pm0.0}$ | $0.9333_{\pm0.029}$ | $0.95_{\pm0.0}$ | $0.95_{\pm0.05}$ |
| BirdChicken | $0.9_{\pm0.0}$ | $0.9_{\pm0.0}$ | $0.9_{\pm0.0}$ | $0.9_{\pm0.0}$ | $0.9_{\pm0.0}$ | $0.9_{\pm0.0}$ | $0.9_{\pm0.0}$ | $\mathbf{0.9333}_{\pm0.029}$ | $0.9_{\pm0.0}$ |
| CBF | $0.9948_{\pm0.002}$ | $0.9956_{\pm0.002}$ | $0.9963_{\pm0.002}$ | $0.9911_{\pm0.001}$ | $0.997_{\pm0.001}$ | $0.9963_{\pm0.001}$ | $\mathbf{0.9985}_{\pm0.001}$ | $0.9974_{\pm0.001}$ | $0.9959_{\pm0.001}$ |
| Car | $0.7556_{\pm0.025}$ | $\mathbf{0.8222}_{\pm0.019}$ | $0.75_{\pm0.029}$ | $0.8111_{\pm0.01}$ | $0.7444_{\pm0.01}$ | $0.75_{\pm0.029}$ | $0.7667_{\pm0.0}$ | $0.7444_{\pm0.025}$ | $0.75_{\pm0.017}$ |
| Chinatown | $0.9534_{\pm0.003}$ | $0.9543_{\pm0.009}$ | $0.9728_{\pm0.002}$ | $\mathbf{0.9776}_{\pm0.002}$ | $0.966_{\pm0.003}$ | $0.9728_{\pm0.004}$ | $0.9592_{\pm0.013}$ | $0.9485_{\pm0.004}$ | $0.9582_{\pm0.002}$ |
| ChlorineConcentration | $0.7041_{\pm0.001}$ | $0.7129_{\pm0.003}$ | $0.7095_{\pm0.003}$ | $0.7152_{\pm0.004}$ | $0.7155_{\pm0.002}$ | $\mathbf{0.7168}_{\pm0.007}$ | $0.7083_{\pm0.002}$ | $0.7133_{\pm0.001}$ | $0.7104_{\pm0.005}$ |
| CinCECGTorso | $0.8089_{\pm0.01}$ | $0.8703_{\pm0.014}$ | $0.8227_{\pm0.012}$ | $0.7756_{\pm0.011}$ | $\mathbf{0.8766}_{\pm0.009}$ | $0.8082_{\pm0.012}$ | $0.8374_{\pm0.007}$ | $0.8428_{\pm0.003}$ | $0.8024_{\pm0.007}$ |
| Coffee | $\mathbf{1.0}_{\pm0.0}$ | $0.9881_{\pm0.021}$ | $\mathbf{1.0}_{\pm0.0}$ | $0.9762_{\pm0.021}$ | $\mathbf{1.0}_{\pm0.0}$ | $0.9881_{\pm0.021}$ | $0.9881_{\pm0.021}$ | $\mathbf{1.0}_{\pm0.0}$ | $0.9881_{\pm0.021}$ |
| Computers | $0.7213_{\pm0.012}$ | $0.7147_{\pm0.008}$ | $0.7173_{\pm0.01}$ | $0.7213_{\pm0.009}$ | $\mathbf{0.752}_{\pm0.004}$ | $0.748_{\pm0.004}$ | $0.7373_{\pm0.002}$ | $\mathbf{0.752}_{\pm0.008}$ | $0.736_{\pm0.007}$ |
| CricketX | $\mathbf{0.7735}_{\pm0.011}$ | $0.7692_{\pm0.007}$ | $0.7658_{\pm0.007}$ | $0.7444_{\pm0.015}$ | $0.7547_{\pm0.013}$ | $0.7701_{\pm0.01}$ | $0.759_{\pm0.005}$ | $0.747_{\pm0.015}$ | $0.765_{\pm0.006}$ |
| CricketY | $0.788_{\pm0.02}$ | $\mathbf{0.8051}_{\pm0.011}$ | $0.7829_{\pm0.014}$ | $0.753_{\pm0.019}$ | $0.7846_{\pm0.004}$ | $0.7872_{\pm0.013}$ | $0.7769_{\pm0.009}$ | $0.7581_{\pm0.001}$ | $0.7932_{\pm0.005}$ |
| CricketZ | $\mathbf{0.8171}_{\pm0.005}$ | $0.8103_{\pm0.014}$ | $0.8154_{\pm0.007}$ | $0.7957_{\pm0.015}$ | $0.8043_{\pm0.005}$ | $0.7974_{\pm0.01}$ | $0.7983_{\pm0.013}$ | $0.7889_{\pm0.006}$ | $0.8085_{\pm0.008}$ |
| Crop | $0.7341_{\pm0.0}$ | $0.7384_{\pm0.001}$ | $\mathbf{0.7492}_{\pm0.002}$ | $0.7489_{\pm0.0}$ | $0.7488_{\pm0.001}$ | $0.748_{\pm0.001}$ | $0.7254_{\pm0.001}$ | $0.7187_{\pm0.003}$ | $0.7327_{\pm0.001}$ |
| DiatomSizeReduction | $0.8301_{\pm0.02}$ | $0.8148_{\pm0.008}$ | $0.8529_{\pm0.02}$ | $0.8736_{\pm0.016}$ | $0.8508_{\pm0.011}$ | $0.8682_{\pm0.013}$ | $0.8529_{\pm0.003}$ | $\mathbf{0.8791}_{\pm0.017}$ | $0.8388_{\pm0.001}$ |
| DistalPhalanxOutlineAgeGroup | $0.7602_{\pm0.008}$ | $\mathbf{0.7986}_{\pm0.007}$ | $0.7746_{\pm0.004}$ | $0.7698_{\pm0.007}$ | $0.7698_{\pm0.0}$ | $0.777_{\pm0.007}$ | $0.7842_{\pm0.007}$ | $0.7698_{\pm0.007}$ | $0.777_{\pm0.007}$ |
| DistalPhalanxOutlineCorrect | $0.7886_{\pm0.002}$ | $0.7911_{\pm0.008}$ | $0.7947_{\pm0.008}$ | $\mathbf{0.8043}_{\pm0.004}$ | $0.7886_{\pm0.006}$ | $0.7838_{\pm0.002}$ | $0.7923_{\pm0.009}$ | $0.7874_{\pm0.008}$ | $0.7935_{\pm0.006}$ |
| DistalPhalanxTW | $\mathbf{0.7074}_{\pm0.011}$ | $0.705_{\pm0.007}$ | $0.7002_{\pm0.018}$ | $0.6859_{\pm0.011}$ | $0.6882_{\pm0.011}$ | $0.6882_{\pm0.017}$ | $0.6906_{\pm0.025}$ | $0.6978_{\pm0.012}$ | $0.705_{\pm0.007}$ |
| DodgerLoopDay | $0.5333_{\pm0.019}$ | $0.5708_{\pm0.029}$ | $0.5875_{\pm0.013}$ | $0.525_{\pm0.022}$ | $0.5667_{\pm0.026}$ | $0.5458_{\pm0.026}$ | $0.5333_{\pm0.014}$ | $0.5208_{\pm0.014}$ | $\mathbf{0.5917}_{\pm0.025}$ |
| DodgerLoopGame | $0.7995_{\pm0.022}$ | $0.8406_{\pm0.019}$ | $0.8454_{\pm0.008}$ | $0.8333_{\pm0.007}$ | $0.8309_{\pm0.027}$ | $\mathbf{0.8696}_{\pm0.007}$ | $0.8502_{\pm0.011}$ | $0.814_{\pm0.017}$ | $0.843_{\pm0.025}$ |
| DodgerLoopWeekend | $0.9517_{\pm0.004}$ | $0.9638_{\pm0.007}$ | $0.9614_{\pm0.008}$ | $0.9783_{\pm0.007}$ | $0.9565_{\pm0.007}$ | $0.9541_{\pm0.004}$ | $0.9589_{\pm0.004}$ | $0.9324_{\pm0.011}$ | $\mathbf{0.9807}_{\pm0.004}$ |
| ECG200 | $0.8367_{\pm0.012}$ | $0.8367_{\pm0.021}$ | $0.84_{\pm0.0}$ | $\mathbf{0.87}_{\pm0.0}$ | $0.8533_{\pm0.006}$ | $0.8567_{\pm0.012}$ | $0.82_{\pm0.01}$ | $0.82_{\pm0.0}$ | $0.8267_{\pm0.015}$ |
| ECG5000 | $0.9381_{\pm0.0}$ | $\mathbf{0.9401}_{\pm0.0}$ | $0.9389_{\pm0.001}$ | $0.9397_{\pm0.001}$ | $0.9388_{\pm0.0}$ | $0.937_{\pm0.001}$ | $0.9391_{\pm0.001}$ | $0.9382_{\pm0.0}$ | $0.9381_{\pm0.001}$ |
| ECGFiveDays | $0.8707_{\pm0.001}$ | $0.9299_{\pm0.003}$ | $0.8788_{\pm0.007}$ | $0.8637_{\pm0.005}$ | $0.9094_{\pm0.025}$ | $0.9237_{\pm0.038}$ | $0.9129_{\pm0.017}$ | $\mathbf{0.9725}_{\pm0.006}$ | $0.8614_{\pm0.016}$ |
| EOGHorizontalSignal | $0.5884_{\pm0.006}$ | $0.6114_{\pm0.004}$ | $0.5902_{\pm0.01}$ | $0.6077_{\pm0.005}$ | $0.5884_{\pm0.005}$ | $0.5801_{\pm0.025}$ | $0.6041_{\pm0.009}$ | $\mathbf{0.6243}_{\pm0.01}$ | $0.5967_{\pm0.006}$ |
| EOGVerticalSignal | $0.4816_{\pm0.011}$ | $0.4788_{\pm0.016}$ | $\mathbf{0.4871}_{\pm0.004}$ | $0.4586_{\pm0.005}$ | $0.4687_{\pm0.02}$ | $0.4595_{\pm0.009}$ | $0.4613_{\pm0.012}$ | $0.4567_{\pm0.009}$ | $0.4678_{\pm0.014}$ |
| Earthquakes | $0.7434_{\pm0.004}$ | $0.7482_{\pm0.0}$ | $0.7458_{\pm0.004}$ | $\mathbf{0.753}_{\pm0.004}$ | $0.7458_{\pm0.004}$ | $0.7482_{\pm0.0}$ | $0.7458_{\pm0.004}$ | $0.741_{\pm0.0}$ | $0.7482_{\pm0.0}$ |
| ElectricDevices | $0.7424_{\pm0.004}$ | $0.7433_{\pm0.001}$ | $0.7491_{\pm0.003}$ | $0.7389_{\pm0.002}$ | $0.7388_{\pm0.004}$ | $0.7505_{\pm0.002}$ | $0.7641_{\pm0.002}$ | $\mathbf{0.7723}_{\pm0.006}$ | $0.7435_{\pm0.005}$ |
| EthanolLevel | $0.3793_{\pm0.014}$ | $0.3373_{\pm0.005}$ | $0.3687_{\pm0.005}$ | $0.4_{\pm0.005}$ | $0.336_{\pm0.011}$ | $0.4187_{\pm0.009}$ | $\mathbf{0.4207}_{\pm0.009}$ | $0.358_{\pm0.016}$ | $0.3653_{\pm0.012}$ |
| FaceAll | $0.7205_{\pm0.002}$ | $0.7178_{\pm0.002}$ | $0.7316_{\pm0.003}$ | $0.7312_{\pm0.002}$ | $\mathbf{0.7351}_{\pm0.003}$ | $0.7243_{\pm0.002}$ | $0.7211_{\pm0.001}$ | $0.7195_{\pm0.003}$ | $0.7209_{\pm0.003}$ |
| FaceFour | $\mathbf{0.9167}_{\pm0.007}$ | $0.8864_{\pm0.02}$ | $0.9129_{\pm0.013}$ | $0.8712_{\pm0.017}$ | $0.9053_{\pm0.033}$ | $0.8977_{\pm0.02}$ | $0.8939_{\pm0.007}$ | $0.8598_{\pm0.017}$ | $0.9053_{\pm0.017}$ |
| FacesUCR | $0.8439_{\pm0.006}$ | $0.8382_{\pm0.005}$ | $\mathbf{0.8636}_{\pm0.004}$ | $0.8439_{\pm0.0}$ | $0.8564_{\pm0.007}$ | $0.838_{\pm0.004}$ | $0.8343_{\pm0.004}$ | $0.8324_{\pm0.002}$ | $0.8398_{\pm0.005}$ |
| FiftyWords | $0.7106_{\pm0.001}$ | $0.7172_{\pm0.011}$ | $0.7304_{\pm0.007}$ | $\mathbf{0.7385}_{\pm0.006}$ | $0.7033_{\pm0.0}$ | $0.704_{\pm0.005}$ | $0.6967_{\pm0.006}$ | $0.7077_{\pm0.01}$ | $0.7077_{\pm0.012}$ |
| Fish | $0.9314_{\pm0.0}$ | $0.9219_{\pm0.003}$ | $0.9276_{\pm0.009}$ | $0.9162_{\pm0.009}$ | $0.9276_{\pm0.003}$ | $0.9333_{\pm0.009}$ | $0.9314_{\pm0.006}$ | $0.9295_{\pm0.003}$ | $\mathbf{0.941}_{\pm0.007}$ |
| FordA | $0.9303_{\pm0.0}$ | $0.9442_{\pm0.004}$ | $0.9333_{\pm0.001}$ | $0.9245_{\pm0.003}$ | $\mathbf{0.9482}_{\pm0.002}$ | $0.9338_{\pm0.002}$ | $0.9265_{\pm0.001}$ | $0.9247_{\pm0.002}$ | $0.9301_{\pm0.004}$ |
| FordB | $0.7979_{\pm0.003}$ | $0.8136_{\pm0.001}$ | $0.7992_{\pm0.004}$ | $0.7942_{\pm0.006}$ | $\mathbf{0.8235}_{\pm0.002}$ | $0.7996_{\pm0.004}$ | $0.8021_{\pm0.005}$ | $0.7856_{\pm0.003}$ | $0.7988_{\pm0.006}$ |
| FreezerRegularTrain | $0.9857_{\pm0.001}$ | $0.987_{\pm0.003}$ | $\mathbf{0.9942}_{\pm0.001}$ | $0.9808_{\pm0.003}$ | $0.987_{\pm0.002}$ | $0.9863_{\pm0.001}$ | $0.988_{\pm0.001}$ | $0.9926_{\pm0.0}$ | $0.9863_{\pm0.002}$ |
| FreezerSmallTrain | $0.935_{\pm0.004}$ | $0.963_{\pm0.013}$ | $0.954_{\pm0.003}$ | $0.8633_{\pm0.006}$ | $0.9157_{\pm0.009}$ | $0.9338_{\pm0.002}$ | $0.938_{\pm0.004}$ | $\mathbf{0.9637}_{\pm0.003}$ | $0.9413_{\pm0.007}$ |
| Fungi | $0.9427_{\pm0.028}$ | $0.9659_{\pm0.016}$ | $0.9355_{\pm0.023}$ | $\mathbf{0.9964}_{\pm0.003}$ | $0.9552_{\pm0.011}$ | $0.9642_{\pm0.016}$ | $0.9283_{\pm0.019}$ | $0.9301_{\pm0.005}$ | $0.9014_{\pm0.006}$ |
| GestureMidAirD1 | $0.7128_{\pm0.018}$ | $\mathbf{0.7564}_{\pm0.004}$ | $0.7308_{\pm0.013}$ | $0.7256_{\pm0.009}$ | $0.7282_{\pm0.004}$ | $0.6744_{\pm0.027}$ | $0.7385_{\pm0.013}$ | $0.7359_{\pm0.018}$ | $0.7256_{\pm0.019}$ |
| GestureMidAirD2 | $0.6154_{\pm0.013}$ | $0.6308_{\pm0.027}$ | $0.6231_{\pm0.008}$ | $0.6205_{\pm0.016}$ | $0.659_{\pm0.039}$ | $\mathbf{0.7154}_{\pm0.027}$ | $0.7_{\pm0.013}$ | $0.6795_{\pm0.025}$ | $0.6308_{\pm0.008}$ |
| GestureMidAirD3 | $0.4385_{\pm0.013}$ | $0.4667_{\pm0.012}$ | $0.4436_{\pm0.004}$ | $0.4256_{\pm0.012}$ | $0.4282_{\pm0.016}$ | $0.4769_{\pm0.02}$ | $0.4769_{\pm0.02}$ | $\mathbf{0.4846}_{\pm0.008}$ | $0.4333_{\pm0.016}$ |
| GesturePebbleZ1 | $0.9205_{\pm0.007}$ | $0.9205_{\pm0.013}$ | $0.9225_{\pm0.003}$ | $0.9244_{\pm0.0}$ | $\mathbf{0.9264}_{\pm0.003}$ | $0.9244_{\pm0.0}$ | $0.9225_{\pm0.003}$ | $\mathbf{0.9264}_{\pm0.003}$ | $0.9244_{\pm0.0}$ |
| GesturePebbleZ2 | $0.9114_{\pm0.011}$ | $0.9093_{\pm0.013}$ | $0.9072_{\pm0.004}$ | $0.9156_{\pm0.019}$ | $0.9093_{\pm0.007}$ | $0.903_{\pm0.016}$ | $\mathbf{0.9283}_{\pm0.01}$ | $0.9072_{\pm0.016}$ | $0.9156_{\pm0.007}$ |
| GunPoint | $0.9778_{\pm0.004}$ | $0.9756_{\pm0.004}$ | $0.9756_{\pm0.004}$ | $0.9822_{\pm0.004}$ | $0.98_{\pm0.0}$ | $0.9778_{\pm0.008}$ | $\mathbf{0.9933}_{\pm0.0}$ | $0.98_{\pm0.007}$ | $0.9778_{\pm0.004}$ |
| GunPointAgeSpan | $0.9842_{\pm0.003}$ | $0.9852_{\pm0.004}$ | $0.9863_{\pm0.005}$ | $\mathbf{0.9895}_{\pm0.002}$ | $0.9873_{\pm0.005}$ | $0.9873_{\pm0.0}$ | $0.9873_{\pm0.0}$ | $0.9821_{\pm0.002}$ | $0.9821_{\pm0.002}$ |
| GunPointMaleVersusFemale | $\mathbf{1.0}_{\pm0.0}$ | $0.9979_{\pm0.002}$ | $\mathbf{1.0}_{\pm0.0}$ | $0.9947_{\pm0.002}$ | $0.9979_{\pm0.002}$ | $0.9947_{\pm0.004}$ | $\mathbf{1.0}_{\pm0.0}$ | $\mathbf{1.0}_{\pm0.0}$ | $\mathbf{1.0}_{\pm0.0}$ |
| GunPointOldVersusYoung | $0.9968_{\pm0.0}$ | $0.9968_{\pm0.0}$ | $\mathbf{0.9989}_{\pm0.002}$ | $0.9958_{\pm0.002}$ | $0.9968_{\pm0.0}$ | $0.9947_{\pm0.002}$ | $0.9968_{\pm0.0}$ | $0.9958_{\pm0.002}$ | $0.9968_{\pm0.0}$ |
| Ham | $0.6698_{\pm0.031}$ | $0.6952_{\pm0.025}$ | $0.6698_{\pm0.024}$ | $\mathbf{0.7397}_{\pm0.011}$ | $0.6762_{\pm0.01}$ | $0.6889_{\pm0.005}$ | $0.6825_{\pm0.011}$ | $0.6635_{\pm0.024}$ | $0.6667_{\pm0.016}$ |
| HandOutlines | $0.9036_{\pm0.006}$ | $0.9144_{\pm0.004}$ | $\mathbf{0.9234}_{\pm0.003}$ | $0.9216_{\pm0.003}$ | $0.9072_{\pm0.004}$ | $0.9189_{\pm0.0}$ | $0.8991_{\pm0.006}$ | $0.9117_{\pm0.006}$ | $0.9_{\pm0.003}$ |
| Haptics | $0.5011_{\pm0.011}$ | $\mathbf{0.5584}_{\pm0.015}$ | $0.4968_{\pm0.01}$ | $0.5141_{\pm0.008}$ | $0.553_{\pm0.011}$ | $0.5325_{\pm0.013}$ | $0.5476_{\pm0.016}$ | $0.5216_{\pm0.013}$ | $0.5206_{\pm0.01}$ |
| Herring | $0.6302_{\pm0.018}$ | $0.6354_{\pm0.033}$ | $0.5885_{\pm0.018}$ | $0.6458_{\pm0.018}$ | $0.6094_{\pm0.041}$ | $\mathbf{0.6719}_{\pm0.027}$ | $0.599_{\pm0.024}$ | $0.5938_{\pm0.016}$ | $0.6458_{\pm0.036}$ |
| HouseTwenty | $0.9524_{\pm0.013}$ | $\mathbf{0.9636}_{\pm0.005}$ | $0.9384_{\pm0.005}$ | $0.944_{\pm0.01}$ | $0.9552_{\pm0.005}$ | $0.9608_{\pm0.013}$ | $0.958_{\pm0.0}$ | $\mathbf{0.9636}_{\pm0.005}$ | $0.9496_{\pm0.0}$ |
| InlineSkate | $0.3855_{\pm0.009}$ | $0.4673_{\pm0.008}$ | $0.3848_{\pm0.006}$ | $0.3691_{\pm0.007}$ | $0.4412_{\pm0.004}$ | $0.4442_{\pm0.012}$ | $0.4091_{\pm0.007}$ | $\mathbf{0.4739}_{\pm0.026}$ | $0.3867_{\pm0.004}$ |
| InsectEPGRegularTrain | $\mathbf{1.0}_{\pm0.0}$ | $\mathbf{1.0}_{\pm0.0}$ | $\mathbf{1.0}_{\pm0.0}$ | $\mathbf{1.0}_{\pm0.0}$ | $\mathbf{1.0}_{\pm0.0}$ | $\mathbf{1.0}_{\pm0.0}$ | $\mathbf{1.0}_{\pm0.0}$ | $\mathbf{1.0}_{\pm0.0}$ | $\mathbf{1.0}_{\pm0.0}$ |
| InsectEPGSmallTrain | $\mathbf{1.0}_{\pm0.0}$ | $\mathbf{1.0}_{\pm0.0}$ | $\mathbf{1.0}_{\pm0.0}$ | $0.996_{\pm0.0}$ | $0.996_{\pm0.004}$ | $0.9866_{\pm0.014}$ | $0.996_{\pm0.007}$ | $0.992_{\pm0.007}$ | $\mathbf{1.0}_{\pm0.0}$ |
| InsectWingbeatSound | $0.6056_{\pm0.009}$ | $0.599_{\pm0.003}$ | $0.6231_{\pm0.005}$ | $0.6391_{\pm0.007}$ | $\mathbf{0.6485}_{\pm0.007}$ | $0.6323_{\pm0.004}$ | $0.5909_{\pm0.003}$ | $0.5933_{\pm0.003}$ | $0.6114_{\pm0.004}$ |

*Table 22.* Self-ensembling and cross-model embedding fusion on UCR. The second part of the table.

| | Mantis | SE-Mantis | Mantis & Catch22+ | Mantis & MOMENT | Mantis & TiRex | Mantis & Chronos2 | Mantis & TiViT-H | Mantis & TiConvNext | Mantis & NuTime |
|---|---|---|---|---|---|---|---|---|---|
| ItalyPowerDemand | $0.9248_{\pm 0.004}$ | $0.943_{\pm 0.004}$ | $0.943_{\pm 0.001}$ | $0.9462_{\pm 0.003}$ | $\mathbf{0.964}_{\pm 0.001}$ | $0.9559_{\pm 0.004}$ | $0.8892_{\pm 0.014}$ | $0.9329_{\pm 0.004}$ | $0.9291_{\pm 0.004}$ |
| LargeKitchenAppliances | $0.7662_{\pm 0.006}$ | $0.7884_{\pm 0.018}$ | $0.8107_{\pm 0.012}$ | $0.7822_{\pm 0.007}$ | $0.7849_{\pm 0.009}$ | $0.8462_{\pm 0.009}$ | $0.8116_{\pm 0.004}$ | $\mathbf{0.8596}_{\pm 0.002}$ | $0.7742_{\pm 0.003}$ |
| Lightning2 | $0.7268_{\pm 0.009}$ | $0.7432_{\pm 0.009}$ | $0.7377_{\pm 0.016}$ | $0.7705_{\pm 0.016}$ | $0.7486_{\pm 0.019}$ | $0.7377_{\pm 0.0}$ | $\mathbf{0.7923}_{\pm 0.034}$ | $0.7596_{\pm 0.041}$ | $0.7158_{\pm 0.019}$ |
| Lightning7 | $0.7215_{\pm 0.044}$ | $0.7397_{\pm 0.036}$ | $0.7215_{\pm 0.021}$ | $0.7808_{\pm 0.014}$ | $0.7443_{\pm 0.021}$ | $0.6986_{\pm 0.027}$ | $\mathbf{0.7945}_{\pm 0.024}$ | $0.758_{\pm 0.021}$ | $0.726_{\pm 0.036}$ |
| Mallat | $0.8887_{\pm 0.002}$ | $\mathbf{0.9201}_{\pm 0.005}$ | $0.8984_{\pm 0.013}$ | $0.9102_{\pm 0.005}$ | $0.9026_{\pm 0.01}$ | $0.9021_{\pm 0.007}$ | $0.9154_{\pm 0.01}$ | $0.8913_{\pm 0.036}$ | $0.882_{\pm 0.005}$ |
| Meat | $0.9111_{\pm 0.025}$ | $0.9167_{\pm 0.017}$ | $\mathbf{0.95}_{\pm 0.017}$ | $0.9278_{\pm 0.019}$ | $0.9111_{\pm 0.01}$ | $0.9333_{\pm 0.0}$ | $0.9278_{\pm 0.01}$ | $0.9_{\pm 0.0}$ | $0.9167_{\pm 0.0}$ |
| MedicalImages | $0.7522_{\pm 0.003}$ | $\mathbf{0.7654}_{\pm 0.005}$ | $0.7566_{\pm 0.003}$ | $0.7566_{\pm 0.006}$ | $0.7522_{\pm 0.005}$ | $0.7421_{\pm 0.006}$ | $0.7575_{\pm 0.008}$ | $0.7588_{\pm 0.004}$ | $0.7623_{\pm 0.001}$ |
| MelbournePedestrian | $0.9513_{\pm 0.001}$ | $0.9554_{\pm 0.002}$ | $0.9587_{\pm 0.002}$ | $0.956_{\pm 0.002}$ | $0.9631_{\pm 0.0}$ | $\mathbf{0.9634}_{\pm 0.002}$ | $0.9519_{\pm 0.002}$ | $0.9505_{\pm 0.001}$ | $0.9552_{\pm 0.001}$ |
| MiddlePhalanxOutlineAgeGroup | $0.5801_{\pm 0.004}$ | $0.5996_{\pm 0.023}$ | $0.5866_{\pm 0.007}$ | $0.5801_{\pm 0.004}$ | $0.5909_{\pm 0.013}$ | $0.5649_{\pm 0.017}$ | $0.5909_{\pm 0.017}$ | $\mathbf{0.6234}_{\pm 0.019}$ | $0.5887_{\pm 0.019}$ |
| MiddlePhalanxOutlineCorrect | $0.8339_{\pm 0.004}$ | $0.8499_{\pm 0.004}$ | $0.8465_{\pm 0.009}$ | $\mathbf{0.8763}_{\pm 0.006}$ | $0.8396_{\pm 0.005}$ | $0.8694_{\pm 0.006}$ | $0.8442_{\pm 0.019}$ | $0.8431_{\pm 0.017}$ | $0.8328_{\pm 0.007}$ |
| MiddlePhalanxTW | $0.5325_{\pm 0.006}$ | $0.5519_{\pm 0.017}$ | $0.5368_{\pm 0.01}$ | $\mathbf{0.5628}_{\pm 0.025}$ | $0.5411_{\pm 0.02}$ | $0.5584_{\pm 0.011}$ | $0.5541_{\pm 0.014}$ | $0.5606_{\pm 0.016}$ | $0.5195_{\pm 0.017}$ |
| MixedShapesRegularTrain | $0.9467_{\pm 0.0}$ | $\mathbf{0.9638}_{\pm 0.001}$ | $0.9478_{\pm 0.0}$ | $0.9485_{\pm 0.001}$ | $0.9577_{\pm 0.001}$ | $0.9524_{\pm 0.002}$ | $0.9535_{\pm 0.002}$ | $0.9578_{\pm 0.002}$ | $0.9491_{\pm 0.001}$ |
| MixedShapesSmallTrain | $0.9157_{\pm 0.001}$ | $\mathbf{0.9457}_{\pm 0.0}$ | $0.9211_{\pm 0.002}$ | $0.9146_{\pm 0.002}$ | $0.9282_{\pm 0.003}$ | $0.9181_{\pm 0.003}$ | $0.9266_{\pm 0.001}$ | $0.9203_{\pm 0.003}$ | $0.9245_{\pm 0.005}$ |
| MoteStrain | $0.931_{\pm 0.004}$ | $0.9481_{\pm 0.01}$ | $0.939_{\pm 0.004}$ | $0.9281_{\pm 0.004}$ | $0.931_{\pm 0.005}$ | $0.9321_{\pm 0.002}$ | $0.9523_{\pm 0.003}$ | $0.9502_{\pm 0.005}$ | $\mathbf{0.9526}_{\pm 0.006}$ |
| NonInvasiveFetalECGThorax1 | $0.864_{\pm 0.002}$ | $0.9045_{\pm 0.003}$ | $0.9009_{\pm 0.001}$ | $\mathbf{0.9116}_{\pm 0.005}$ | $0.8894_{\pm 0.003}$ | $0.8741_{\pm 0.001}$ | $0.884_{\pm 0.002}$ | $0.8816_{\pm 0.001}$ | $0.864_{\pm 0.0}$ |
| NonInvasiveFetalECGThorax2 | $0.8867_{\pm 0.0}$ | $0.9211_{\pm 0.002}$ | $0.9264_{\pm 0.004}$ | $\mathbf{0.9328}_{\pm 0.002}$ | $0.9101_{\pm 0.001}$ | $0.9009_{\pm 0.001}$ | $0.904_{\pm 0.005}$ | $0.9006_{\pm 0.001}$ | $0.8911_{\pm 0.004}$ |
| OSULeaf | $0.9353_{\pm 0.002}$ | $0.9656_{\pm 0.002}$ | $0.927_{\pm 0.006}$ | $0.8953_{\pm 0.002}$ | $0.9656_{\pm 0.006}$ | $0.9366_{\pm 0.005}$ | $0.9532_{\pm 0.005}$ | $\mathbf{0.978}_{\pm 0.002}$ | $0.9339_{\pm 0.007}$ |
| OliveOil | $0.8667_{\pm 0.033}$ | $0.8444_{\pm 0.019}$ | $0.8667_{\pm 0.0}$ | $\mathbf{0.9}_{\pm 0.0}$ | $0.8889_{\pm 0.019}$ | $0.8444_{\pm 0.019}$ | $0.8_{\pm 0.033}$ | $0.8556_{\pm 0.019}$ | $0.8889_{\pm 0.019}$ |
| PLAID | $0.8187_{\pm 0.004}$ | $0.8218_{\pm 0.004}$ | $0.8547_{\pm 0.01}$ | $0.8299_{\pm 0.006}$ | $0.8808_{\pm 0.0}$ | $0.874_{\pm 0.003}$ | $0.8876_{\pm 0.005}$ | $\mathbf{0.897}_{\pm 0.006}$ | $0.8299_{\pm 0.003}$ |
| PhalangesOutlinesCorrect | $0.824_{\pm 0.002}$ | $0.8287_{\pm 0.003}$ | $0.831_{\pm 0.003}$ | $\mathbf{0.845}_{\pm 0.004}$ | $0.8345_{\pm 0.006}$ | $0.8399_{\pm 0.003}$ | $0.8236_{\pm 0.005}$ | $0.8174_{\pm 0.003}$ | $0.8252_{\pm 0.006}$ |
| Phoneme | $0.3641_{\pm 0.005}$ | $\mathbf{0.4223}_{\pm 0.008}$ | $0.3708_{\pm 0.003}$ | $0.3367_{\pm 0.003}$ | $0.3901_{\pm 0.004}$ | $0.4023_{\pm 0.004}$ | $0.379_{\pm 0.005}$ | $0.3806_{\pm 0.007}$ | $0.3588_{\pm 0.003}$ |
| PickupGestureWiimoteZ | $0.7867_{\pm 0.031}$ | $0.7467_{\pm 0.031}$ | $0.7333_{\pm 0.012}$ | $0.7267_{\pm 0.023}$ | $0.76_{\pm 0.02}$ | $0.7133_{\pm 0.042}$ | $0.84_{\pm 0.02}$ | $\mathbf{0.8467}_{\pm 0.042}$ | $0.7467_{\pm 0.012}$ |
| PigAirwayPressure | $0.4904_{\pm 0.017}$ | $0.5272_{\pm 0.026}$ | $0.4808_{\pm 0.021}$ | $0.3109_{\pm 0.015}$ | $0.4968_{\pm 0.015}$ | $0.4535_{\pm 0.029}$ | $0.5561_{\pm 0.003}$ | $\mathbf{0.625}_{\pm 0.021}$ | $0.5_{\pm 0.01}$ |
| PigArtPressure | $0.9343_{\pm 0.003}$ | $0.9343_{\pm 0.01}$ | $0.9199_{\pm 0.006}$ | $0.7869_{\pm 0.018}$ | $0.9487_{\pm 0.019}$ | $0.9038_{\pm 0.005}$ | $0.9103_{\pm 0.012}$ | $\mathbf{0.9535}_{\pm 0.003}$ | $0.9375_{\pm 0.013}$ |
| PigCVP | $0.8974_{\pm 0.02}$ | $0.8718_{\pm 0.018}$ | $0.883_{\pm 0.003}$ | $0.8285_{\pm 0.027}$ | $\mathbf{0.9151}_{\pm 0.006}$ | $0.8622_{\pm 0.015}$ | $0.8718_{\pm 0.007}$ | $0.8606_{\pm 0.005}$ | $0.9071_{\pm 0.007}$ |
| Plane | $\mathbf{1.0}_{\pm 0.0}$ | $\mathbf{1.0}_{\pm 0.0}$ | $\mathbf{1.0}_{\pm 0.0}$ | $\mathbf{1.0}_{\pm 0.0}$ | $\mathbf{1.0}_{\pm 0.0}$ | $\mathbf{1.0}_{\pm 0.0}$ | $\mathbf{1.0}_{\pm 0.0}$ | $\mathbf{1.0}_{\pm 0.0}$ | $\mathbf{1.0}_{\pm 0.0}$ |
| PowerCons | $0.9648_{\pm 0.003}$ | $0.9667_{\pm 0.006}$ | $\mathbf{0.9833}_{\pm 0.006}$ | $0.9759_{\pm 0.014}$ | $0.963_{\pm 0.003}$ | $0.9407_{\pm 0.006}$ | $0.9278_{\pm 0.02}$ | $0.9333_{\pm 0.015}$ | $0.9593_{\pm 0.008}$ |
| ProximalPhalanxOutlineAgeGroup | $0.8293_{\pm 0.005}$ | $0.839_{\pm 0.005}$ | $0.8325_{\pm 0.003}$ | $0.8374_{\pm 0.003}$ | $\mathbf{0.8472}_{\pm 0.006}$ | $0.8423_{\pm 0.003}$ | $0.8439_{\pm 0.0}$ | $0.8439_{\pm 0.008}$ | $0.8293_{\pm 0.0}$ |
| ProximalPhalanxOutlineCorrect | $0.8671_{\pm 0.005}$ | $0.8866_{\pm 0.0}$ | $0.8774_{\pm 0.002}$ | $0.8694_{\pm 0.0}$ | $0.8866_{\pm 0.003}$ | $\mathbf{0.89}_{\pm 0.006}$ | $0.8694_{\pm 0.007}$ | $0.8603_{\pm 0.009}$ | $0.8729_{\pm 0.006}$ |
| ProximalPhalanxTW | $\mathbf{0.8211}_{\pm 0.016}$ | $0.8195_{\pm 0.013}$ | $0.8195_{\pm 0.0}$ | $0.8179_{\pm 0.007}$ | $0.813_{\pm 0.003}$ | $0.8098_{\pm 0.005}$ | $0.8146_{\pm 0.005}$ | $0.8033_{\pm 0.006}$ | $0.8163_{\pm 0.006}$ |
| RefrigerationDevices | $0.5636_{\pm 0.006}$ | $0.5369_{\pm 0.015}$ | $0.5653_{\pm 0.0}$ | $0.544_{\pm 0.0}$ | $0.5627_{\pm 0.007}$ | $0.5698_{\pm 0.006}$ | $0.5627_{\pm 0.005}$ | $\mathbf{0.5813}_{\pm 0.009}$ | $0.5547_{\pm 0.007}$ |
| Rock | $0.82_{\pm 0.02}$ | $0.8467_{\pm 0.031}$ | $0.8_{\pm 0.02}$ | $0.8267_{\pm 0.012}$ | $0.8667_{\pm 0.012}$ | $0.84_{\pm 0.02}$ | $0.8667_{\pm 0.031}$ | $\mathbf{0.88}_{\pm 0.02}$ | $0.7867_{\pm 0.042}$ |
| ScreenType | $0.4596_{\pm 0.014}$ | $0.4924_{\pm 0.008}$ | $0.4524_{\pm 0.004}$ | $0.4284_{\pm 0.013}$ | $0.4773_{\pm 0.005}$ | $0.4862_{\pm 0.006}$ | $\mathbf{0.5778}_{\pm 0.013}$ | $0.5484_{\pm 0.016}$ | $0.5031_{\pm 0.01}$ |
| SemgHandGenderCh2 | $0.9139_{\pm 0.006}$ | $0.9417_{\pm 0.007}$ | $0.9294_{\pm 0.001}$ | $0.9028_{\pm 0.007}$ | $0.9433_{\pm 0.004}$ | $\mathbf{0.9461}_{\pm 0.004}$ | $0.9067_{\pm 0.004}$ | $0.9061_{\pm 0.005}$ | $0.9189_{\pm 0.004}$ |
| SemgHandMovementCh2 | $0.7311_{\pm 0.008}$ | $0.7667_{\pm 0.012}$ | $0.7689_{\pm 0.006}$ | $0.6711_{\pm 0.006}$ | $0.7556_{\pm 0.002}$ | $0.763_{\pm 0.008}$ | $0.7437_{\pm 0.017}$ | $\mathbf{0.7756}_{\pm 0.008}$ | $0.74_{\pm 0.006}$ |
| SemgHandSubjectCh2 | $0.8341_{\pm 0.011}$ | $0.8437_{\pm 0.006}$ | $0.863_{\pm 0.001}$ | $0.8119_{\pm 0.013}$ | $0.857_{\pm 0.006}$ | $0.8674_{\pm 0.007}$ | $0.8726_{\pm 0.003}$ | $\mathbf{0.8867}_{\pm 0.006}$ | $0.8348_{\pm 0.003}$ |
| ShakeGestureWiimoteZ | $\mathbf{0.9333}_{\pm 0.012}$ | $\mathbf{0.9333}_{\pm 0.023}$ | $0.9267_{\pm 0.023}$ | $0.9_{\pm 0.0}$ | $\mathbf{0.9333}_{\pm 0.031}$ | $0.9267_{\pm 0.031}$ | $0.9_{\pm 0.02}$ | $0.8467_{\pm 0.012}$ | $0.9267_{\pm 0.023}$ |
| ShapeletSim | $0.9593_{\pm 0.003}$ | $0.9907_{\pm 0.004}$ | $0.9852_{\pm 0.003}$ | $0.9815_{\pm 0.008}$ | $0.9685_{\pm 0.018}$ | $\mathbf{1.0}_{\pm 0.0}$ | $\mathbf{1.0}_{\pm 0.0}$ | $\mathbf{1.0}_{\pm 0.0}$ | $0.9704_{\pm 0.03}$ |
| ShapesAll | $0.87_{\pm 0.009}$ | $0.8811_{\pm 0.003}$ | $0.8817_{\pm 0.002}$ | $0.875_{\pm 0.007}$ | $0.8811_{\pm 0.005}$ | $\mathbf{0.8967}_{\pm 0.002}$ | $0.8939_{\pm 0.004}$ | $0.88_{\pm 0.004}$ | $0.8756_{\pm 0.015}$ |
| SmallKitchenAppliances | $\mathbf{0.8373}_{\pm 0.0}$ | $0.8213_{\pm 0.005}$ | $0.8338_{\pm 0.002}$ | $0.8187_{\pm 0.009}$ | $0.8284_{\pm 0.007}$ | $0.8311_{\pm 0.004}$ | $0.8293_{\pm 0.003}$ | $0.8249_{\pm 0.003}$ | $0.8302_{\pm 0.003}$ |
| SmoothSubspace | $0.9511_{\pm 0.004}$ | $0.9444_{\pm 0.008}$ | $0.9622_{\pm 0.008}$ | $\mathbf{0.9733}_{\pm 0.0}$ | $0.9556_{\pm 0.008}$ | $0.9556_{\pm 0.008}$ | $0.9533_{\pm 0.007}$ | $0.9556_{\pm 0.004}$ | $0.9444_{\pm 0.01}$ |
| SonyAIBORobotSurface1 | $0.8231_{\pm 0.011}$ | $\mathbf{0.8502}_{\pm 0.013}$ | $0.8286_{\pm 0.009}$ | $0.8458_{\pm 0.008}$ | $0.8342_{\pm 0.02}$ | $0.7842_{\pm 0.017}$ | $0.7965_{\pm 0.012}$ | $0.7909_{\pm 0.013}$ | $0.8192_{\pm 0.004}$ |
| SonyAIBORobotSurface2 | $0.922_{\pm 0.009}$ | $\mathbf{0.9503}_{\pm 0.005}$ | $0.936_{\pm 0.005}$ | $0.844_{\pm 0.006}$ | $0.8695_{\pm 0.004}$ | $0.8961_{\pm 0.024}$ | $0.9307_{\pm 0.006}$ | $0.9307_{\pm 0.003}$ | $0.9244_{\pm 0.006}$ |
| StarLightCurves | $0.9806_{\pm 0.0}$ | $0.9805_{\pm 0.0}$ | $0.9808_{\pm 0.0}$ | $\mathbf{0.9816}_{\pm 0.001}$ | $0.98_{\pm 0.0}$ | $0.981_{\pm 0.0}$ | $0.9794_{\pm 0.0}$ | $0.9806_{\pm 0.0}$ | $0.9809_{\pm 0.0}$ |
| Strawberry | $0.9595_{\pm 0.007}$ | $0.9514_{\pm 0.003}$ | $0.9622_{\pm 0.003}$ | $\mathbf{0.9631}_{\pm 0.003}$ | $0.9568_{\pm 0.0}$ | $0.9559_{\pm 0.002}$ | $0.9505_{\pm 0.006}$ | $0.9486_{\pm 0.003}$ | $0.9613_{\pm 0.002}$ |
| SwedishLeaf | $0.9547_{\pm 0.004}$ | $0.9536_{\pm 0.003}$ | $0.9563_{\pm 0.004}$ | $0.9573_{\pm 0.001}$ | $0.9589_{\pm 0.004}$ | $\mathbf{0.96}_{\pm 0.003}$ | $\mathbf{0.96}_{\pm 0.003}$ | $0.9536_{\pm 0.006}$ | $0.952_{\pm 0.002}$ |
| Symbols | $0.9698_{\pm 0.005}$ | $\mathbf{0.9829}_{\pm 0.004}$ | $0.9729_{\pm 0.005}$ | $0.9625_{\pm 0.005}$ | $0.9715_{\pm 0.008}$ | $0.9796_{\pm 0.003}$ | $0.9822_{\pm 0.003}$ | $0.9812_{\pm 0.002}$ | $0.9715_{\pm 0.006}$ |
| SyntheticControl | $0.99_{\pm 0.0}$ | $0.9889_{\pm 0.002}$ | $0.9922_{\pm 0.002}$ | $0.9911_{\pm 0.002}$ | $0.9933_{\pm 0.0}$ | $0.9933_{\pm 0.003}$ | $0.9933_{\pm 0.0}$ | $\mathbf{0.9967}_{\pm 0.0}$ | $0.9911_{\pm 0.004}$ |
| ToeSegmentation1 | $0.9547_{\pm 0.007}$ | $0.9561_{\pm 0.008}$ | $0.9518_{\pm 0.012}$ | $0.943_{\pm 0.008}$ | $\mathbf{0.9635}_{\pm 0.014}$ | $0.9503_{\pm 0.003}$ | $0.9532_{\pm 0.003}$ | $0.9561_{\pm 0.004}$ | $0.9459_{\pm 0.005}$ |
| ToeSegmentation2 | $0.8821_{\pm 0.004}$ | $0.8923_{\pm 0.008}$ | $0.8795_{\pm 0.004}$ | $0.8641_{\pm 0.004}$ | $\mathbf{0.8949}_{\pm 0.004}$ | $0.8846_{\pm 0.008}$ | $0.8641_{\pm 0.012}$ | $0.8692_{\pm 0.015}$ | $0.8692_{\pm 0.0}$ |
| Trace | $\mathbf{1.0}_{\pm 0.0}$ | $\mathbf{1.0}_{\pm 0.0}$ | $\mathbf{1.0}_{\pm 0.0}$ | $\mathbf{1.0}_{\pm 0.0}$ | $\mathbf{1.0}_{\pm 0.0}$ | $\mathbf{1.0}_{\pm 0.0}$ | $\mathbf{1.0}_{\pm 0.0}$ | $\mathbf{1.0}_{\pm 0.0}$ | $\mathbf{1.0}_{\pm 0.0}$ |
| TwoLeadECG | $0.9956_{\pm 0.001}$ | $0.9965_{\pm 0.001}$ | $0.9959_{\pm 0.001}$ | $0.9871_{\pm 0.001}$ | $0.9971_{\pm 0.001}$ | $0.9959_{\pm 0.001}$ | $\mathbf{0.9982}_{\pm 0.001}$ | $0.9886_{\pm 0.004}$ | $0.9968_{\pm 0.001}$ |
| TwoPatterns | $0.9819_{\pm 0.001}$ | $0.9822_{\pm 0.0}$ | $\mathbf{0.9932}_{\pm 0.001}$ | $0.9863_{\pm 0.002}$ | $0.9914_{\pm 0.001}$ | $0.981_{\pm 0.001}$ | $0.9908_{\pm 0.01}$ | $0.9919_{\pm 0.001}$ | $0.9839_{\pm 0.001}$ |
| UMD | $0.9815_{\pm 0.004}$ | $0.9861_{\pm 0.007}$ | $\mathbf{0.9931}_{\pm 0.0}$ | $0.9861_{\pm 0.0}$ | $0.9861_{\pm 0.007}$ | $\mathbf{0.9931}_{\pm 0.007}$ | $0.9861_{\pm 0.007}$ | $0.9907_{\pm 0.004}$ | $0.9815_{\pm 0.011}$ |
| UWaveGestureLibraryAll | $0.8848_{\pm 0.001}$ | $0.9161_{\pm 0.003}$ | $\mathbf{0.9415}_{\pm 0.001}$ | $0.9257_{\pm 0.001}$ | $0.9261_{\pm 0.003}$ | $0.9362_{\pm 0.001}$ | $0.8969_{\pm 0.001}$ | $0.9036_{\pm 0.004}$ | $0.9093_{\pm 0.002}$ |
| UWaveGestureLibraryX | $0.813_{\pm 0.001}$ | $0.826_{\pm 0.003}$ | $\mathbf{0.8279}_{\pm 0.005}$ | $0.82_{\pm 0.001}$ | $0.8183_{\pm 0.004}$ | $0.818_{\pm 0.004}$ | $0.8172_{\pm 0.003}$ | $0.8268_{\pm 0.001}$ | $0.8227_{\pm 0.001}$ |
| UWaveGestureLibraryY | $0.7572_{\pm 0.003}$ | $0.7714_{\pm 0.004}$ | $0.768_{\pm 0.003}$ | $0.7498_{\pm 0.004}$ | $0.7588_{\pm 0.004}$ | $0.7687_{\pm 0.003}$ | $0.7637_{\pm 0.002}$ | $\mathbf{0.7799}_{\pm 0.003}$ | $0.7674_{\pm 0.002}$ |
| UWaveGestureLibraryZ | $0.7587_{\pm 0.0}$ | $0.7658_{\pm 0.001}$ | $0.7661_{\pm 0.005}$ | $0.7657_{\pm 0.003}$ | $\mathbf{0.7695}_{\pm 0.004}$ | $0.76_{\pm 0.0}$ | $0.7626_{\pm 0.004}$ | $0.7677_{\pm 0.002}$ | $0.7618_{\pm 0.003}$ |
| Wafer | $0.9944_{\pm 0.001}$ | $0.9985_{\pm 0.001}$ | $0.9966_{\pm 0.0}$ | $0.9951_{\pm 0.0}$ | $0.9978_{\pm 0.0}$ | $0.9942_{\pm 0.0}$ | $0.9977_{\pm 0.0}$ | $\mathbf{0.9992}_{\pm 0.0}$ | $0.9948_{\pm 0.0}$ |
| Wine | $0.8086_{\pm 0.06}$ | $0.7037_{\pm 0.074}$ | $0.821_{\pm 0.043}$ | $0.8642_{\pm 0.043}$ | $0.784_{\pm 0.047}$ | $\mathbf{0.8704}_{\pm 0.0}$ | $0.784_{\pm 0.021}$ | $0.7593_{\pm 0.0}$ | $0.8148_{\pm 0.032}$ |
| WordSynonyms | $0.6181_{\pm 0.006}$ | $0.6092_{\pm 0.01}$ | $0.6374_{\pm 0.011}$ | $\mathbf{0.6379}_{\pm 0.005}$ | $0.6019_{\pm 0.007}$ | $0.605_{\pm 0.017}$ | $0.5852_{\pm 0.003}$ | $0.5977_{\pm 0.018}$ | $0.6191_{\pm 0.003}$ |
| Worms | $0.7576_{\pm 0.02}$ | $0.8182_{\pm 0.013}$ | $0.7576_{\pm 0.015}$ | $0.7619_{\pm 0.02}$ | $0.8009_{\pm 0.027}$ | $0.7662_{\pm 0.013}$ | $\mathbf{0.8485}_{\pm 0.007}$ | $0.8182_{\pm 0.0}$ | $0.7706_{\pm 0.007}$ |
| WormsTwoClass | $0.8182_{\pm 0.013}$ | $\mathbf{0.8485}_{\pm 0.02}$ | $0.8182_{\pm 0.013}$ | $0.8182_{\pm 0.013}$ | $0.7965_{\pm 0.027}$ | $0.8009_{\pm 0.02}$ | $0.8268_{\pm 0.007}$ | $0.8355_{\pm 0.015}$ | $0.8009_{\pm 0.007}$ |
| Yoga | $0.848_{\pm 0.003}$ | $0.8463_{\pm 0.003}$ | $0.8532_{\pm 0.007}$ | $\mathbf{0.8594}_{\pm 0.005}$ | $0.8418_{\pm 0.004}$ | $0.8381_{\pm 0.005}$ | $0.8342_{\pm 0.003}$ | $0.8357_{\pm 0.005}$ | $0.8541_{\pm 0.006}$ |

