# OpenReview forum: "Mantis: Lightweight Foundation Model for Time Series Classification"
_ICML.cc/2026/Conference — ICML 2026 regular_

### Official Review · Reviewer_Arid · 2026-03-09

**Soundness:** 3
**Presentation:** 3
**Significance:** 3
**Originality:** 2
**Overall Recommendation:** 5
**Confidence:** 4

**Summary:**

This paper addresses the relatively underexplored problem of time series classification within the foundation model paradigm, a research area where most prior work has focused on forecasting. The authors propose Mantis, a transformer-based foundation model that is pre-trained exclusively on synthetic data using self-supervised contrastive learning. A central contribution claimed by the paper is a novel token generator unit, which the authors argue is crucial for enabling transformers to effectively process time series by improving tokenization of raw sequences. In addition, the paper introduces an enhanced test-time inference pipeline designed to close the performance gap between the proposed foundation model and specialized task-specific methods. This pipeline combines several techniques, including the use of intermediate-layer representations, input-perturbation self-ensembling, and cross-model embedding fusion. According to the experimental evaluation, conducted across four diverse collections of time series datasets spanning multiple application domains, the proposed approach achieves state-of-the-art performance among existing time series foundation models and shows competitive results relative to strong specialized baselines. The authors position Mantis as evidence that foundation models trained with synthetic data and contrastive objectives can serve as effective zero-shot feature extractors for time series classification tasks.

**Compliance With Llm Reviewing Policy:**

Affirmed.

**Final Justification:**

The quality of this paper is sufficient for acceptance at ICML, and the authors have substantively addressed all of my concerns in the rebuttal.

**Key Questions For Authors:**

Please check the Weaknesses part.

**Limitations:**

yes

**Strengths And Weaknesses:**

Strengths:
1. This paper focuses on lightweight foundation models for time series classification, a domain that has received relatively limited dedicated attention.
2. The paper is clearly written, easy to read, and does not contain any obvious writing errors.

Weaknesses:
1. The framework relies on relatively standard techniques and does not present clear technical or theoretical innovations.
2. The paper lacks sufficient comparisons with existing methods. The authors should compare Mantis with existing approaches—especially those based on contrastive learning—to demonstrate the design advantages of Mantis.
3. When processing multichannel sequences, the method handles each channel independently and concatenates them in the representation dimension. This approach cannot model the relationships between channels, nor does it allow the model to focus on particularly important channels.
4. The authors should further check the details in the tables, for example the bold formatting for the SelfRegulationSCP1 dataset in Table 2.

---

> ### Author Rebuttal · Authors · 2026-03-31
>
> We thank the reviewer for their positive assessment and constructive comments.
> > The framework relies on relatively standard techniques and does not present clear innovations.
>
> While building on established techniques, Mantis introduces key novelties: (1) It is the first time series classification foundation model pre-trained via a contrastive objective, requiring extensive ablation (see below). (2) A novel token generator unit that demonstrably enhances performance. (3) A new methodology maximizing zero-shot feature extraction. Our observation that intermediate representations unlock data scaling laws is novel, suggesting more efficient layer-by-layer pre-training.
>
> Augmentation Ablation
>
> We tested several contrastive augmentations: Random Crop Resize (RCR, our choice), white noise, a Fourier transform surrogate (FTS, Rommel et al., 2022), and Random Masking (RMask). RCR outperforms the others on UCR (table below). We will include this in the revision.
>
> | Augmentation | Noise | FTS | RMask | RCR |
> | :--- | :--- | :--- | :--- | :--- |
> | Acc (UCR) | 0.751 | 0.768 | 0.698 | **0.788** |
>
> (Rommel et al., 2022) Data augmentation for learning predictive models on EEG: a systematic comparison.
>
> > *The paper lacks comparisons with contrastive learning methods*
>
> We appreciate this suggestion. To address it, we conducted an additional experiment pre-training Mantis with a triplet contrastive loss (TLoss; Franceschi et al., 2019) to compare it against our InfoNCE-based objective. The results on the UCR benchmark (see table below) confirm that our chosen pre-training loss outperforms TLoss.
>
> | Contrastive learning type | Ours (InfoNCE-based) | TLoss |
> | :--- | :--- | :--- |
> | Acc (UCR) | **0.788** | 0.767|
>
> Furthermore, Figure 1 already compares Mantis against per-dataset contrastive methods like TLoss and TS2Vec. Retraining these specialized models foundationally requires prohibitive computational effort.
>
> > *On channel independence*
>
> We agree that multivariate classification is critical but challenging. We focused on univariate encoding because:
>
> - Few foundation models gain significantly from cross-channel relationships (e.g., Chronos2 shows negligible multivariate improvement, Fig. 4a in their paper).
> - Classification uses linear probing. Concatenating channel embeddings means the final linear head inherently models deep feature interactions.
> - Flexibility: Multivariate topologies vary wildly (e.g., 2D EEG correlations), and cross-channel attention on datasets with thousands of channels (DuckDuckGeese, PEMS-SF) is memory-prohibitive. Our lightweight design keeps Mantis deployable on edge devices (e.g., HAR wearables).
>
> **A Partial Solution: Adapters**
> To model channel interactions, we can prepend input adapters (Benechehab et al., ICML '25) to linearly mix original channels. Keeping min(10, d) components via PCA, or using a 10-channel Linear layer, noticeably improved performance on 10 UEA datasets:
>
> |Adapter                | None |  PCA  | Linear|
> | :--- | :--- | :--- | :--- |
> |CharacterTrajectories  |0.976 | 0.977 | 0.988 |
> |DuckDuckGeese          |0.54  | 0.58  | 0.6   |
> |EigenWorms             |0.814 | 0.863 | 0.855 |
> |FaceDetection          |0.549 | 0.553 | 0.592 |
> |HandMovementDirection  |0.297 | 0.338 | 0.338 |
> |Handwriting            |0.281 | 0.315 | 0.514 |
> |MotorImagery           |0.467 | 0.48  | 0.5   |
> |NATOPS                 |0.889 | 0.939 | 0.906 |
> |SelfRegulationSCP2     |0.517 | 0.533 | 0.533 |
> |SpokenArabicDigits     |0.937 | 0.955 | 0.987 |
>
> Thus, adapters can be a simple, yet efficient solution to increase the performance on multivariate datasets.
>
> **Hidden Obstacles for a Multivariate Foundation Model**
> Extending Mantis to a native multivariate setup proved highly non-trivial during our preliminary research:
>
> 1. Pre-training data: CauKer, our synthetic data generator, would need to be re-engineered for multivariate generation.
> 2. Architecture: Designing an architecture that efficiently handles varying channel counts without bloating the model size is complex.
> 3. Contrastive learning: we found that our contrastive objective converges too quickly on multivariate data, as the model easily finds shortcuts to distinguish between multivariate examples in a batch. Making augmentations more aggressive did not solve this, indicating a need to significantly change the pipeline.
>
> (Benechehab et al., 2025) Adapts: Adapting univariate foundation models to probabilistic multivariate time series forecasting. ICML'25.
>
> > *The authors should further check the details in the tables, for example the bold formatting for the SelfRegulationSCP1 dataset in Table 2.*
>
> Thank you for catching this formatting error. We will carefully proofread the tables and correct the bolding for SelfRegulationSCP1 in the revised manuscript.

---

> > ### Author Rebuttal · Reviewer_Arid · 2026-04-01
> >
> > Thank you for your response. Regarding Weakness 2, the paper does not include a dedicated related work section and only provides a brief discussion in the introduction. I fully understand that space limitations may prevent a more detailed review. You may consider adding a survey-style comparison with existing TSFMs (rather than experimental results) in the appendix to better clarify the positioning of your work within TSFMs.

---

> > > ### Author Response · Authors · 2026-04-02
> > >
> > > We sincerely thank the reviewer for increasing their score and for this helpful clarification.
> > >
> > > We completely agree that a detailed literature review is necessary to properly position Mantis within the rapidly evolving landscape of TSFMs. We will add a dedicated Related Work section to the Appendix or to the main paper, if an extra page is allowed for the camera-ready version, as was the case last year.
> > >
> > > In this section, we will detail the differences between Mantis and other contrastive learning approaches, review different pre-training objectives, pre-training corpora, and key recent TSFMs. Finally, we will outline the recent advances in time series classification, forecasting, and anomaly detection to better highlight their inherent differences.
> > >
> > > We greatly appreciate this suggestion, as it will undoubtedly improve the completeness and context of our paper.

---

### Official Review · Reviewer_NVnr · 2026-03-10

**Soundness:** 4
**Presentation:** 4
**Significance:** 4
**Originality:** 3
**Overall Recommendation:** 5
**Confidence:** 4

**Summary:**

The paper addresses the problem of time series classification, esp.
by means of a time series foundation model. The authors propose
a simple model called Mantis, that scales every univariate time series
to a fixed length and then extracts a fixed number of segments/tokens
by a convolution over its values, its differences and mean and stddev
of its values, and then feeds these tokens into a vanilla transformer.
The output representation is the concatenation of the latent representation
of a CLS token and average pooling of all others. The model is
trained with synthetic data. In experiments on two large time series
classification benchmarks, UCR and UEA-27, they show that Mantis
outperforms tabular and time series foundation models w.r.t.
accuracy, while for UCR the tabular foundation model TabPFN wins
in a higher number of datasets.

**Compliance With Llm Reviewing Policy:**

Affirmed.

**Final Justification:**

Thanks to the authors for their additional analysis. My very positive view on this paper is confirmed.

**Key Questions For Authors:**

- q1. Did you analyze, why Mantis improves the classification accuracy by a larger
  margin for some datasets in UCR so large, that it still wins on average accuracy,
  despite being only second on number of wins, would also have been interesting?
  (w1)

**Strengths And Weaknesses:**

strong points.
- s1. simple and straightforward model: a patch transformer based on plausible
  patch representations (values, differences, summary statistics).
- s2. large scale evaluation with many baselines and two large benchmark datasets.
- s3. strong performance, esp. on UEA-27, setting a new state-of-the-art.
- s4. very clearly written.
- s5. good reproducibility: source code is to be released.

weak points.
- w1. a in-depth analysis why Mantis improves the classification accuracy by a larger
  margin for some datasets in UCR so large, that it still wins on average accuracy,
  despite being only second on number of wins, would also have been interesting.

small points:
- p. 2 "proportional to 32" --> "a multiple of 32".
- p. 2+3 "this normalization is implemented as part of the model architecture".
  on first reading, it was unclear why this is mentioned here. Maybe tell the reader
  that you use mean and stddev as additional features, so you cannot do this during
  preprocessing.
- p. 15, table in sect. B.1: is the accuracy really "0.7829", or this is a typo and you
  meant "78.29" ?

---

> ### Author Rebuttal · Authors · 2026-03-31
>
> We thank the reviewer for their strong positive evaluation and for raising interesting, constructive questions. Please find our detailed response below.
>
> > a in-depth analysis why Mantis improves the classification accuracy by a larger margin for some datasets in UCR so large, that it still wins on average accuracy, despite being only second on number of wins, would also have been interesting.
> Did you analyze, why Mantis improves the classification accuracy by a larger margin for some datasets in UCR so large, that it still wins on average accuracy, despite being only second on number of wins, would also have been interesting?*
>
> We thank the reviewer for highlighting this interesting observation. To better understand this dynamic, we conducted a deeper comparison between Mantis and TabPFN (which has the highest overall win rate on UCR).Specifically, we computed the average number of training samples, sequence length, and number of classes across those UCR datasets where each model outperforms the other. The table below summarizes these results. On average, Mantis excels on datasets with fewer training samples, longer sequence lengths, and a larger number of unique classes.
>
> | | TabPFN | Mantis|
> | :--- | :--- | :--- |
> | Num. of train samples |521| 435.8|
> | Sequence length | 452 | 597 |
> | Num. of classes | 7.3 | 9.8 |
>
> Furthermore, we stratified the UCR datasets based on these characteristics to compare subgroup performance.
>
> Number of training samples (n):
>
> | | TabPFN | Mantis|
> | :--- | :--- | :--- |
> |16<=n<55 |0.857| 0.91|
> |55<=n<200 |0.743 | 0.831 |
> |200<=n<400 |0.654 | 0.703 |
> |n>=400 |0.868 | 0.834 |
>
> Sequence length (t):
> |  | TabPFN | Mantis|
> | :--- | :--- | :--- |
> |15<=t<144 | 0.865 | 0.854 |
> |144<=t<345 | 0.829 | 0.857 |
> |345<=t<720 | 0.767 | 0.82 |
> |t>=720 |0.661 | 0.748 |
>
> Number of classes (K):
> |  | TabPFN | Mantis|
> | :--- | :--- | :--- |
> |K<4 |0.843 | 0.871 |
> |4<=K<10 |0.8 | 0.802 |
> |10<=K<20 |0.735 | 0.771 |
> |K>=20 |0.529 | 0.713 |
>
> These breakdowns reveal two specific subgroups where TabPFN outperforms Mantis:
>
> 1. Short sequence lengths (t < 144): We believe that shorter sequences more closely resemble a standard tabular setup, naturally playing to TabPFN's strengths.
>
> 2. Large training sets (n >= 400): Because TabPFN acts as a supervised feature extractor, its quality scales with the number of samples. It shines with larger sample sizes, albeit at a high computational cost (as discussed in Section 3.3).
>
> Conversely, it is also worth to note where TabPFN struggles significantly. For example, on the PigCVP, PigAirwayPressure, and PigArtPressure datasets, which are extreme few-shot learning problems (only 2 training points per class) with 52 classes, TabPFN's performance degrades catastrophically, whereas Mantis remains robust. We will incorporate this detailed analysis into the revised manuscript, as it provides valuable context to our results.
>
> > *p. 15, table in sect. B.1: is the accuracy really "0.7829", or this is a typo and you meant "78.29" ?*
>
> Thank you for catching this! It is indeed a typo; we meant 78.29. We will correct this in the revision.
>
> > Other minor points:
>
> We appreciate these careful reading notes. We will update the phrasing to "a multiple of 32." and better clarify why normalization cannot be performed entirely during preprocessing and must be part of the architecture.

---

### Official Review · Reviewer_KvPU · 2026-03-12

**Soundness:** 3
**Presentation:** 3
**Significance:** 3
**Originality:** 3
**Overall Recommendation:** 5
**Confidence:** 5

**Summary:**

This paper introduces Mantis, a foundation model for time series classification.
Compared to other time series foundation models, Mantis does not pretrain on imputation or forecasting. Instead, it uses the InfoNCE loss, used in contrastive unsupervised learning.
This loss encourages invariance for the selected Random Crop Resize augmentation whilst at the same time maximizing separability against other time series.
Mantis uses a novel Token Generator Unit which incorporates various types of information about the time series.
Mantis is pretrained only on synthetic data.
This work further introduces a set of strategies on how to apply a pretrained model to a downstream classification task.
These strategies include:
layer-wise representations: considering not only the last layer as the optimal one for classification, but instead analyzing which layer leads to the best classification performance.
Output token aggregation: Instead of only using the representation of the last token for classification, also the average of the other tokens is used.
Further investigated, but not included in main comparison:
Self ensemble (using Mantis on different input perturbations) and Cross model embedding fusion (combining embeddings of multiple different foundation models).

This paper tests mantis on a variety of classification tasks and demonstrates overall improvements in performance over SOTA.

**Compliance With Llm Reviewing Policy:**

Affirmed.

**Final Justification:**

My final recommendation is to accept this paper, due to high quality in writing and experiments.
The InfoNCE added in the rebuttal is appreciated and highlights that a main novelty of the paper (applying contrastive predictive coding to timeseries) is of importance.
This further strengthens my confidence to accept this paper.

**Key Questions For Authors:**

What are the limits in terms of minimum and maximum allowed sequence length, due to the fixed number of patches?

Did you also try using both real and cauker data during pretraining?

**Limitations:**

yes

**Strengths And Weaknesses:**

Soundness:
+ Overall the paper is sound.
+ The methodology is clearly explained
+ The main claim is improved classification performance, which clearly supported by the main results
+ The ablations show that the different novelties of this paper contribute to the performance
+ Ablations provide additional insight into multiple aspects of pretraining and fine-tuning of the entire network or only the classifier
- SE-Mantis doesn't appear to have any benefit
- Mantis was only applied to classification, further experiments could apply it also to imputation or forecasting
- The limitations of the fixed number of tokes was not addressed. 2 datasets from UEA were omitted due to short sequences, is this not possible with Mantis?
- Ablating InfoNCE against other foundation model strategies (mainly imputation) with the same model would have provided further insights into foundation model training. It is not clear whether the contrastive loss improved performance

Presentation:
The overall quality of the paper is good.
- An illustration showing in more detail how the tokenizer extracts patches, and how this changes with interpolation for SE-Mantis  (or different time series lengths in general) would've been helpfull

Significance:
+ Mantis demonstrates significant improvements over the SOTA in time series classification.
+ The paper introduces multiple new techniques and provides overall many insights about foundation modelling for classification
- Application limited to classification

Originality:
+ This paper introduces a novel tokenization strategy
+ it applies contrastive predictive coding for time series foundation modeling

---

> ### Author Rebuttal · Authors · 2026-03-31
>
> We sincerely thank the reviewer for their positive evaluation of our work and for providing such insightful feedback.
>
> > SE-Mantis doesn't appear to have any benefit.
>
> We apologize for any confusion regarding the presentation of these results. On the subset of 91 UCR datasets used to compare against self-supervised baselines, self-ensembling (SE) indeed does not improve average performance. However, on the full 128 datasets, SE improves the vanilla model (Appendix B5). We view SE as an optional technique that yields significant boosts when interpolation size is critical (e.g., Car, Phoneme, Worms; Table 12). We will clarify this in the text.
>
> >Mantis was only applied to classification, further experiments could apply it also to imputation or forecasting
>
> While a universal model is ideal, we are not aware of any current model that achieves SOTA in both classification and forecasting simultaneously. This could due to the fact that forecasting and classification rely on fundamentally different cues (low-frequency global trends vs. high-frequency local patterns). In our preliminary forecasting experiments, Mantis's contrastive pre-training provided no benefit as it lacks a reconstruction concept. Combining different loss functions during pre-training to find a universally strong checkpoint is an excellent direction for future work. Furthermore, certain architectural choices tailored for classification (e.g., mean pooling) harms forecasting quality. This implies that designing a truly versatile foundation model requires complex compromises. We appreciate you raising this point and will add a discussion of these insights to the manuscript.
>
> > The limitations of the fixed number of tokens. 2 datasets from UEA were omitted... limits in terms of minimum and maximum allowed sequence length?
>
> First, we omitted three UEA datasets due to structural issues: AtrialFibrillation and StandWalkJump have only 15 test points (a single error drops accuracy by 6.7%), and PenDigits (length 8) is too short for meaningful temporal patterns. Results are below and will be added to the appendix.
>
> | Dataset | Catch22+ | TiRex | Chronos2 | Mantis|
> | :--- | :--- | :--- | :--- | :--- |
> |AtrialFibrillation| 0.2 | 0.2 | 0.133 | 0.133 |
> |PenDigits| 0.913| 0.976 | 0.974 |0.955|
> |StandWalkJump| 0.067 | 0.4 | 0.467 | 0.4|
>
> Second, regarding sequence length limits: Mantis only requires inputs to be a multiple of 32, which we achieve via linear interpolation, allowing it to handle any length. It performs strongly even on ultra-short series. The table below shows the average performance on 10 datasets with the shortest lengths: Chinatown (t=24), Crop (t=46), ItalyPowerDemand (t=24), JapaneseVowels (t=29), LSST (t=36),  Libras (t=45), MelbournePedestrian(t=24), NATOPS(t=51), RacketSports(t=30), SmoothSubspace (t=15). Mantis maintains the highest average performance.
>
> | Dataset | Catch22+ | TabPFN | TabICL | MOMENT | TiRex | Chronos2 | TiViT-H | TiConvNext | NuTime | Mantis|
> | :--- | :--- | :--- | :--- | :--- | :--- | :--- | :--- | :--- | :--- | :--- |
> |Avg | 0.856 | 0.836 |0.834 | 0.851 | 0.848 | 0.841 | 0.834 | 0.827 | 0.839 | **0.884** |
>
> Acknowledging that fixed number of tokens could be a limitation for extreme sequence lengths (very short/long), we introduced self-ensembling that can help mitigate the issue by generating multi-resolution features.
>
> > Ablating InfoNCE against other foundation model strategies (mainly imputation)...
>
> We ablated three imputation-based pre-training schemes using the Mantis architecture: (a) Patch Masked Autoencoder (MAE), (b) Token MAE, and (c) Next-patch prediction. As shown below, contrastive learning significantly outperforms these schemes. While Mantis's architecture may naturally favor contrastive learning over imputation, we will include this ablation to validate our choice of InfoNCE.
>
> | Pre-training type | Contrastive Learning | MAE (Patch Reconstruction) | MAE (Token Reconstruction) | Next Patch Prediction |
> | :--- | :--- | :--- | :--- | :--- |
> | Acc (UCR) | **0.788** | 0.716 | 0.676 | 0.707 |
>
> > Did you also try using both real and cauker data during pretraining?
>
> To prevent data leakage given the scarcity of public classification datasets, we pre-trained Mantis exclusively on synthetic data. Preliminary experiments mixing real and synthetic data showed no immediate advantage. However, combining modalities or employing continuous pre-training on large, unlabelled domain-specific corpora (e.g., EEG) remains a highly promising future direction.

---

> > ### Author Rebuttal · Reviewer_KvPU · 2026-04-01
> >
> > I thank the authors for their rebuttal and additional clarifications.
> > I acknowledge and largely agree with the points made in the rebuttal.
> >
> > In particular, the ablation regarding InfoNCE are appreciated, which further motivates a main component of the paper.
> >
> > This further strengthens my confidence in the original rating, which is to accept this paper, due to high quality, and strong performance.
> >
> > For an even higher rating, the work would need to demonstate either higher novelty and/or more significant performance margins.
> > I wish the authors all the best with this work and future research.

---

> > > ### Author Response · Authors · 2026-04-02
> > >
> > > We sincerely thank the reviewer for their continued support, the positive evaluation, and their kind well-wishes. We are glad that our additional experiments and clarifications were helpful. Your constructive feedback has undoubtedly strengthened the final version of our paper.
> > >
> > > Thank you once again for your time and valuable insights during this review process!

---

### Official Review · Reviewer_7Aue · 2026-03-15

**Soundness:** 4
**Presentation:** 2
**Significance:** 2
**Originality:** 2
**Overall Recommendation:** 4
**Confidence:** 3

**Summary:**

The paper tackles the time series classification problem. The main architecture innovation is the Token Generator Unit which combines a variety of ideas such as using a first order differencing feature, and numerically multi-scaled encodings. The model is pre-trained with a contrastive loss, on synthetic data generated by the CauKer method. The proposed approach also includes test-time tricks such as selecting intermediate representations rather then the final layer's representations, as well as ensembling with other models. Experiments demonstrate the strength of the proposed method.

**Compliance With Llm Reviewing Policy:**

Affirmed.

**Final Justification:**

concerns addressed

**Key Questions For Authors:**

* How were the intermediate layers chosen? It seems to have been chosen based on performance, what is the experimental set up to do so?
* Why was the choice of not using a linear classifier as per prior work on time series classification not being chosen, but instead using the "Standard Scaler and Logistic Regression"? Was it only for Mantis that this was used, or also for the baselines?

**Limitations:**

Limitations have not been discussed.

**Strengths And Weaknesses:**

### Strengths
* The paper provides details on the proposed method and experimental setup.
* Extensive ablations and additional details in the appendix

### Weaknesses
It is unclear to me what is the key contributions of this paper. It is quite confusing as "Mantis" does not seem to be a new model, but has already been used in the CauKer paper. However, there are no comparisons with the model from CauKer. It seems odd to me that although the CauKer paper was acknowledged, it was not being added as a baseline. As I understand it, it seems then that the innovations are mainly selecting intermediate layer representations, and combining representations with other models.

---

> ### Author Rebuttal · Authors · 2026-03-31
>
> We thank the reviewer for their careful reading of our manuscript and their constructive comments. Please find our detailed, point-by-point response below.
>
> > *Why standard scaler + log. regression?*
>
> We thank the reviewer for this interesting question, which touches upon a rather under-discussed aspect of representation learning. While simple linear probing is conventional in the deep learning community, it implicitly assumes that feature variances (across the dimensions of the embeddings) share the same order of magnitude. In our setup, this is not the case. We hypothesize two reasons for this:
>
> 1. Mantis and other modern architectures use layer normalization, which standardizes features within a single sample. However, this does not ensure that features have a consistent variance across the entire dataset.
>
> 2. We pre-train our model using a self-supervised objective. Unlike supervised learning (where the linear head is co-trained as part of the architecture), self-supervised pre-training does not guarantee similarly distributed features. Consequently, standard scaling or batch normalization is generally recommended when probing self-supervised models (Marks et al., 2024).
>
> | Sklearn Log. Regression (L-BFGS-B Opt.)  | |                | PyTorch Linear (Adam Opt.) |            |            | Random Forest |
> |------------------------------------------|-----------------------------------------|-----------------|----------------------------|------------|------------|---------------|
> | W/o Norm                                 | MinMax Scaler                           | Standard Scaler | W/o Norm                   | Layer Norm | Batch Norm |               |
> |  0.763                                | 0.829                                | **0.837**        | 0.669                   | 0.71    | 0.827   | 0.82      |
>
> To empirically support our reasoning, we compared several probing approaches (see Table below). We evaluated a scikit-learn logistic regression (optimized by L-BFGS-B) without normalization, with a MinMaxScaler, and with a Standard Scaler (our chosen method). We also tested a PyTorch logistic regression (nn.Linear, Cross-Entropy loss, Adam optimizer) without pre-normalization, with a layer norm step, and with a batch norm step. A random forest classifier was included as a reference. The results show that performance degrades drastically without normalization. While layer pre-normalization helps, it does not fully resolve the issue, which supports our first hypothesis. Conversely, scaling features over the entire dataset or batch yields superior results. Because the Standard Scaler baseline slightly outperformed both the MinMaxScaler and Batch Normalization, we adopted this pipeline for our setup and used it for all time series foundation model baselines. We will expand on this rationale in the revised manuscript.
>
> (Marks et al., 2024) A Closer Look at Benchmarking Self-Supervised Pre-training with Image Classification. arXiv:2407.1221
>
> > *How were the intermediate layers chosen?*
>
> Unlike traditional ML setups where optimal hyperparameters are tuned per dataset, we identify a single optimal intermediate layer for each model just once. Specifically, we select the layer that yields the highest average validation accuracy across the 128 UCR datasets. This layer is then fixed and used to generate embeddings for all downstream benchmarks (UCR, UEA, HAR, and EEG). Interestingly, this model selection is consistent: average accuracy across UCR validation sets, UCR test sets, and UEA test sets are all maximized using the embeddings from the 3rd layer for Mantis. A similar conclusion holds true for Chronos 2. We have included this experiment in the revised version of the paper.
>
> > *cauker vs mantis.*
>
> We would like to clarify that CauKer does not propose a model architecture for time series classification; its contribution is strictly an algorithm for generating synthetic time series data. We will make the distinction clearer in the text to avoid any confusion regarding the novelty of Mantis.

---

> > ### Author Rebuttal · Reviewer_7Aue · 2026-04-04
> >
> > -

---

> > > ### Author Response · Authors · 2026-04-07
> > >
> > > We sincerely thank the reviewer for taking the time to read our rebuttal, confirming that our response fully resolved your concerns, and raising your score.
> > >
> > > We truly appreciate the comments of the reviewer that has helped us significantly improve our manuscript. Thank you again for your time and effort during this review process!

---

### Decision · Program_Chairs · 2026-04-30

**Decision:**

Accept (regular)

**Comment:**

All four reviewers recommend acceptance. They find the paper sound, clearly written, and well-evaluated. Key strengths include: a novel tokenization strategy, contrastive predictive coding for time series foundation models, extensive ablations, strong reproducibility, and state-of-the-art performance on UEA‑27. Minor suggestions (e.g., testing on imputation/forecasting, addressing the fixed token length limitation, clarifying the tokenizer illustration) do not undermine the overall positive assessment.

Overall, the paper makes a solid contribution to lightweight foundation models for time series classification, which is a relatively underexplored area. The methodology is clearly explained, the experiments are thorough, and the results convincingly support the claims. The limitations noted by reviewers are acknowledged but do not prevent acceptance.